# "Why Not Looking backward?" A Robust Two-Step Method to Automatically Terminate Bayesian Optimization

**Shuang Li**
Control and Simulation Center, Harbin Institute of Technology, China.
National Key Laboratory of Modeling and Simulation for Complex Systems, China.
ShuangLi.hit@outlook.com

**Ke Li**
Department of Computer Science
University of Exeter, EX4 4RN, Exeter, UK.
k.li@exeter.ac.uk

**Wei Li**[*]
Control and Simulation Center, Harbin Institute of Technology, China.
National Key Laboratory of Modeling and Simulation for Complex Systems, China.
frank@hit.edu.cn

## Abstract

Bayesian Optimization (BO) is a powerful method for tackling expensive black-box optimization problems. As a sequential model-based optimization strategy, BO iteratively explores promising solutions until a predetermined budget, either iterations or time, is exhausted. The decision on when to terminate BO significantly influences both the quality of solutions and its computational efficiency. In this paper, we propose a simple, yet theoretically grounded, two-step method for automatically terminating BO. Our core concept is to proactively identify if the search is within a convex region by examining previously observed samples. BO is halted once the local regret within this convex region falls below a predetermined threshold. To enhance numerical stability, we propose an approximation method for calculating the termination indicator by solving a bilevel optimization problem. We conduct extensive empirical studies on diverse benchmark problems, including synthetic functions, reinforcement learning, and hyperparameter optimization. Experimental results demonstrate that our proposed method saves up to $\approx 80\%$ computational budget yet is with an order of magnitude smaller performance degradation, comparing against the other peer methods. In addition, our proposed termination method is robust in terms of the setting of its termination criterion.

## 1 Introduction

*"Nature does not hurry, yet everything is accomplished."* — Lao Tzu

In this paper, we consider the black-box optimization problem (BBOP) defined as follows:

$$\underset{\mathbf{x}\in\Omega}{\text{maximize}} \ f(\mathbf{x}), \tag{1}$$

---

[*]Wei Li is the corresponding author of this paper.

37th Conference on Neural Information Processing Systems (NeurIPS 2023).

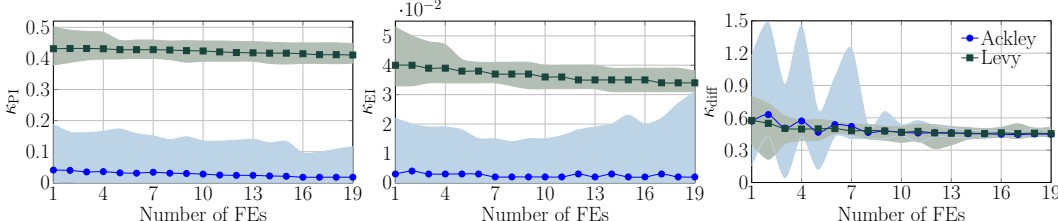

Figure 1: Trajectories of termination criteria used in [22], [28] and [24] on Ackley and Levy function where $n = 1$. Results are collected from 21 independent runs of vanilla BO while the mean value of termination indicator of each termination criterion is plotted as the solid line associated with the confidence interval. Please refer to Section 3.2 for a description of these termination criteria, as well as the meaning of $\kappa_{\mathrm{PI}}$, $\kappa_{\mathrm{EI}}$ and $\kappa_{\mathrm{diff}}$.

where $\mathbf{x} = (x_1, \cdots, x_n)^\top$ is a decision vector (variable), $\Omega = [x_i^{\mathrm{L}}, x_i^{\mathrm{U}}]_{i=1}^n \subset \mathbb{R}^n$ represents the search space, and $f : \Omega \to \mathbb{R}$ corresponds to the attainable set in the objective space. In real-world scenarios, function evaluations (FEs) of $f(\mathbf{x})$ can be costly, giving rise to expensive BBOPs. Bayesian optimization (BO) has emerged as one of the most effective methods for addressing expensive BBOPs. BO is a sequential model-based optimization technique consisting of two iterative steps: $i$) employing limited expensive FEs to construct a surrogate model of the physical objective function, such as a Gaussian process (GP) model [35]; and $ii$) selecting the next point of interest for costly FE by optimizing an acquisition function, e.g., probability of improvement (PI) [18], expected improvement (EI) [16], and upper confidence bound (UCB) [31]. Numerous theoretical and methodological advancements have been made in BO. Interested readers can refer to comprehensive survey papers [29, 11] and a recent textbook [13] for further information.

Nevertheless, the question of when to terminate the search process of BO remains a largely underexplored area in the literature. At present, the most prevalent termination criterion is a pre-specified budget, such as the number of FEs or wall-clock time. Though intuitive, this approach neglects the search dynamics inherent to different BBOPs. As a result, this strategy is rigid while it does not offer a general rule for determining an appropriate budget across various problem settings. If the budget is too small, BO may terminate prematurely, yielding a suboptimal solution. On the contrary, an excessive budget may lead to wasted computational resources. Another simple termination method involves stopping BO if the current best solution remains unchanged for a predetermined number of consecutive FEs. However, as highlighted by [24], this strategy also fails to consider the observed data during the sequential model-based optimization process and relies on a pre-defined threshold.

Beyond the aforementioned 'naïve' approaches, a limited number of dedicated efforts have been made to address the termination of BO. One notable method involves monitoring the progress of BO by termination indicators, such as the maximum of EI [28, 16] or PI [22]. In this approach, BO is terminated when the corresponding termination indicator falls below a pre-specified threshold. Very recently, Makarova *et al.* proposed using the difference between the minimal of the lower confidence bound (LCB) and UCB as the termination indicator. As illustrated in Figure 1, we observe that all criteria used in these termination approaches exhibit significant oscillation during the optimization process. This can be attributed to: $i$) the stochastic nature of BO itself, and $ii$) numerical errors arising from the non-convex optimization of acquisition functions. Furthermore, as shown in Figures 1(a) and (b), the variation range of the same criterion can differ substantially when addressing problems with distinct fitness landscapes. These factors make determining a universally applicable threshold in practice challenging, resulting in fragile and less intuitive termination criteria compared to simply establishing a budget. Additionally, we find that these termination criteria are 'myopic', as decision-making is based solely on the observations at the current step, leading to a lagged termination. For instance, consider the selected samples shown in Figure 2; it is difficult, if not impossible, to determine when to terminate BO until $t = 20$. However, if we look backward to $t = 5$, it becomes evident that BO is likely to converge by $t = 10$.

**Our contributions.**  In light of the aforementioned challenges, this paper proposes a novel termination method for BO that proactively detects whether the search is located in a convex region of $-f(\mathbf{x})$ by examining previously observed samples. BO is terminated if the local regret within this convex region falls below a predetermined threshold. To improve numerical stability, we introduce an approx-

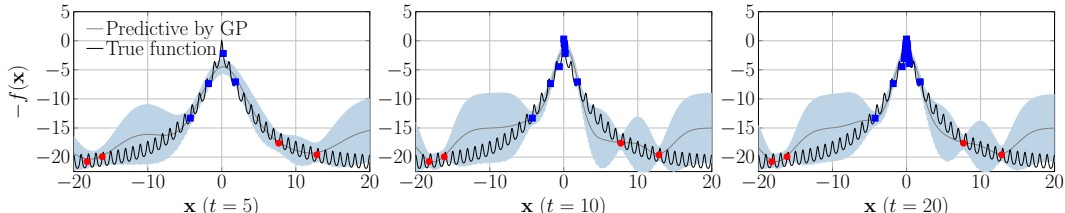

Figure 2: Search dynamics of vanilla BO on the Ackley function ($n = 1$) at different time steps after the initialization. In particular, $t = 5$ indicates five new samples are collected after the initialization.

imation method for calculating the termination indicator by solving a bilevel optimization problem. Our proposed termination method is simple, yet it offers theoretical guarantees. To demonstrate its effectiveness, we compare the performance of our proposed method against four peer methods on a variety of benchmark problems, encompassing synthetic functions, reinforcement learning, and hyperparameter optimization.

## 2 Proposed Method

This section starts with a gentle tutorial of vanilla BO. Then, we delineate the implementation of our proposed termination method, followed by a theoretical analysis at the end.

### 2.1 Vanilla Bayesian Optimization

As a gradient-free optimization method, BO comprises two major steps. The first step involves constructing a surrogate model based on GP to approximate the expensive objective function. Given a set of training data $\mathcal{D} = \{\langle \mathbf{x}^i, f(\mathbf{x}^i) \rangle\}_{i=1}^N$, GP learns a latent function $g(\mathbf{x})$, such that $\forall \mathbf{x} \in \mathcal{D}$, we have $f(\mathbf{x}) = g(\mathbf{x}) + \epsilon$, where $\epsilon \sim \mathcal{N}(0, \sigma_\epsilon^2)$ is an i.i.d. Gaussian noise. For each testing input vector $\mathbf{z}^* \in \Omega$, the mean and variance of the target $f(\mathbf{z}^*)$ are predicted as follows:

$$
\begin{aligned}
\mu(\mathbf{z}^*) &= \mathbf{k}^{*\top}(K + \sigma_\epsilon^2 I)^{-1}\mathbf{f}, \\
\sigma^2(\mathbf{z}^*) &= \mathbf{k}(\mathbf{z}^*, \mathbf{z}^*) - \mathbf{k}^{*\top}(K + \sigma_\epsilon^2 I)^{-1}\mathbf{k}^*,
\end{aligned}
\tag{2}
$$

where $X = (\mathbf{x}^1, \cdots, \mathbf{x}^N)^\top$ and $\mathbf{f} = (f(\mathbf{x}^1), \cdots, f(\mathbf{x}^N))^\top$. $\mathbf{k}^*$ is the covariance vector between $X$ and $\mathbf{z}^*$, and $K$ is the covariance matrix of $X$. In this paper, we use the Matérn $5/2$ kernel as the covariance function to measure the similarity between a pair of data points. The second step consists of an infill criterion based on the optimization of an acquisition function, which determines the next point of merit $\tilde{\mathbf{x}}^*$ to be evaluated by the actual expensive objective function:

$$
\tilde{\mathbf{x}}^* = \operatorname*{argmax}_{\mathbf{x} \in \Omega} f^{\text{acq}}(\mathbf{x}).
\tag{3}
$$

where $f^{\text{acq}}(\mathbf{x}) = \mu(\mathbf{x}) + \omega\sigma(\mathbf{x})$ is the widely used UCB [31] to facilitate our theoretical analysis. Specifically, the parameter $\omega > 0$, determined according to the confidence level set as $0.95$ in this paper, controls the trade-off between exploration and exploitation. Subsequently, the next point of merit $\tilde{\mathbf{x}}^*$ is used to update the training dataset as $\mathcal{D} = \mathcal{D} \bigcup \{\tilde{\mathbf{x}}^*\}$, and BO iterates between the two aforementioned steps sequentially until a termination criterion is met. The convergence of BO can be evaluated by regret:

$$
r = f(\mathbf{x}^\star) - f(\tilde{\mathbf{x}}^\star),
\tag{4}
$$

where $\mathbf{x}^\star$ represents the ground truth global optimum and $\tilde{\mathbf{x}}^\star = \operatorname*{argmax}_{\mathbf{x} \in \mathcal{D}} f(\mathbf{x})$ denotes the current best-found solution.

### 2.2 Proposed Termination Criterion

Inspired by the observations illustrated in Figure 2, we propose a termination method that involves 'looking back' at the last $\tau > 1$ observed points in the dataset $\mathcal{D}$, and storing these in a temporary archive, denoted as $\tilde{\mathcal{D}}$. The termination criterion we propose is predicated on two primary conditions.

**Condition 1.** *The BO search process is deemed to have converged within a convex hull $\tilde{\Omega}$ if the following condition is satisfied:*

$$
\sum_{j=1}^{\binom{\tau+1}{2}} \mathbb{1}\left(\mu\left(\frac{\mathbf{x} + \mathbf{x}'}{2}\right) \geq \frac{f(\mathbf{x}) + f(\mathbf{x}')}{2}\right) = \binom{\tau+1}{2},
\tag{5}
$$

where $\mathbb{1}(\cdot)$ denotes the indicator function, returning $1$ if the argument holds true and $0$ otherwise. $\mathbf{x}$ and $\mathbf{x}'$ are points selected randomly and distinctively from $\tilde{\mathcal{D}}$. The convex hull, $\tilde{\Omega} = [\tilde{x}_i^{\mathrm{L}}, \tilde{x}_i^{\mathrm{U}}]_{i=1}^n$, is a subset of $\Omega$, where $\tilde{x}_i^{\mathrm{L}} = \underset{\mathbf{x} \in \tilde{\mathcal{D}}}{\operatorname{argmin}}\, x_i$ and $\tilde{x}_i^{\mathrm{U}} = \underset{\mathbf{x} \in \tilde{\mathcal{D}}}{\operatorname{argmax}}\, x_i$.

**Condition 2.** *Assuming Condition 1 is satisfied, and $\tilde{\mathbf{x}}$ denotes the most recently observed point in $\mathcal{D}$, we calculate the local regret $\tilde{r}$ as follows:*

$$\tilde{r} = \mu(\dot{\mathbf{x}}) - \mu(\tilde{\mathbf{x}}) + \omega\left(\sigma(\ddot{\mathbf{x}}) + \sigma(\tilde{\mathbf{x}})\right), \tag{6}$$

*where $\dot{\mathbf{x}} = \underset{\mathbf{x} \in \tilde{\Omega}}{\operatorname{argmax}}\, \mu(\mathbf{x})$ and $\ddot{\mathbf{x}} = \underset{\mathbf{x} \in \tilde{\Omega}}{\operatorname{argmax}}\, \sigma^2(\mathbf{x})$. The BO process terminates if the following inequality is satisfied:*

$$\frac{\tilde{r}}{\omega\sigma_\epsilon} \leq \eta_{\mathrm{lb}}, \tag{7}$$

*where $\frac{\tilde{r}}{\omega\sigma_\epsilon}$ is used as the termination indicator, denoted as $\kappa_{\mathrm{lb}}$, and $\eta_{\mathrm{lb}}$ is a predetermined threshold.*

**Remark 1.** *The inequality within the indicator function $\mathbb{1}(\cdot)$ in equation (5) is derived from Jensen's inequality [4], which yields a convex function:*

$$-f(\alpha\mathbf{x} + (1-\alpha)\mathbf{x}') \leq -\alpha f(\mathbf{x}) - (1-\alpha)f(\mathbf{x}'), \tag{8}$$

*where $\alpha \in [0,1]$ and $\mathbf{x}, \mathbf{x}' \in \tilde{\Omega}$. In order to avoid the necessity of additional function evaluations when computing $f(\frac{\mathbf{x}+\mathbf{x}'}{2})$, we substitute $\mu(\frac{\mathbf{x}+\mathbf{x}'}{2})$ into equation (5).*

**Remark 2.** *In equation (6), we employ the widely-used L-BFGS algorithm [6] to compute $\dot{\mathbf{x}}$ and $\ddot{\mathbf{x}}$. To ensure numerical stability, we suggest the following strategies for initializing the algorithm and defining its termination criterion:*

1. *For $\dot{\mathbf{x}}$, L-BFGS is initialized at a point randomly selected from $\tilde{\Omega}$. The algorithm terminates when $\|\bigtriangledown \mu(\mathbf{x})\|_2 \leq \lambda$. In our work, we set $\lambda = 10^{-6}$, following Proposition 1.*

2. *For $\ddot{\mathbf{x}}$, L-BFGS is initialized at the point $\underset{\mathbf{x} \in \tilde{\Omega}}{\operatorname{argmax}}\, \underline{\sigma}^2(\mathbf{x})$, where $\underline{\sigma}^2(\mathbf{x})$ denotes the lower bound of $\sigma^2(\mathbf{x})$ over $\tilde{\Omega}$. The termination criterion is $\|\bigtriangledown \sigma^2(\mathbf{x})\|_2 \leq \lambda$, as per Proposition 2.*

**Remark 3.** *Considering equation (7), given that $\frac{\mu(\dot{\mathbf{x}}) - \mu(\ddot{\mathbf{x}})}{\omega\sigma_\epsilon} \geq 0$ and $\frac{\sigma(\dot{\mathbf{x}}) + \sigma(\ddot{\mathbf{x}})}{\sigma_\epsilon} \geq 2$, we deduce that $\eta_{\mathrm{lb}} \geq 2$. The upper bound of $\eta_{\mathrm{lb}}$ is empirically determined, as detailed in Section 4.1.*

**Remark 4.** *When the GP model is overfitting, BO tends to converge within the local region of the current best solution. In this case, both Condition 1 and Condition 2 are easily met while BO will be terminated prematurely. On the other hand, when the model is underfitting, BO will explore $\Omega$ in a random manner. In this case, satisfying Condition 1 becomes challenging, and BO will face the risk of failing to be terminated. Therefore, we designed three mitigation strategies: 1) restrict the lengthscale to $[0.05, 200]$ during GP training to prevent lengthscales from becoming excessively large or small; 2) normalize the input of training data to $[0,1]$; and 3) standardize the output of the training data by centering it on the mean and scaling it by the variance.*

**Proposition 1.** *Consider $\forall \mathbf{x} \in \tilde{\Omega}$, where $-\mu(\mathbf{x})$ represents a convex function. If $\|\bigtriangledown \mu(\mathbf{x})\|_2 \leq \lambda$, we can establish:*

$$\mu(\dot{\mathbf{x}}) - \mu(\mathbf{x}) \leq \xi, \tag{9}$$

*where $\lambda = (2m_1\xi)^{1/2}$, $\xi$ is a positive constant, and $m_1$ denotes the strong convexity parameter of $-\mu(\mathbf{x})$ [4].*

**Lemma 1.** *Assume the GP employs a stationary kernel $k(\cdot, \cdot)$. For $\forall \mathbf{x} \in \tilde{\Omega}$, the lower bound of $\sigma^2(\mathbf{x})$ is given by:*

$$\underline{\sigma}^2(\mathbf{x}) = k(\mathbf{x}, \mathbf{x}) + c\sum_{i=1}^{|\mathcal{D}|} k^2(\mathbf{x}, \mathbf{x}^i), \tag{10}$$

*where $c < 0$ is a constant and $\mathbf{x}^i \in \mathcal{D}$ for $i \in \{1, \cdots, |\mathcal{D}|\}$.*

**Lemma 2.** *Given Lemma 1, determining* $\underset{\mathbf{x} \in \tilde{\Omega}}{\arg\max} \underline{\sigma}^2(\mathbf{x})$ *is equivalent to solving the following bilevel optimization problem:*

$$
\begin{aligned}
\underset{\mathbf{x} \in \tilde{\Omega}}{\text{minimize}} \quad & d(\mathbf{x}, \mathbf{x}^1, \mathbf{x}^2) & = \|\mathbf{x} - \mathbf{x}^1\|_2^2 + \|\mathbf{x} - \mathbf{x}^2\|_2^2 \\
\text{subject to} \quad & \{\mathbf{x}^1, \mathbf{x}^2\} & = \underset{\substack{\mathbf{x}^1, \mathbf{x}^2 \in \mathcal{D} \cap \tilde{\Omega} \\ \mathbf{x}^1 \neq \mathbf{x}^2, \, \hat{\Omega} \cap \mathcal{D} = \emptyset}}{\arg\max} \|\mathbf{x}^1 - \mathbf{x}^2\|_2^2,
\end{aligned}
\tag{11}
$$

*where* $\hat{\Omega} = [\hat{x}_i^{\mathrm{L}}, \hat{x}_i^{\mathrm{U}}]_{i=1}^n \subset \tilde{\Omega}$, $\hat{x}_i^{\mathrm{L}} = \min(x_i^1, x_i^2)$ *and* $\hat{x}_i^{\mathrm{U}} = \max(x_i^1, x_i^2)$. *Given that the lower-level optimization can be addressed via exhaustive search, the analytical solution of* (11) *is given by* $\hat{\mathbf{x}} = (\hat{x}_1^{\mathrm{L}} + \frac{\hat{x}_1^{\mathrm{U}} - \hat{x}_1^{\mathrm{L}}}{2}, \cdots, \hat{x}_n^{\mathrm{L}} + \frac{\hat{x}_n^{\mathrm{U}} - \hat{x}_n^{\mathrm{L}}}{2})^\top$.

**Proposition 2.** *Leveraging Lemma 2, suppose* $\underset{\mathbf{x} \in \tilde{\Omega}}{\text{minimize}} -\sigma^2(\mathbf{x})$ *exhibits convexity in its local optimal regions, the following inequality is satisfied when* $\|\nabla \sigma^2(\mathbf{x})\|_2 \leq \lambda$:

$$
\sigma^2(\ddot{\mathbf{x}}) - \sigma^2(\mathbf{x}) \leq \beta + \xi,
\tag{12}
$$

*where* $\lambda = (2m_2\xi)^{1/2}$, $\xi > 0$, $m_2 > 0$ *represents the strong convexity parameter of* $-\sigma^2(\mathbf{x})$ *in its local optimal regions [4], and* $\beta$ *is constrained by* $0 \leq \beta \leq \sigma^2(\ddot{\mathbf{x}}) - \sigma^2(\hat{\mathbf{x}})$.

### 2.3 Theoretical Analysis of the Proposed Termination Criterion

In this subsection, we delve into the theoretical underpinnings of the proposed termination method, focusing on the convergence of BO when the UCB is utilized as the acquisition function.

**Lemma 3.** *As per Srinivas* et al.*, the optimization process in BO can be conceptualized as a sampling process from a GP. Hence, for any* $\mathbf{x} \in \Omega$*, we have:*

$$
\Pr\left(|f(\mathbf{x}) - \mu(\mathbf{x})| \leq \omega\sigma(\mathbf{x})\right) > \delta,
\tag{13}
$$

*where* $\delta > 0$ *signifies the confidence level adhered to by the UCB.*

**Corollary 1.** *Based on Lemma 3 and Condition 2, we deduce that:*

$$
\Pr\left(f^{\mathrm{acq}}(\tilde{\mathbf{x}}^\star) + \varepsilon \geq f(\mathbf{x}^\star)\right) > \delta,
\tag{14}
$$

*where* $\varepsilon$ *is a numerical error when optimizing the acquisition function,* $\tilde{\mathbf{x}}^\star = \underset{\mathbf{x} \in \Omega}{\arg\max}\, f^{\mathrm{acq}}(\mathbf{x})$*, and* $\mathbf{x}^\star$ *represents the true global optimum. Furthermore,*

$$
0 \leq \varepsilon \leq \mu(\dot{\mathbf{x}}) + \omega\sigma(\ddot{\mathbf{x}}) - f^{\mathrm{acq}}(\tilde{\mathbf{x}}^\star),
\tag{15}
$$

*where* $\dot{\mathbf{x}}$*,* $\ddot{\mathbf{x}}$*, and* $\tilde{\mathbf{x}}^\star$ *are elements of* $\tilde{\Omega}$*, while* $\delta > 0$ *denotes the confidence level of the UCB.*

**Theorem 1.** *Leveraging Corollary 1, when employing the termination method proposed in this paper, we deduce that the global regret bound of BO as:*

$$
\Pr\left(r \leq 2\omega\sigma(\tilde{\mathbf{x}}^\star) + \varepsilon\right) > \delta,
\tag{16}
$$

*where* $\delta > 0$ *signifies the confidence level associated with the UCB.*

**Theorem 2.** *Building upon Condition 1 and Condition 2, and employing the termination method proposed in this paper, we establish the local regret bound of BO as:*

$$
\Pr\left(f(\mathbf{x}^\star) - f(\mathbf{x}) \leq \tilde{r}\right) > \delta,
\tag{17}
$$

*where* $\mathbf{x} \in \tilde{\Omega}$*,* $\mathbf{x}^\star$ *denotes the true global optimum in* $\tilde{\Omega}$*, and* $\delta > 0$ *is the confidence level of the UCB.*

**Remark 5.** *Drawing from Theorem 1 and Theorem 2, we observe that if* $\varepsilon$ *can be considered negligible when* $\tilde{\mathbf{x}}^\star$ *is accurately determined by optimizing the UCB,* $\tilde{r}$ *subsequently represents the upper bound of BO regret within the domain* $\Omega$*. Conversely, if* $\varepsilon$ *cannot be disregarded,* $\tilde{r}$ *is posited as the upper bound of BO regret within the restricted domain* $\tilde{\Omega}$*.*

## 3 Experimental Settings

In this section, we present the experimental setup for our empirical study, which encompasses the benchmark test problems, the peer algorithms, and the performance metrics used for evaluation.

### 3.1 Benchmark Problems

We evaluate the performance of our proposed method on three types of benchmark problems.

- Synthetic functions: We consider Ackley, Levy, and Schwefel functions [33] with $n \in \{2, 5, 10\}$. The objective function $f(\mathbf{x})$ is contaminated by Gaussian noise $\zeta \sim \mathcal{N}(0.0, 0.2)$. The maximal number of FEs is set to $N_{\mathrm{FE}} = 50n$, with $5n$ allocated to initialization.

- Reinforcement learning (RL): We examine two RL tasks chosen from OpenAI Gym [5]: Lunar Lander with $n = 12$ and Swimmer with $n = 16$. We set $N_{\mathrm{FE}} = 50n$, with $5n$ FEs allocated to initialization.

- Hyperparameter optimization (HPO): We consider 5 HPO tasks picked up from the HPOBench [9] for tuning support vector machine (SVM) with $n = 2$, multi-layer perceptron (MLP) with $n = 5$, random forest with $n = 4$ and XGBoost with $n = 8$. The computational budget is set the same as in the RL tasks.

Note that, due to the use of termination criteria, it may not be necessary to exhaust the entire allocated computational budget to terminate BO. To ensure statistical significance, each experiment is independently conducted 21 times with different random seeds.

### 3.2 Peer Algorithms

As discussed in Section 1, the termination criterion for BO is an understudied topic in the literature. In our experiments, we compare our proposed method with the following four termination methods.

- Naïve method: This method ceases BO when $\tilde{\mathbf{x}}^\star$ stays unchanged for $\kappa_{\mathrm{n}}$ consecutive iterations. Here, $\kappa_{\mathrm{n}}$ is also the termination indicator. In our experiments, we test three settings of the thresholds $\eta_{\mathrm{n}}$ as 150, 337 and 524, respectively.

- Nguyen's method [28]: In each iteration of BO, the optimization of acquisition function produces the current optimal EI. By using this as the termination indicator, denoted as $\kappa_{\mathrm{EI}}$, the Nguyen's method terminates BO when it falls below a predetermined threshold $\eta_{\mathrm{EI}}$. In our experiments, we consider three settings of $\eta_{\mathrm{EI}}$ as 0.01, 0.04 and 0.06, respectively.

- Lorenz's method [22]: Analogous to the Nguyen's method, the Lorenz's method replaces the EI with PI as the termination indicator, denoted as $\kappa_{\mathrm{PI}}$. In our experiments, the termination threshold $\eta_{\mathrm{PI}}$ is set as 0.07, 0.2 and 0.33, respectively.

- Makarova's method [24]: Similar to the previous two methods, the Makarova's method uses the difference between the lower and upper confidence bounds as the termination indicator, denoted as $\kappa_{\mathrm{diff}}$. It terminates BO when $\kappa_{\mathrm{diff}} \le \eta_{\mathrm{diff}}$, a predetermined threshold and is set as 0.26, 0.62 and 0.97, respectively, in our experiments.

- Our proposed method: According to Condition 1 and Condition 2, our proposed method terminates BO when $\kappa_{\mathrm{lb}}$ falls below a predetermined threshold $\eta_{\mathrm{lb}}$, which is set as 2.02, 2.05 and 2.08, respectively. Furthermore, we introduce a hyperparameter $\tau$ to control the number of observed samples being looked backward, which is set to $\tau = 10$ in our experiments. The code is available at `https://github.com/COLA-Laboratory/OptimalStoping_NeurIPS2023`.

According to the aforementioned settings, it is evident that the naïve method tends to delay termination when a large $\eta_{\mathrm{n}}$ is used. On the other hand, other methods may incur a delayed termination if a small threshold is used. Note that the choices of the corresponding termination thresholds and the sensitivity of $\tau$ are empirically examined in Sections 4.1 and 4.2.

### 3.3 Performance Metrics

In our experiments, we consider the following three performance metrics to measure the effectiveness of a termination method.

- Empirical cumulative probability of a termination indicator:

$$\mathrm{I}_{\mathrm{cdf}} = \frac{1}{N_{\mathrm{FE}} \times 21} \sum_{i=0}^{N_{\mathrm{FE}} \times 21} \mathbb{1}(\kappa \le \tilde{\kappa}_i), \qquad (18)$$

where $\tilde{\kappa}_i = \underline{\kappa} + \frac{(\bar{\kappa} - \underline{\kappa}) \times i}{N_{\mathrm{FE}} \times 21}$, and $i \in \{0, \cdots, N_{\mathrm{FE}} \times 21\}$. For a given termination method, $\kappa$ represents its termination indicator as outlined in Section 3.2. The minimum and maximum

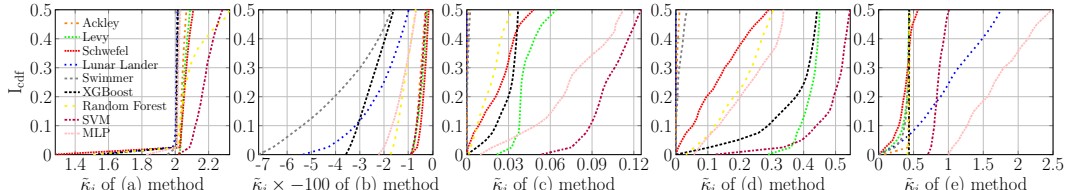

Figure 3: Trajectories of $I_{cdf}$ collected on different benchmark problems. Here we only show some results without loss of generality, while full results can be found in the supplementary document. Different subplots are (a) our proposed method, (b) Naïve method, (c) Nguyen's method, (d) Lorenz's method, and (e) Makarova's method, respectively.

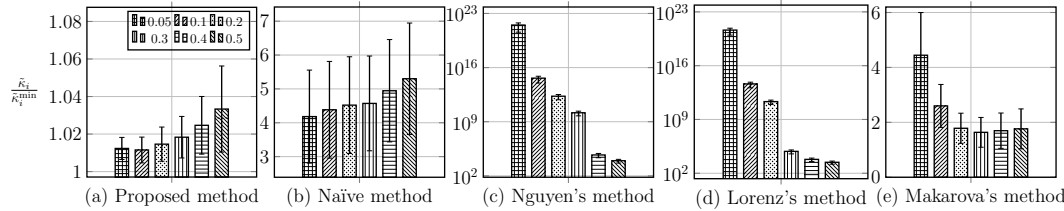

Figure 4: Bar charts with error bars of normalized $\tilde{\kappa}_i$ for different termination methods when $I_{cdf}$ is set as 0.05, 0.1, 0.2, 0.3, 0.4, and 0.5 respectively.

values of $\kappa$, represented by $\underline{\kappa}$ and $\bar{\kappa}$ respectively, are determined across all 21 repeated experiments on each benchmark problem. If $I_{cdf}$ exhibits consistency across a range of benchmark problems, it implies that the threshold choice for the corresponding termination method is consistent and not dependent on the specific problem.

- The relative computational cost:

$$I_{cost} = \frac{\tilde{N}_{FE}}{N_{FE}}, \tag{19}$$

where $\tilde{N}_{FE}$ is the number of FEs used by a termination criterion when early stopping occurs. A lower value of $I_{cost}$ indicates a higher degree of computational budget saving.

- The relative performance degradation incurred by early stopping:

$$I_{perf} = \frac{f(\bar{\mathbf{x}}) - f(\tilde{\mathbf{x}}^{\star})}{f(\bar{\mathbf{x}}) - f(\underline{\mathbf{x}})}, \tag{20}$$

where $\bar{\mathbf{x}}$ and $\underline{\mathbf{x}}$ are the best and the worst solutions found by BO when consuming all $N_{FE}$ FEs. $\tilde{\mathbf{x}}^{\star}$ signifies the best solution found when early stopping is prompted by a termination criterion. A smaller $I_{perf}$ value indicates less performance degradation resulting from the application of the corresponding termination criterion.

## 4 Empirical Studies

In this section, our experiments[2] aim to investigate three aspects: $i$) the robustness of the termination threshold for different termination methods; $ii$) the trade-off between the computational budget saving versus the performance degradation; and $iii$) the sensitivity of $\tau$ in our proposed termination method.

### 4.1 Robustness of the Selection of Termination Threshold

In this subsection, we use the $I_{cdf}$ metric to scrutinize the threshold choice of various termination methods across different problems. As per equation (18), it is evident that $I_{cdf} \propto \tilde{\kappa}_i$. As discussed earlier in Section 3.2, a large $\tilde{\kappa}_i$ can lead to premature early stopping. Consequently, we confine our analysis to instances where $I_{cdf} \leq 0.5$. As shown in Figure 3, the trajectories of $I_{cdf}$ for our proposed method appear to converge, whereas those for the other methods diverge with different magnitudes. More specifically, as shown in Figure 3(a), $\tilde{\kappa}_i = 2$ can be regarded as a transition point where $I_{cdf} \geq 0.95$ if $\tilde{\kappa}_i \geq 2$. This empirical observation corroborates the theoretical result derived

---

[2]Due to page limits, additional ablation experiments can be found in the supplementary document.

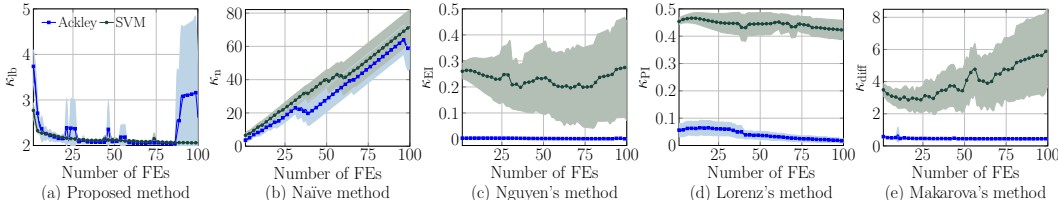

Figure 5: Trajectories of different termination indicators versus the number of FEs during the BO process on Ackley ($n = 2$) and HPO for SVM.

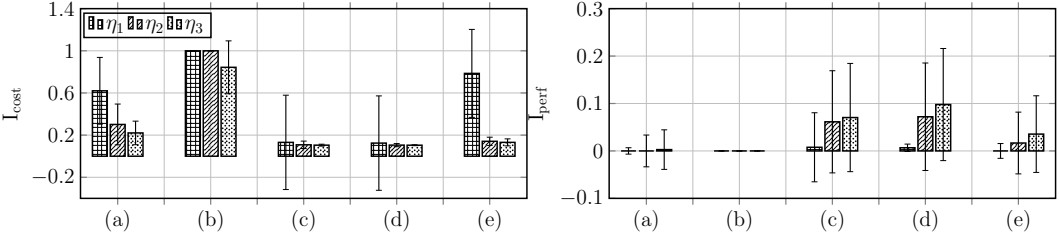

Figure 6: Bar charts with error bars of $\mathrm{I}_{\mathrm{cost}}$ and $\mathrm{I}_{\mathrm{perf}}$ obtained by using different settings of termination threshold suggested in Section 3.2, denoted as $\eta_1$, $\eta_2$ and $\eta_3$ respectively. Subplots (a) to (e) correspond to our proposed, Naïve, Nguyen's, Lorenz's, and Makarova's methods respectively.

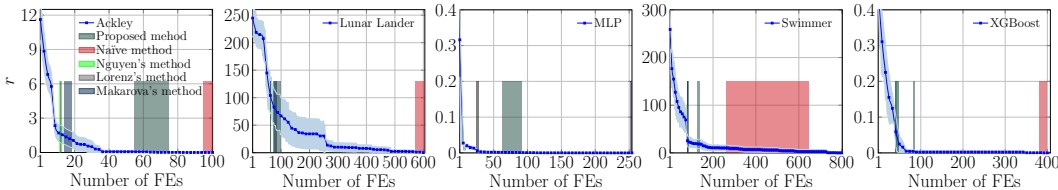

Figure 7: Trajectories of the regret of BO versus the number of FEs during the BO process on five selected problems. Full results can be found in the supplementary document.

in Condition 2. In contrast, there do not exist a consistent lower bound for the other termination methods. To further elucidate these observations, we plot the distributions of $\tilde{\kappa}_i$ when $\mathrm{I}_{\mathrm{cdf}}$ ranges from 0.05 to 0.5 in Figure 4. It is clear that the bar charts exhibit the least variation for our proposed method. For the naïve method, $\tilde{\kappa}_i$ increases as $\mathrm{I}_{\mathrm{cdf}}$ grows. However, the bars for the other three methods show significant fluctuations, particularly for the Nguyen's and the Lorenz's methods. These observations are further substantiated by the trajectories of the termination indicators throughout the BO process, as shown in Figure 5. We present results for the Ackley and HPO for SVM problems here, while complete results are available in the supplementary document. These plots reveal that the trajectories for our proposed method converge to a certain threshold, while those for the other methods not only diverge but also differ significantly on different problems. Based on this discussion, we use $\mathrm{I}_{\mathrm{cdf}} = 0.05$ as the capping point to guide the selection of the termination threshold for different termination methods: $\eta_{\mathrm{lb}} \in [2, 2.1]$, $\eta_{\mathrm{n}} \in [57, 617]$, $\eta_{\mathrm{EI}} \in [3.8 \times 10^{-24}, 0.08]$, $\eta_{\mathrm{PI}} \in [2 \times 10^{-21}, 0.39]$, $\eta_{\mathrm{diff}} \in [0.09, 1.15]$. In our experiments, we apply the Latin hypercube design method [26] to choose three settings as listed in Section 3.2.

## 4.2 Computational budget saving versus performance degradation

There is a trade-off when early terminating BO, i.e., the performance of BO can be compromised when using less FEs. In this subsection, we employ $\mathrm{I}_{\mathrm{cost}}$ and $\mathrm{I}_{\mathrm{perf}}$ to characterize such trade-off. From the comparison results shown in Figure 6 and Table 1, we can see that although the naïve method achieves the best $\mathrm{I}_{\mathrm{perf}}$, it consumes almost all FEs. In contrast, our proposed method saves up to $\approx 80\%$ computation budget while the performance degradation is up to a order of magnitude smaller than the other three termination methods. As the trajectories of the regret of BO versus the number of FEs shown in Figure 7, we can see that the other three termination methods suffer from a premature early stopping.

Table 1: The statistical comparison results of different termination methods on $I_{\text{cost}}$ and $I_{\text{perf}}$.

| Metrics | Thresholds | Naïve method | Nguyen's method | Lorenz's method | Makarova's method | Proposed method |
|---|---|---|---|---|---|---|
| $I_{\text{cost}}$ | $\eta_1$ | 1(0)† | 0.1313(4.48E-1)‡ | 0.1244(4.48E-1)‡ | 0.7856(4.17E-1)† | 0.6206(3.17E-1) |
| | $\eta_2$ | 1(0)† | 0.1082(3.5E-2)‡ | 0.1053(1.47E-2)‡ | 0.1414(3.89E-2)‡ | 0.3012(1.94E-1) |
| | $\eta_3$ | 0.8343(2.51E-1)† | 0.1048(9.01E-3)‡ | 0.1044(4.20E-3)‡ | 0.1313(3.33E-2)‡ | 0.2209(1.12E-1) |
| $I_{\text{perf}}$ | $\eta_1$ | 0(0)‡ | 0.0077(7.28E-2)† | 0.0067(7.71E-2)† | 0(1.56E-2)† | 0(6.81E-3) |
| | $\eta_2$ | 0(0)‡ | 0.0614(1.08E-1)† | 0.0721(1.13E-1)† | 0.0167(6.51E-2)† | 0(3.35E-2) |
| | $\eta_3$ | 0(0)‡ | 0.0704(1.14E-1)† | 0.0978(1.18E-1)† | 0.0355(8.08E-2)† | 0.0028(4.17E-2) |

† denotes the performance of our proposed method is significantly better than the other peers according to the Wilcoxon's rank sum test at a 0.05 significance level; ‡ denotes the opposite case.

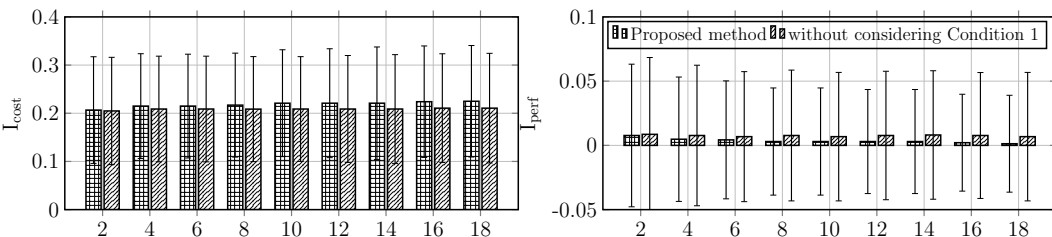

Figure 8: Bar charts with error bars of $I_{\text{cost}}$ and $I_{\text{perf}}$ when using $\tau \in \{2i\}_{i=1}^{9}$ in our proposed termination method.

## 4.3 Parameter Sensitivity Study

In this subsection, we investigate the sensitivity of our proposed termination method with respect to the parameter $\tau$. We consider various settings of $\tau \in \{2i\}_{i=1}^{9}$ and repeat the experiments on all benchmark problems introduced in Section 3.1. The aggregated comparison results for $I_{\text{cost}}$ and $I_{\text{perf}}$ are illustrated as bar charts with error bars in Figure 8. Specifically, we present the results for $\eta_{\text{lb}} = 2.05$, while the complete results can be found in the supplementary document. The plots show that the choice of $\tau$ has minimal impact on the results, except for cases with $\tau = 2$ and $\tau = 4$. This is reasonable, as the termination method may not utilize sufficient previous information when only considering a few observed samples. Additionally, we examine the scenario where the equality constraint in Condition 1, i.e., equation (5) is relaxed. The comparison results in Figure 8 reveal similar observations regarding the settings of $\tau$. However, we also notice a slight performance degradation and more aggressive early stopping in this case. These findings demonstrate that the Condition 1 helps mitigate the risk of premature early stopping.

## 5 Other Partially Related Works

Despite the limited number of dedicated studies on termination criteria for BO, various efforts have been made to explore early stopping strategies in different contexts.

The first category primarily focuses on detecting change points in sequential processes [34], with applications spanning various fields such as financial analysis [20], bioinformatics [7], and network traffic data analysis [23], among others. However, modeling the automatic termination of BO as a change point detection (CPD) problem may present several challenges. These include: 1) the absence of suitable stopping metrics that can provide signals for CPD in the optimization process of BO; 2) the unknown and uncertain nature of signal distribution, the number of change points, and change point consistency; 3) limited data available for CPD; and 4) the necessity to further evaluate change points in order to determine an appropriate moment for terminating BO.

The second category primarily focuses on determining the statistically optimal stopping moment for generalized sequential decision-making processes [14, 13]. For instance, in the classical secretary problem, termination criteria are developed to identify the maximum of an unknown distribution with minimal cost through sequential search [12]. They typically establish relationships between the costs and rewards of decision-making using cost coefficients [8, 2, 3], unknown observation costs [15, 37, 30, 25] or discount factors [36], subsequently deriving statistically optimal stopping conditions. However, quantifying the relationship between the improvement of the fitness and the

cost of BO remains challenging. Furthermore, these criteria do not leverage the information provided by the surrogate model, which is crucial in BO.

The third category primarily aims to balance exploration and exploitation in the optimization process. Among them, heuristic methods, exemplified by simulated annealing, are widely employed to halt the local search step of optimization algorithms [17, 1, 21]. However, such methods' hyperparameters lack interpretability and must be fine-tuned according to different problem characteristics. Additionally, McLeod *et al.* propose a regret-based strategy for switching between local and global optimization. Although promising for complex functions, this approach has certain limitations, including reliance on the authors' proposed regret reduction acquisition function and the potential need for additional computational resources to approximate intractable integrals. Furthermore, Eriksson *et al.* developed a trust-region-based BO that balances exploitation and exploration. This algorithm terminates local search when the trust region size is reduced to zero. However, the termination criteria lack theoretical guarantees and are bound to the proposed trust region maintenance mechanism.

## 6   Conclusion

In this paper, we developed a simple yet theoretically grounded two-step method for automatically terminating BO. The key insight is to proactively detect the local convex region and it terminates BO whenever the termination indicator built upon the local regret therein falls below a predetermined threshold. Our proposed termination method naturally strikes a balance between the quality of solution found by BO versus its computational efficiency. The proposed termination method is supported by robust theoretical underpinnings, and we have additionally introduced an approximation method to enhance the numerical stability by solving a bilevel optimization problem. Our extensive empirical studies, conducted across a variety of benchmark problems, including synthetic functions, reinforcement learning, and hyperparameter optimization, consistently demonstrated the better performance of our proposed method compared to other state-of-the-art techniques.

Besides, experimental results also show that the termination criterion of our proposed method is robust across different problems. This property paves an additional opportunity for our proposed termination method to go beyond automatically terminate BO, but to a broader range of applications, such as early stopping to avoid overfitting in neural network traing, change point or anomaly detection in data stream, and even a new perspective to strike the balance between exploitation and exploration under a bandit setting. The primary limitation of the proposed termination criterion is that it requires a predefined termination threshold, which needs to be determined based on prior knowledge or empirical observations. Although a recommended threshold selection range is given here, finding an optimal threshold that suits a wide range of optimization problems remains a challenge.

## Author Contributions

SL implemented the theoretical derivations and experiments, as well as drafted the manuscript; KL piloted the idea and re-wrote the manuscript; WL proofread the manuscript.

## Acknowledgement

This work was supported in part by the UKRI Future Leaders Fellowship under Grant MR/S017062/1 and MR/X011135/1; NSFC under Grant 62376056 and 62076056; the Royal Society under Grant IES/R2/212077; the Kan Tong Po Fellowship (KTP/R1/231017); the EPSRC under Grant 2404317; the Amazon Research Award and Alan Turing Fellowship; and the National Natural Science Foundation of China under Grant 62273119.

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
