# Supplementary Document

## 1 Pseudo-code

The pseudo-code of plugging our method into the vanilla BO is summarised in Algorithm 1. It is clear to see that the modifications are so minor that the corresponding BO algorithmic framework is kept intact. Therefore, our method is applicable to any other variants of BO in a plug-in manner.

---

**Algorithm 1:** BO with the proposed termination criterion

---

**Input:** $N_\mathrm{I}$, $FE$, N, $\eta_{lb}$

**Data:** Sampled initial solutions $\mathcal{X} \leftarrow \{\mathbf{x}^i\}_{i=1}^{N_\mathrm{I}}$ from $\Omega$, evaluated objective function values
$\quad\quad \mathcal{Y} \leftarrow \{f(\mathbf{x}^i)\}_{i=1}^{N_\mathrm{I}}$, training data set $\mathcal{D} \leftarrow \{(\mathbf{x}^i, f(\mathbf{x}^i))\}_{i=1}^{N_\mathrm{I}}$

**Output:** $\mathcal{D}$

Sample N recently observed points in $\mathcal{D}$ and store them in a temporary archive $\tilde{\mathcal{D}}$;

**for** $t = N_\mathrm{I} + 1$ **to** $FE$ **do**
$\quad$ Train the GP model based on $\mathcal{D}$;
$\quad$ Optimize an acquisition function to obtain the candidate solution $\mathbf{x}^t$;
$\quad$ Update $\tilde{\Omega}$ based on $\tilde{\mathcal{D}}$;
$\quad$ **if** *Condition 1 is met* **then**
$\quad\quad$ **if** *Condition 2 is met* **then**
$\quad\quad\quad$ Terminate BO loop;
$\quad\quad\quad$ **return** $\mathcal{D}$;
$\quad$ Evaluate the objective function value $f(\mathbf{x}^t)$ and update $\mathcal{D} \leftarrow \mathcal{D} \cup \{(\mathbf{x}^t, f(\mathbf{x}^t))\}$;
$\quad$ Update $\tilde{\mathcal{D}}$;
**end**
**return** $\mathcal{D}$;

---

## 2 Proofs of the Theoretical Results in Section 2

In this section, we present the proofs associated with the theoretical assertions from Section 2. To ensure the material is self-contained and to aid reader comprehension, we replicate the corresponding theoretical statements here.

**Proposition 1.** *Consider $\forall \mathbf{x} \in \tilde{\Omega}$, where $-\mu(\mathbf{x})$ represents a convex function. If $\| \bigtriangledown \mu(\mathbf{x}) \|_2 \leq \lambda$, we can establish:*

$$\mu(\dot{\mathbf{x}}) - \mu(\mathbf{x}) \leq \xi, \tag{1}$$

*where $\lambda = (2m_1 \xi)^{1/2}$, $\xi$ is a positive constant, and $m_1$ denotes the strong convexity parameter of $-\mu(\mathbf{x})$ [1].*

14 *Proof.* Given that $-\mu(\mathbf{x})$ is a convex function with a strong convexity parameter $m_1$, by denoting
15 $-\mu(\mathbf{x})$ as $H(\mathbf{x})$, we apply the second-order Taylor expansion upon $H(\mathbf{x})$:

$$H(\mathbf{x}^1) = H(\mathbf{x}^2) + \nabla H(\mathbf{x}^2)^\top(\mathbf{x}^1 - \mathbf{x}^2) + \frac{1}{2}(\mathbf{x}^1 - \mathbf{x}^2)^\top \nabla^2 H(\mathbf{x}^3)(\mathbf{x}^1 - \mathbf{x}^2), \qquad (2)$$

16 where $\mathbf{x}^1, \mathbf{x}^2 \in \tilde{\Omega}$, and $\mathbf{x}^3 = \alpha\mathbf{x}^1 + (1-\alpha)\mathbf{x}^2$, $\alpha \in [0,1]$. Since $\nabla^2 H(\mathbf{x}) \succeq m_1\mathbf{I}^1$, where $\mathbf{I}$ is an
17 identity diagonal matrix, we have

$$H(\mathbf{x}^1) \geq H(\mathbf{x}^2) + \nabla H(\mathbf{x}^2)^\top(\mathbf{x}^1 - \mathbf{x}^2) + \frac{m_1}{2}\|\mathbf{x}^1 - \mathbf{x}^2\|_2^2, \qquad (3)$$

18 where $\mathbf{x}^1, \mathbf{x}^2 \in \tilde{\Omega}$. Let $\hat{\mathbf{x}}^1 = \mathbf{x}^2 - \frac{1}{m_1}\nabla H(\mathbf{x}^2) = \underset{\mathbf{x}^1 \in \tilde{\Omega}}{\operatorname{argmin}} H(\mathbf{x}^2) + \nabla H(\mathbf{x}^2)^\top(\mathbf{x}^1 - \mathbf{x}^2) + \frac{m_1}{2}\|\mathbf{x}^1 -$
19 $\mathbf{x}^2\|_2^2$, we have

$$\begin{aligned} H(\mathbf{x}^1) &\geq H(\mathbf{x}^2) + \nabla H(\mathbf{x}^2)^\top(\mathbf{x}^1 - \mathbf{x}^2) + \frac{m_1}{2}\|\mathbf{x}^1 - \mathbf{x}^2\|_2^2 \\ &\geq H(\mathbf{x}^2) + \nabla H(\mathbf{x}^2)^\top(\hat{\mathbf{x}}^1 - \mathbf{x}^2) + \frac{m_1}{2}\|\hat{\mathbf{x}}^1 - \mathbf{x}^2\|_2^2 \\ &= H(\mathbf{x}^2) - \frac{1}{2m_1}\|\nabla H(\mathbf{x}^2)\|_2^2, \end{aligned} \qquad (4)$$

20 where $\mathbf{x}^1, \mathbf{x}^2 \in \tilde{\Omega}$. Thereafter, we have

$$H(\mathbf{x}) - H(\dot{\mathbf{x}}) \leq \frac{1}{2m_1}\|\nabla H(\mathbf{x})\|_2^2, \qquad (5)$$

21 where $\dot{\mathbf{x}} = \underset{\mathbf{x} \in \tilde{\Omega}}{\operatorname{argmax}} \mu(\mathbf{x})$. By replacing $H(\mathbf{x})$ with $-\mu(\mathbf{x})$, we have

$$\mu(\dot{\mathbf{x}}) - \mu(\mathbf{x}) \leq \frac{1}{2m_1}\|\nabla \mu(\mathbf{x})\|_2^2. \qquad (6)$$

22 Given $\|\nabla \mu(\mathbf{x})\|_2 \leq \lambda = (2m_1\xi)^{1/2}$, we have

$$\mu(\dot{\mathbf{x}}) - \mu(\mathbf{x}) \leq \xi. \qquad (7)$$

23 $\square$

24 **Lemma 1.** *Assume the GP employs a stationary kernel $k(\cdot, \cdot)$. For $\forall \mathbf{x} \in \tilde{\Omega}$, the lower bound of*
25 $\sigma^2(\mathbf{x})$ *is given by:*

$$\underline{\sigma}^2(\mathbf{x}) = k(\mathbf{x}, \mathbf{x}) + c\sum_{i=1}^{|\mathcal{D}|} k^2(\mathbf{x}, \mathbf{x}^i), \qquad (8)$$

26 *where $c < 0$ is a constant and $\mathbf{x}^i \in \mathcal{D}$ for $i \in \{1, \cdots, |\mathcal{D}|\}$.*

27 *Proof.* Given an input vector $\mathbf{x} \in \tilde{\Omega}$, the variance of $f(\mathbf{x})$ is predicted as:

$$\sigma^2(\mathbf{x}) = k(\mathbf{x}, \mathbf{x}) - \mathbf{k}^{*\top}(K + \sigma_\epsilon^2 I)^{-1}\mathbf{k}^*, \qquad (9)$$

28 where $X = (\mathbf{x}^1, \cdots, \mathbf{x}^N)^\top$, $N = |\mathcal{D}|$. $\mathbf{k}^*$ is the covariance vector between $X$ and $\mathbf{x}$, and $K$ is the
29 covariance matrix of $X$. By applying the Cholesky Factorization on $K$, we rewrite equation (9) as:

$$\sigma^2(\mathbf{x}) = k(\mathbf{x}, \mathbf{x}) - (L^{-1}\mathbf{k}^*)^\top L^{-1}\mathbf{k}^*, \qquad (10)$$

30 where $LL^\top = K + \sigma_\epsilon^2 I$. Let us denote

$$L^{-1} = \begin{bmatrix} l_{11} & \cdots & 0 \\ \vdots & \ddots & \vdots \\ l_{1N} & \cdots & l_{NN} \end{bmatrix}, \qquad (11)$$

31 then we have

$$L^{-1}\mathbf{k}^* = \begin{bmatrix} l_{11}k\left(\mathbf{x}, \mathbf{x}^1\right) \\ \vdots \\ \sum_{i=1}^{N} l_{iN}k(\mathbf{x}, \mathbf{x}^i) \end{bmatrix}. \qquad (12)$$

---

$^1\succeq$ means that the eigenvalues of $\nabla^2 H(\mathbf{x})$ are greater than $m_1$.

32    By introducing equation (12) into equation (10), we have

$$\sigma^2(\mathbf{x}) = k(\mathbf{x}, \mathbf{x}) - \left( \left( l_{11} k(\mathbf{x}, \mathbf{x}^1) \right)^2 + \cdots + \left( \sum_{i=1}^{N} l_{iN} k(\mathbf{x}, \mathbf{x}^i) \right)^2 \right), \tag{13}$$

33    By applying the Cauchy Schwarz's inequalities, we have:

$$\sigma^2(\mathbf{x}) \geq k(\mathbf{x}, \mathbf{x}) - \left( l_{11}^2 k^2\left(\mathbf{x}, \mathbf{x}^1\right) + \cdots + \left( \sum_{i=1}^{N} l_{iN}^2 \right) \left( \sum_{i=1}^{N} k^2(\mathbf{x}, \mathbf{x}^i) \right) \right), \tag{14}$$

34    Let $c$ be a negative constant and $c \leq -N \max\{l_{11}^2, \cdots, \sum_{i=1}^{N} l_{iN}^2\}$, we have:

$$\underline{\sigma}^2(\mathbf{x}) = k(\mathbf{x}, \mathbf{x}) + c \sum_{i=1}^{N} k^2\left(\mathbf{x}, \mathbf{x}^i\right) \leq \sigma^2(\mathbf{x}). \tag{15}$$

35    $\qquad\qquad\qquad\qquad\qquad\qquad\qquad\qquad\qquad\qquad\qquad\qquad\qquad\qquad\qquad$ $\square$

36    **Lemma 2.** *Given Lemma 1, determining $\underset{\mathbf{x} \in \tilde{\Omega}}{\arg\max}\, \underline{\sigma}^2(\mathbf{x})$ is equivalent to solving the following bilevel*
37    *optimization problem:*

$$\begin{aligned} \underset{\mathbf{x} \in \tilde{\Omega}}{\text{minimize}} \quad & d(\mathbf{x}, \mathbf{x}^1, \mathbf{x}^2) = \|\mathbf{x} - \mathbf{x}^1\|_2^2 + \|\mathbf{x} - \mathbf{x}^2\|_2^2 \\ \text{subject to} \quad & \{\mathbf{x}^1, \mathbf{x}^2\} = \underset{\substack{\mathbf{x}^1, \mathbf{x}^2 \in \mathcal{D} \cap \tilde{\Omega} \\ \mathbf{x}^1 \neq \mathbf{x}^2, \hat{\Omega} \cap \mathcal{D} = \emptyset}}{\arg\max} \|\mathbf{x}^1 - \mathbf{x}^2\|_2^2, \end{aligned} \tag{16}$$

38    *where $\hat{\Omega} = [\hat{x}_i^{\mathrm{L}}, \hat{x}_i^{\mathrm{U}}]_{i=1}^{n} \subset \tilde{\Omega}$, $\hat{x}_i^{\mathrm{L}} = \min(x_i^1, x_i^2)$ and $\hat{x}_i^{\mathrm{U}} = \max(x_i^1, x_i^2)$. Given that the lower-level*
39    *optimization can be addressed via exhaustive search, the analytical solution of* (19) *is given by*
40    $\hat{\mathbf{x}} = (\hat{x}_1^{\mathrm{L}} + \frac{\hat{x}_1^{\mathrm{U}} - \hat{x}_1^{\mathrm{L}}}{2}, \cdots, \hat{x}_n^{\mathrm{L}} + \frac{\hat{x}_n^{\mathrm{U}} - \hat{x}_n^{\mathrm{L}}}{2})^{\top}$.

41    *Proof.* Given a stationary and isotropic kernel $k(\cdot, \cdot)$, we have

$$k^2\left(\mathbf{x}, \mathbf{x}^i\right) \propto -\|\mathbf{x} - \mathbf{x}^i\|_2^2, \tag{17}$$

42    where $\mathbf{x} \in \tilde{\Omega}$ and $\mathbf{x}^i \in \mathcal{D}$. Therefore, we can find the $\hat{\Omega}$ that contain $\underset{\mathbf{x} \in \tilde{\Omega}}{\arg\max}\, \underline{\sigma}^2(\mathbf{x})$ by solving the
43    following optimization problem:

$$\begin{aligned} \{\mathbf{x}^1, \mathbf{x}^2\} = & \underset{\mathbf{x}^1 \in \mathcal{D} \cap \tilde{\Omega}, \, \mathbf{x}^2 \in \mathcal{D} \cap \tilde{\Omega}}{\text{maximize}} \|\mathbf{x}^1 - \mathbf{x}^2\|_2^2 \\ \text{subject to} \quad & \mathbf{x}^1 \neq \mathbf{x}^2, \hat{\Omega} \cap \mathcal{D} = \emptyset. \end{aligned} \tag{18}$$

44    Since $\mathcal{D} \cap \tilde{\Omega}$ consists of limited elements, we can find the exact solution of (18) by exhaustive search.
45    Since $\hat{\Omega} \cap \mathcal{D} = \emptyset$ and $k(\cdot, \cdot)$ is isotropic, determining $\underset{\mathbf{x} \in \hat{\Omega}}{\text{minimize}}\, d(\mathbf{x}, \mathbf{x}^1, \mathbf{x}^2) = \|\mathbf{x} - \mathbf{x}^1\|_2^2 + \|\mathbf{x} - \mathbf{x}^2\|_2^2$
46    is equivalent to solving $\underset{\mathbf{x} \in \hat{\Omega}}{\text{maximize}}\, \underline{\sigma}^2(\mathbf{x})$. To sum up, we can find the exact solution of $\underset{\mathbf{x} \in \tilde{\Omega}}{\arg\max}\, \underline{\sigma}^2(\mathbf{x})$
47    by solving the following bi-level optimization problem:

$$\begin{aligned} \underset{\mathbf{x} \in \tilde{\Omega}}{\text{minimize}} \quad & d(\mathbf{x}, \mathbf{x}^1, \mathbf{x}^2) = \|\mathbf{x} - \mathbf{x}^1\|_2^2 + \|\mathbf{x} - \mathbf{x}^2\|_2^2 \\ \text{subject to} \quad & \{\mathbf{x}^1, \mathbf{x}^2\} = \underset{\substack{\mathbf{x}^1, \mathbf{x}^2 \in \mathcal{D} \cap \tilde{\Omega} \\ \mathbf{x}^1 \neq \mathbf{x}^2, \hat{\Omega} \cap \mathcal{D} = \emptyset}}{\arg\max} \|\mathbf{x}^1 - \mathbf{x}^2\|_2^2, \end{aligned} \tag{19}$$

48    where $\underset{\mathbf{x} \in \tilde{\Omega}}{\arg\min}\, d(\mathbf{x}, \mathbf{x}^1, \mathbf{x}^2)$ has an analytical solution $\hat{\mathbf{x}} = (\hat{x}_1^{\mathrm{L}} + \frac{\hat{x}_1^{\mathrm{U}} - \hat{x}_1^{\mathrm{L}}}{2}, \cdots, \hat{x}_n^{\mathrm{L}} + \frac{\hat{x}_n^{\mathrm{U}} - \hat{x}_n^{\mathrm{L}}}{2})^{\top}$. $\quad\square$

49    **Proposition 2.** *Leveraging Lemma 2, suppose $\underset{\mathbf{x} \in \tilde{\Omega}}{\text{minimize}} - \sigma^2(\mathbf{x})$ exhibits convexity in its local*
50    *optimal regions, the following inequality is satisfied when $\| \triangledown \sigma^2(\mathbf{x})\|_2 \leq \lambda$:*

$$\sigma^2(\ddot{\mathbf{x}}) - \sigma^2(\mathbf{x}) \leq \beta + \xi, \tag{20}$$

51    *where $\lambda = (2m_2\xi)^{1/2}$, $\xi > 0$, $m_2 > 0$ represents the strong convexity parameter of $-\sigma^2(\mathbf{x})$ in its*
52    *local optimal regions [1], and $\beta$ is constrained by $0 \leq \beta \leq \sigma^2(\ddot{\mathbf{x}}) - \sigma^2(\hat{\mathbf{x}})$.*

53    *Proof.* By applying $\hat{\mathbf{x}}$ as the initialization point for $\underset{\mathbf{x}\in\Omega}{\text{maximize}}\ \sigma^2(\mathbf{x})$, we have:

$$\beta = \sigma^2(\ddot{\mathbf{x}}) - \sigma^2(\hat{\mathbf{x}}) \geq \sigma^2(\ddot{\mathbf{x}}) - \sigma^2(\bar{\mathbf{x}}), \tag{21}$$

54    where $\bar{\mathbf{x}}, \hat{\mathbf{x}} \in \bar{\Omega} \subseteq \tilde{\Omega}$, $\bar{\mathbf{x}}$ is a local optimal solution and $\bar{\Omega}$ is a local optimal region of $\underset{\mathbf{x}\in\Omega}{\text{maximize}}\ \sigma^2(\mathbf{x})$.

55    Since $\underset{\mathbf{x}\in\bar{\Omega}}{\text{minimize}} - \sigma^2(\mathbf{x})$ exhibits convexity in its local optimal regions, given Proposition 1 and

56    $\|\bigtriangledown \sigma^2(\mathbf{x})\|_2 \leq \lambda$, we have

$$\sigma^2(\bar{\mathbf{x}}) - \sigma^2(\mathbf{x}) \leq \xi, \tag{22}$$

57    where $\mathbf{x} \in \bar{\Omega}$. By introducing equation (22) in to equation (21), we have

$$\sigma^2(\ddot{\mathbf{x}}) - \sigma^2(\mathbf{x}) \leq \beta + \xi. \tag{23}$$

58    □

59    **Lemma 3.** *As per Srinivas* et al.*, the optimization process in BO can be conceptualized as a sampling*
60    *process from a GP. Hence, for any* $\mathbf{x} \in \Omega$*, we have:*

$$\Pr\left(\,|f(\mathbf{x}) - \mu(\mathbf{x})| \leq \omega\sigma(\mathbf{x})\right) > \delta, \tag{24}$$

61    *where* $\delta > 0$ *signifies the confidence level adhered to by the UCB.*

62    *Proof.* This lemma is directly from Srinivas *et al.*. The proof can be found therein.    □

63    **Corollary 1.** *Based on Lemma 3 and Condition 2, we deduce that:*

$$\Pr\left(f^{\mathrm{acq}}(\tilde{\mathbf{x}}^\star) + \varepsilon \geq f(\mathbf{x}^\star)\right) > \delta, \tag{25}$$

64    *where* $\tilde{\mathbf{x}}^\star = \underset{\mathbf{x}\in\Omega}{\text{argmax}}\ f^{\mathrm{acq}}(\mathbf{x})$*, and* $\mathbf{x}^\star$ *represents the true global optimum. Furthermore,*

$$0 \leq \varepsilon \leq \mu(\dot{\mathbf{x}}) + \omega\sigma(\ddot{\mathbf{x}}) - f^{\mathrm{acq}}(\tilde{\mathbf{x}}^\star), \tag{26}$$

65    *where* $\dot{\mathbf{x}}$*,* $\ddot{\mathbf{x}}$*, and* $\tilde{\mathbf{x}}^\star$ *are elements of* $\tilde{\Omega}$*, while* $\delta > 0$ *denotes the confidence level of the UCB.*

66    *Proof.* Given Lemma 3, we have $\Pr\left(\mu(\mathbf{x}) + \omega\sigma(\mathbf{x}) \geq f(\mathbf{x})\right) > \delta$. Then, let $f^{\mathrm{acq}}(\mathbf{x}) = \mu(\mathbf{x}) + \omega\sigma(\mathbf{x})$
67    and $\tilde{\mathbf{x}}^\star = \underset{\mathbf{x}\in\Omega}{\text{argmax}}\ f^{\mathrm{acq}}(\mathbf{x})$, we have

$$\Pr(f^{\mathrm{acq}}(\tilde{\mathbf{x}}^\star) \geq f^{\mathrm{acq}}(\mathbf{x}^\star) \geq f(\mathbf{x}^\star)) > \delta, \tag{27}$$

68    where $\mathbf{x}^\star = \text{argmax}\ f(\mathbf{x})$. In practice, there exist numeric errors between $\underset{\mathbf{x}\in\Omega}{\sup}\ f^{\mathrm{acq}}(\mathbf{x})$ and $f^{\mathrm{acq}}(\tilde{\mathbf{x}}^\star)$.

69    By denoting the upper bound of the errors as $\varepsilon$, we have

$$\Pr\left(f^{\mathrm{acq}}(\tilde{\mathbf{x}}^\star) + \varepsilon \geq f(\mathbf{x}^\star)\right) > \delta. \tag{28}$$

70    Since $\dot{\mathbf{x}} = \underset{\mathbf{x}\in\tilde{\Omega}}{\text{argmax}}\ \mu(\mathbf{x})$ and $\ddot{\mathbf{x}} = \underset{\mathbf{x}\in\bar{\Omega}}{\text{argmax}}\ \sigma^2(\mathbf{x})$, $f^{\mathrm{acq}}$ has the following upper bound:

$$\underset{\mathbf{x}\in\tilde{\Omega}}{\sup}\ f^{\mathrm{acq}}(\mathbf{x}) \leq \mu(\dot{\mathbf{x}}) + \sigma(\ddot{\mathbf{x}}). \tag{29}$$

71    Therefore, the $\varepsilon$ is bounded by

$$0 \leq \varepsilon \leq \mu(\dot{\mathbf{x}}) + \omega\sigma(\ddot{\mathbf{x}}) - f^{acq}(\tilde{\mathbf{x}}^\star), \tag{30}$$

72    where $\dot{\mathbf{x}}, \ddot{\mathbf{x}}, \tilde{\mathbf{x}}^\star \in \tilde{\Omega}$.    □

73    **Theorem 1.** *Leveraging Corollary 1, when employing the termination method proposed in this paper,*
74    *we deduce that the global regret bound of BO as:*

$$\Pr\left(r \leq 2\omega\sigma(\tilde{\mathbf{x}}^\star) + \varepsilon\right) > \delta, \tag{31}$$

75    *where* $\delta > 0$ *signifies the confidence level associated with the UCB.*

*Proof.* We formulate the regret of BO as follows:
$$r = f(\mathbf{x}^\star) - \mu(\mathbf{x}) + \mu(\mathbf{x}) - f(\tilde{\mathbf{x}}^\star). \tag{32}$$
Based on Lemma 3 and Corollary 1, by introducing equation (24) and equation (25) into equation (32), we have
$$\Pr\left(r \le 2\omega\sigma(\tilde{\mathbf{x}}^\star) + \varepsilon\right) > \delta. \tag{33}$$
$\square$

**Theorem 2.** *Building upon Condition 1 and Condition 2, and employing the termination method proposed in this paper, we establish the local regret bound of BO as:*
$$\Pr\left(f(\mathbf{x}^\star) - f(\mathbf{x}) \le \tilde{r}\right) > \delta, \tag{34}$$
*where $\mathbf{x} \in \tilde{\Omega}$, $\mathbf{x}^\star$ denotes the true global optimum in $\tilde{\Omega}$, and $\delta > 0$ is the confidence level of the UCB.*

*Proof.* Based on Theorem 1, by introducing equation (30) into equation (33), we have
$$\Pr\left(f(\mathbf{x}^\star) - f(\mathbf{x}) \le \tilde{r}\right) > \delta, \tag{35}$$
where $\tilde{r} = \mu(\dot{\mathbf{x}}) - \mu(\tilde{\mathbf{x}}) + \omega\left(\sigma(\ddot{\mathbf{x}}) + \sigma(\tilde{\mathbf{x}})\right)$. $\square$

# 3 Experimental Results

## 3.1 Compelmentary Results

In this subsection, we provide the complete experimental results, offering supplementary findings to those presented in the main manuscript. More specifically,

- Figures Figure 1 to Figure 16 presents the complete results of the trajectories of $I_{\mathrm{cdf}}$ collected on all comparisons.
- Figures 17 to 28 show the complete results of the trajectories of different termination indicators versus the number of FEs during the BO process collected on all comparisons.
- Figures 29 to 47 provide the complete results of the trajectories of the regret of BO versus the number of FEs during the BO process collected on all comparisons.

To ensure the material is self-contained and to aid reader comprehension, we replicate the corresponding statements of the three types of benchmark problems here.

- Synthetic functions: We consider Ackley, Levy, and Schwefel functions [6] with $n \in \{2, 5, 10\}$. The objective function $f(\mathbf{x})$ is contaminated by Gaussian noise $\zeta \sim \mathcal{N}(0.0, 0.2)$. The maximal number of FEs is set to $N_{\mathrm{FE}} = 50n$, with $5n$ allocated to initialization.
- Reinforcement learning (RL): We examine two RL tasks chosen from OpenAI Gym [2]: Lunar Lander with $n = 12$ and Swimmer with $n = 16$. We set $N_{\mathrm{FE}} = 50n$, with $5n$ FEs allocated to initialization.
- Hyperparameter optimization (HPO): We consider 5 HPO tasks (task ID: $\{53, 10101, 167149, 167162, 167170\}$) picked up from the HPOBench [3] for tuning support vector machine (SVM) with $n = 2$, multi-layer perceptron (MLP) with $n = 5$, random forest with $n = 4$ and XGBoost with $n = 8$. The computational budget is set the same as in the RL tasks.

## 3.2 Ablation Experiments of the Initialization Strategies for L-BFGS

As discussed in Remark 2 of Section 2.2 in the main manuscript, we suggest initializing L-BFGS at the point of $\underset{\mathbf{x} \in \tilde{\Omega}}{\mathrm{argmax}}\, \underline{\sigma}^2(\mathbf{x})$, where $\underline{\sigma}^2(\mathbf{x})$ denotes the lower bound of $\sigma^2(\mathbf{x})$ over $\tilde{\Omega}$, to ensure the numerical stability of $\ddot{\mathbf{x}}$. To validate the effectiveness of this initialization strategy, we compare it with regard to a variant (dubbed `Random`) that samples a randomly generated solution from $\tilde{\Omega}$ as the starting point. From the statistical comparison results shown in Table 1, we can see that our proposed initialization strategy outperforms `Random`. More specifically, it achieves statistically significant better median regrets in more than half of the comparisons. Note that even `Random` can obtain better regret than ours, many of them do not have statistical significance.

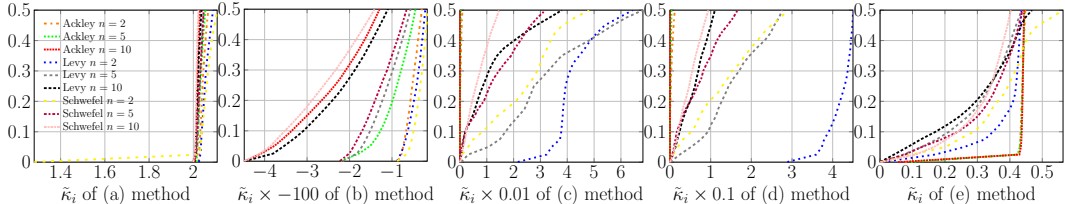

Figure 1: Trajectories of $I_{cdf}$ obtained by applying UCB to solve synthetic functions. Different subplots are (a) our proposed method, (b) Naïve method, (c) Nguyen's method, (d) Lorenz's method, and (e) Makarova's method, respectively.

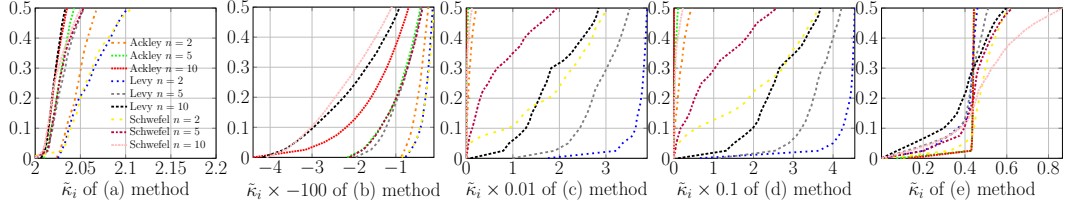

Figure 2: Trajectories of $I_{cdf}$ obtained by applying EI to solve synthetic functions. Different subplots are (a) our proposed method, (b) Naïve method, (c) Nguyen's method, (d) Lorenz's method, and (e) Makarova's method, respectively.

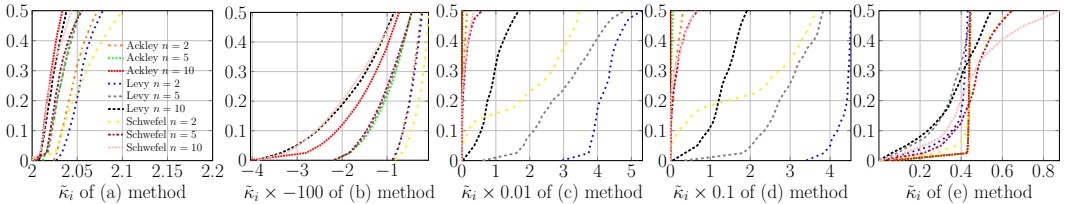

Figure 3: Trajectories of $I_{cdf}$ obtained by applying PI to solve synthetic functions. Different subplots are (a) our proposed method, (b) Naïve method, (c) Nguyen's method, (d) Lorenz's method, and (e) Makarova's method, respectively.

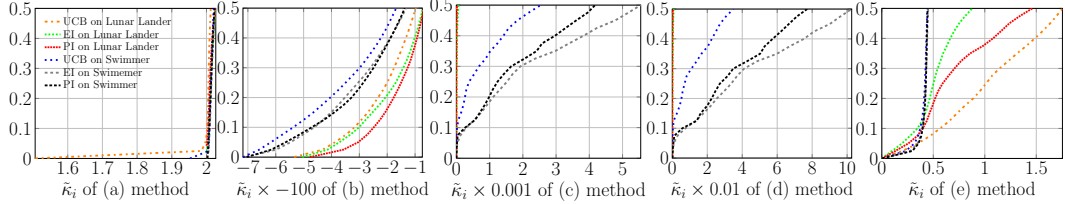

Figure 4: Trajectories of $I_{cdf}$ obtained by applying UCB, EI, PI to solve RL problems. Different subplots are (a) our proposed method, (b) Naïve method, (c) Nguyen's method, (d) Lorenz's method, and (e) Makarova's method, respectively.

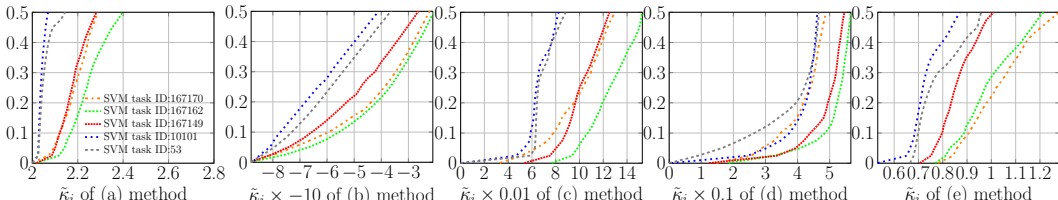

Figure 5: Trajectories of $I_{cdf}$ obtained by applying UCB to tune SVM on different tasks. Different subplots are (a) our proposed method, (b) Naïve method, (c) Nguyen's method, (d) Lorenz's method, and (e) Makarova's method, respectively.

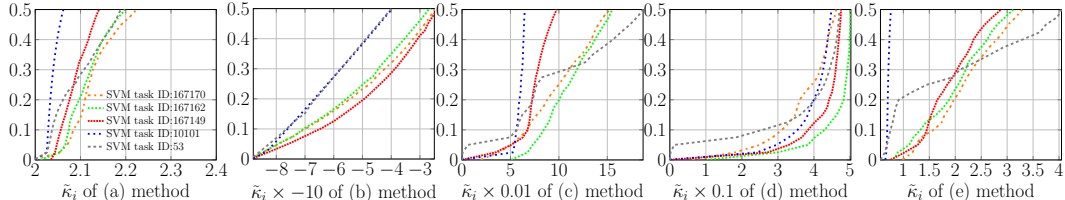

Figure 6: Trajectories of $I_{cdf}$ obtained by applying EI to tune SVM on different tasks. Different subplots are (a) our proposed method, (b) Naïve method, (c) Nguyen's method, (d) Lorenz's method, and (e) Makarova's method, respectively.

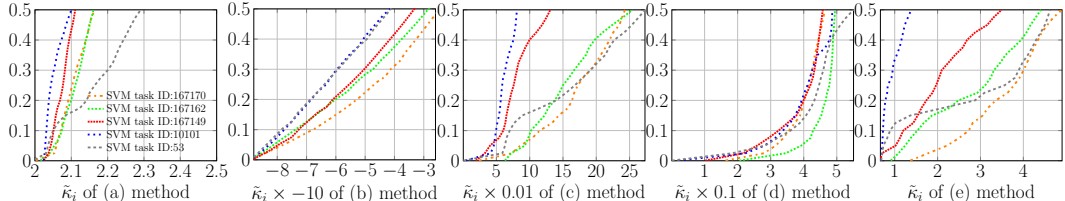

Figure 7: Trajectories of $I_{cdf}$ obtained by applying PI to tune SVM on different tasks. Different subplots are (a) our proposed method, (b) Naïve method, (c) Nguyen's method, (d) Lorenz's method, and (e) Makarova's method, respectively.

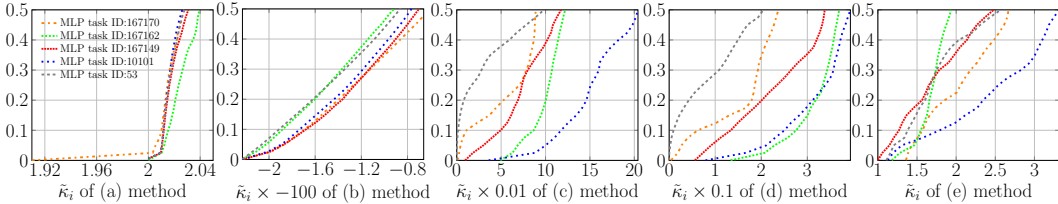

Figure 8: Trajectories of $I_{cdf}$ obtained by applying UCB to tune MLP on different tasks. Different subplots are (a) our proposed method, (b) Naïve method, (c) Nguyen's method, (d) Lorenz's method, and (e) Makarova's method, respectively.

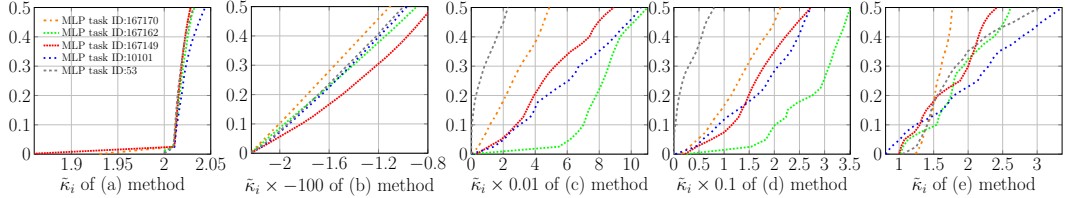

Figure 9: Trajectories of $I_{cdf}$ obtained by applying EI to tune MLP on different tasks. Different subplots are (a) our proposed method, (b) Naïve method, (c) Nguyen's method, (d) Lorenz's method, and (e) Makarova's method, respectively.

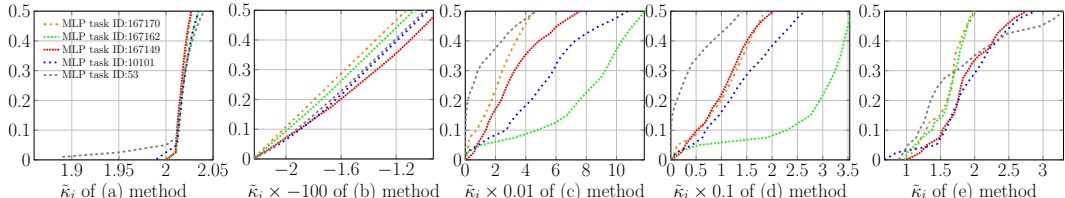

Figure 10: Trajectories of $I_{cdf}$ obtained by applying PI to tune MLP on different tasks. Different subplots are (a) our proposed method, (b) Naïve method, (c) Nguyen's method, (d) Lorenz's method, and (e) Makarova's method, respectively.

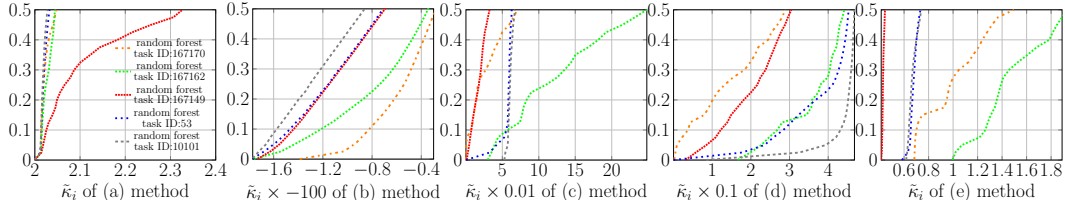

Figure 11: Trajectories of $I_{cdf}$ obtained by applying UCB to tune random forest on different tasks. Different subplots are (a) our proposed method, (b) Naïve method, (c) Nguyen's method, (d) Lorenz's method, and (e) Makarova's method, respectively.

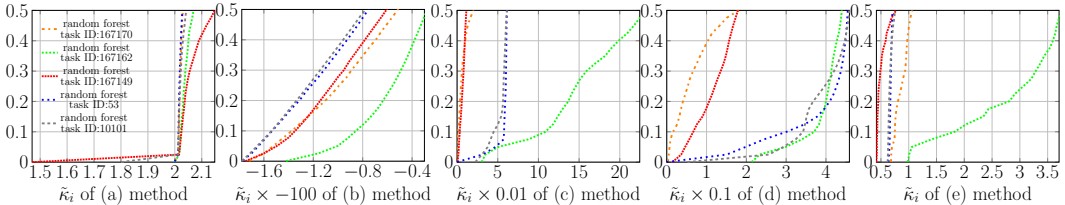

Figure 12: Trajectories of $I_{cdf}$ obtained by applying EI to tune random forest on different tasks. Different subplots are (a) our proposed method, (b) Naïve method, (c) Nguyen's method, (d) Lorenz's method, and (e) Makarova's method, respectively.

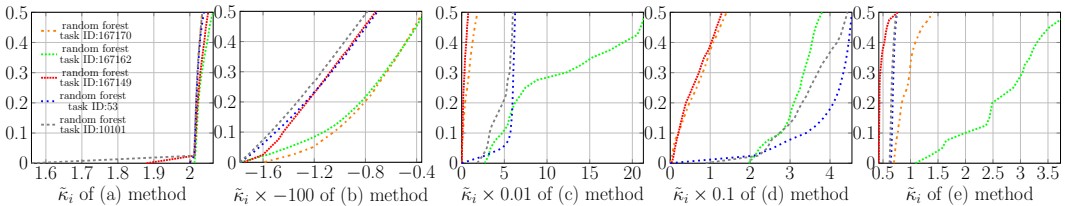

Figure 13: Trajectories of $I_{cdf}$ obtained by applying PI to tune random forest on different tasks. Different subplots are (a) our proposed method, (b) Naïve method, (c) Nguyen's method, (d) Lorenz's method, and (e) Makarova's method, respectively.

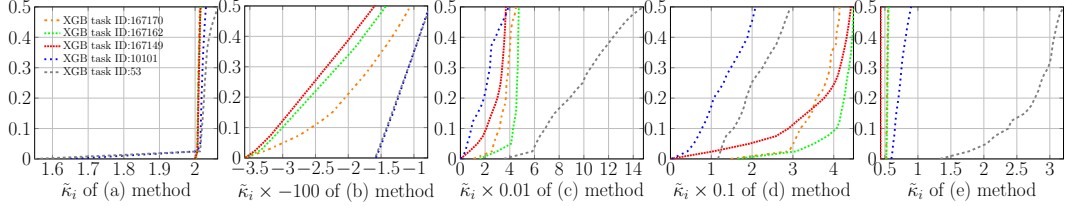

Figure 14: Trajectories of $I_{cdf}$ obtained by applying UCB to tune XGBoost on different tasks. Different subplots are (a) our proposed method, (b) Naïve method, (c) Nguyen's method, (d) Lorenz's method, and (e) Makarova's method, respectively.

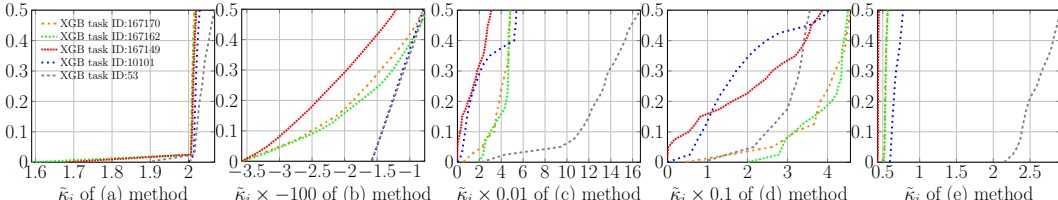

Figure 15: Trajectories of $I_{cdf}$ obtained by applying EI to tune XGBoost on different tasks. Different subplots are (a) our proposed method, (b) Naïve method, (c) Nguyen's method, (d) Lorenz's method, and (e) Makarova's method, respectively.

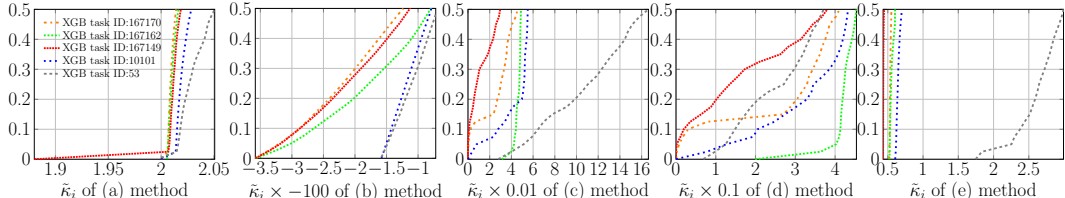

Figure 16: Trajectories of $I_{cdf}$ obtained by applying PI to tune XGBoost on different tasks. Different subplots are (a) our proposed method, (b) Naïve method, (c) Nguyen's method, (d) Lorenz's method, and (e) Makarova's method, respectively.

Table 1: The statistical comparison results of different initialization strategies in L-BFGS.

|  | Ackley ($n = 2$) | Ackley ($n = 5$) | Ackley ($n = 10$) | Levy ($n = 2$) |
|---|---|---|---|---|
| Our | $0.6151(1.59\text{E-1})^\star$ | $0.7458(2.19\text{E-1})^\star$ | $0.6599(1.82\text{E-1})^\dagger$ | $0.322(1.60\text{E-1})^\star$ |
| Random | $0.7348(1.30\text{E}{-}1)$ | $0.846(1.89\text{E}{-}1)$ | $0.6406(1.92\text{E}{-}1)$ | $0.337(1.82\text{E}{-}1)$ |
|  | Levy ($n = 5$) | Levy ($n = 10$) | Schwefel ($n = 2$) | Schwefel ($n = 5$) |
| Our | $0.7557(1.51\text{E}{-}1)^\star$ | $0.7655(1.20\,E{-}1)^\dagger$ | $0.8311(9.36\,E-2)^\dagger$ | $0.8179(9.54\text{E}{-}2)^\dagger$ |
| Random | $0.6966(1.61\text{E}{-}1)$ | $0.679(1.01\text{E}{-}1)$ | $0.7683(1.83\text{E}{-}1)$ | $0.6804(2.92\text{E}{-}1)$ |
|  | Schwefel ($n = 10$) | Lunar lander | Swimmer | SVM (Task #5) |
| Our | $0.8515(7.81\text{E}{-}2)^\dagger$ | $1.1246(2.68\text{E}{-}2)^\dagger$ | $1.1071(2.02\text{E}{-}2)^\dagger$ | $0.6713(2.25\text{E}{-}1)^\star$ |
| Random | $0.478(2.35\text{E}{-}1)$ | $1.1166(3.10\text{E}{-}2)$ | $1.1009(1.65\text{E}{-}2)$ | $0.5905(2.34\text{E}{-}1)$ |
|  | SVM (Task #2) | SVM (Task #3) | SVM (Task #4) | SVM (Task #5) |
| Our | $0.6574(1.84\text{E}{-}1)^\star$ | $0.4855(1.43\text{E}{-}1)^\dagger$ | $0.7607(1.98\text{E}{-}1)^\star$ | $0.6349(2.01\text{E}{-}1)^\dagger$ |
| Random | $0.5228(2.14\text{E}{-}1)$ | $0.3722(1.95\text{E}{-}1)$ | $0.5971(2.38\text{E}{-}1)$ | $0.4865(2.01\text{E}{-}1)$ |
|  | MLP (Task #1) | MLP (Task #2) | MLP (Task #3) | MLP (Task #4) |
| Our | $0.9063(1.56\text{E}{-}1)^\dagger$ | $0.8762(1.20\text{E}{-}1)^\dagger$ | $0.9029(1.54\text{E}{-}1)^\dagger$ | $0.8767(6.47\text{E}{-}2)^\star$ |
| Random | $0.8359(1.77\text{E}{-}1)$ | $0.7995(1.56\text{E}{-}1)$ | $0.8279(3.06\text{E}{-}1)$ | $0.8108(1.13\text{E}{-}1)$ |
|  | MLP (Task #5) | Random forest (Task #1) | Random forest (Task #2) | Random forest (Task #3) |
| Our | $0.8879(1.50\text{E}{-}1)^\dagger$ | $0.3479(1.39\text{E}{-}1)^\star$ | $0.3493(1.76\text{E}{-}1)^\star$ | $0.3377(1.79\text{E}{-}1)^\dagger$ |
| Random | $0.817(2.47\text{E}{-}1)$ | $0.2927(1.99\text{E}{-}1)$ | $0.3493(1.65\text{E}{-}1)$ | $0.3414(1.20\text{E}{-}1)$ |
|  | Random forest (Task #4) | Random forest (Task #5) | XGBoost (Task #1) | XGBoost (Task #2) |
| Our | $0.3242(1.26\text{E}{-}1)^\dagger$ | $0.3841(9.13\text{E}{-}2)^\star$ | $0.8762(2.19\text{E}{-}1)^\star$ | $0.8601(2.01\text{E}{-}1)^\dagger$ |
| Random | $0.2903(9.90\text{E}{-}2)$ | $0.3841(1.56\text{E}{-}1)$ | $0.8417(2.22\text{E}{-}1)$ | $0.783(3.10\text{E}{-}1)$ |
|  | XGBoost (Task #3) | XGBoost (Task #4) | XGBoost (Task #5) |  |
| Our | $0.6982(2.36\text{E}{-}1)^\star$ | $0.7185(2.55\,\text{E}{-}1)^\star$ | $0.8375(1.58\text{E}{-}1)^\star$ |  |
| Random | $0.5493(2.27\text{E}{-}1)$ | $0.583(2.69\text{E}{-}1)$ | $0.6956(1.86\text{E}{-}1)$ |  |

$^\dagger$ denotes the performance of our proposed initialization strategy is significantly better than the random initialization strategy according to the Wilcoxon's rank sum test at a 0.05 significance level. Whereas $^\star$ denotes the better result does not have any statistical significance. The Task #1 to the Task #5 corresponds to the task ID: {53, 10101, 167149, 167162, 167170} of HPOBench [3], respectively.

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

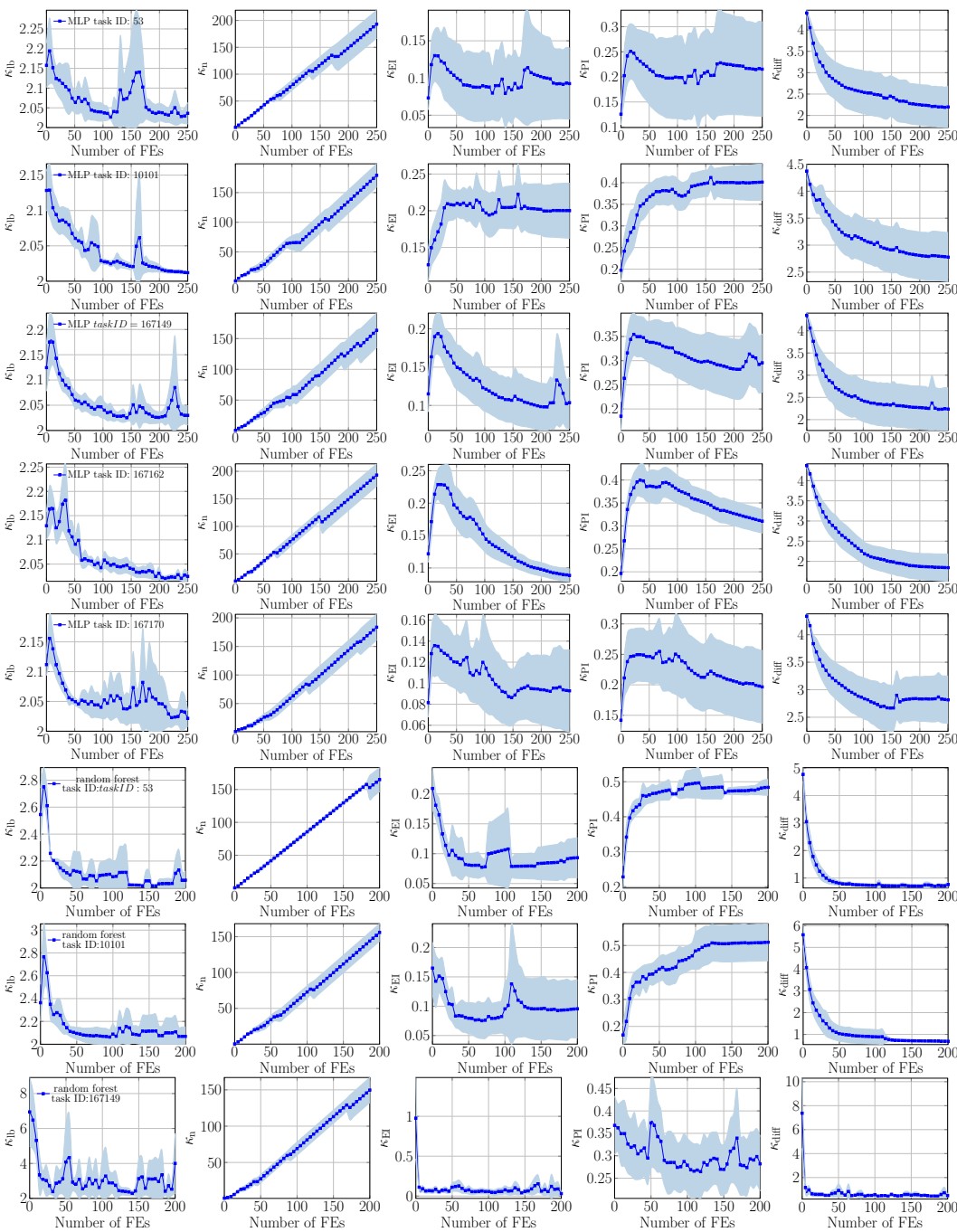

Figure 19: Trajectories of different termination indicators versus the number of FEs during the BO process of applying UCB on MLP (task ID: {53, 10101, 167149, 167162, 167170}) and random forest (task ID: {53, 10101, 167149}).

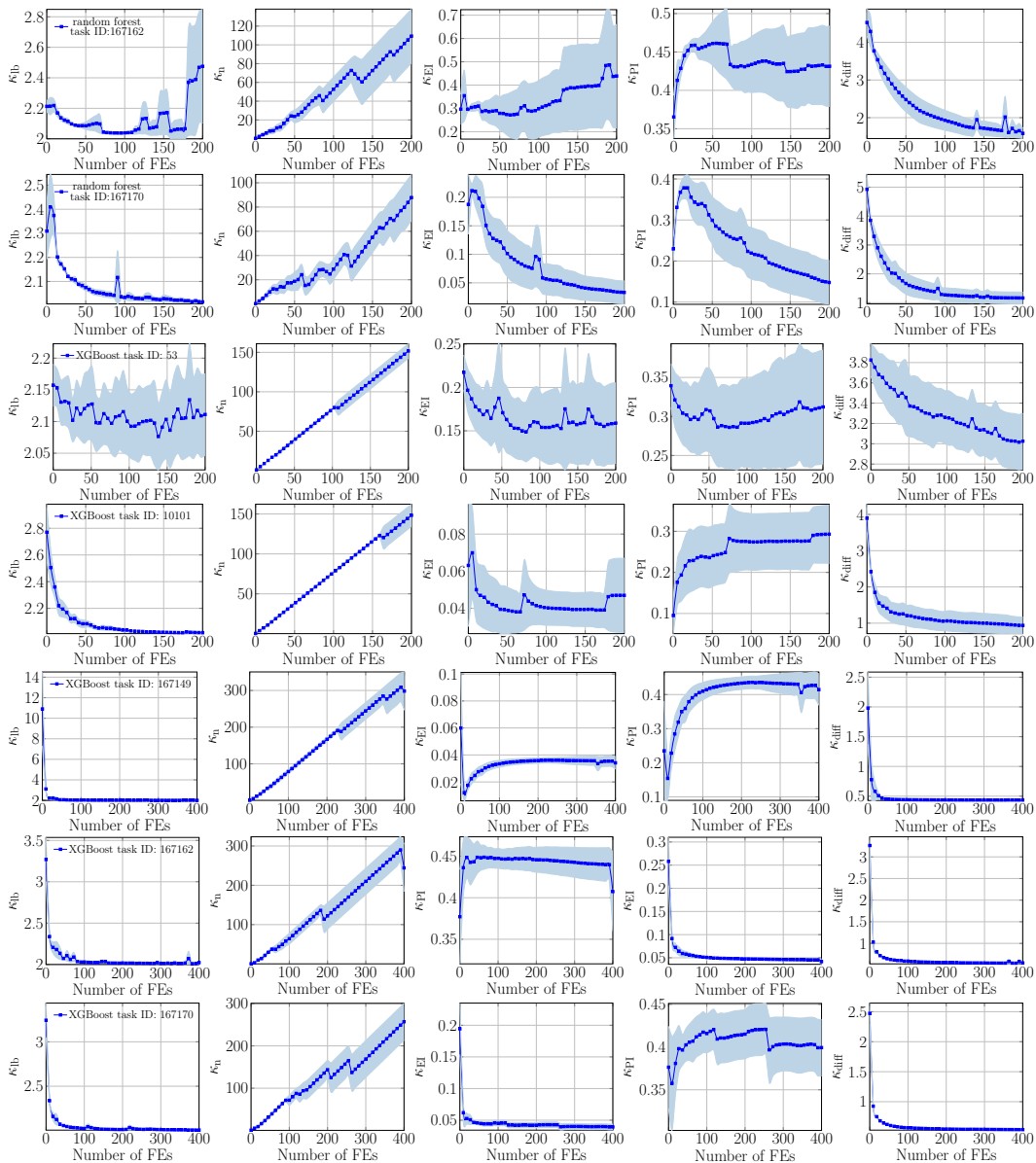

Figure 20: Trajectories of different termination indicators versus the number of FEs during the BO process of applying UCB on random forest (task ID: $\{167162, 167170\}$) and XGBoost (task ID: $\{53, 10101, 167149, 167162, 167170\}$).

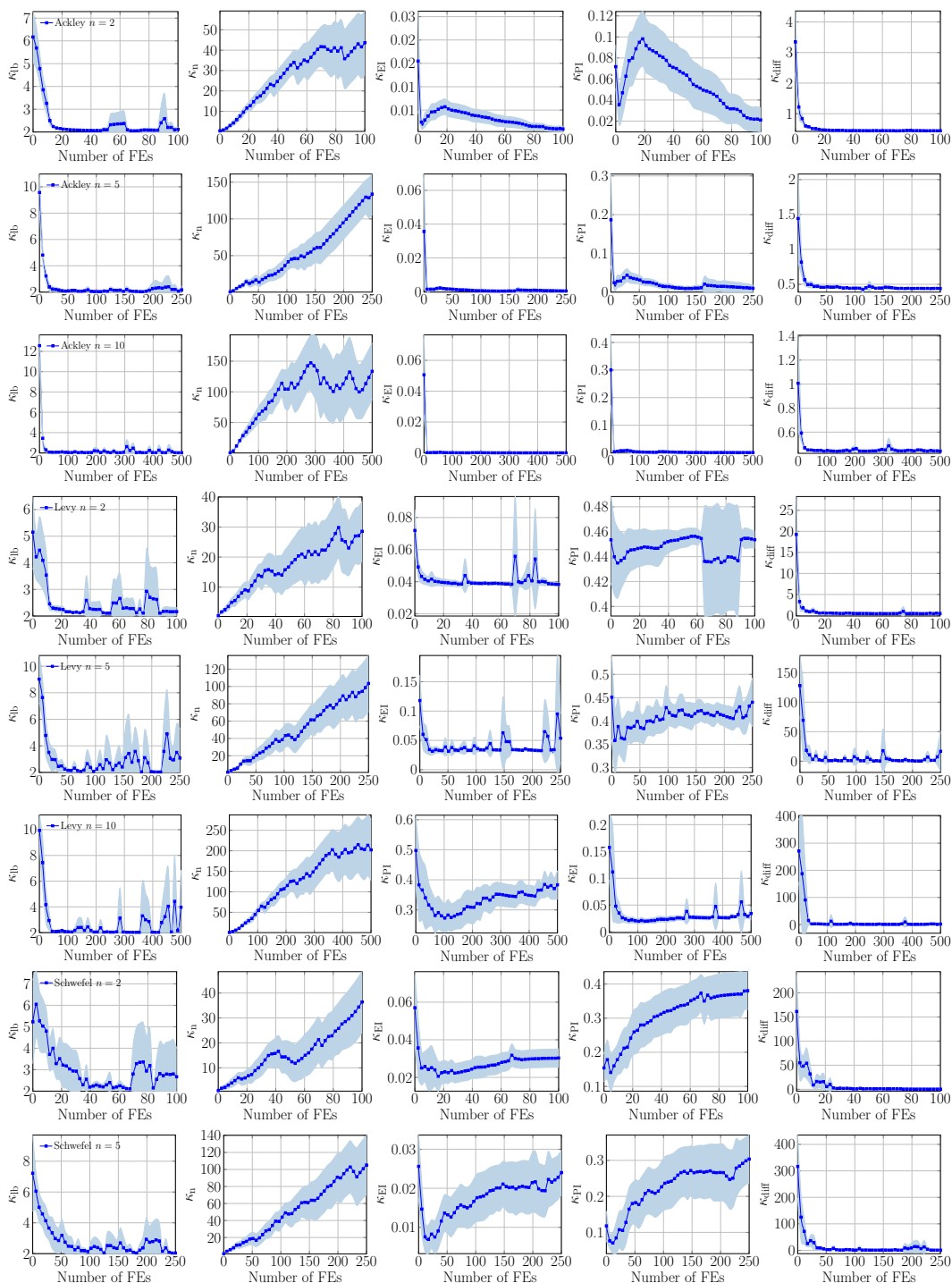

Figure 21: Trajectories of different termination indicators versus the number of FEs during the BO process of applying EI on Ackley ($n \in \{2, 5, 10\}$), Levy ($n \in \{2, 5, 10\}$) and Schwefel ($n \in \{2, 5\}$).

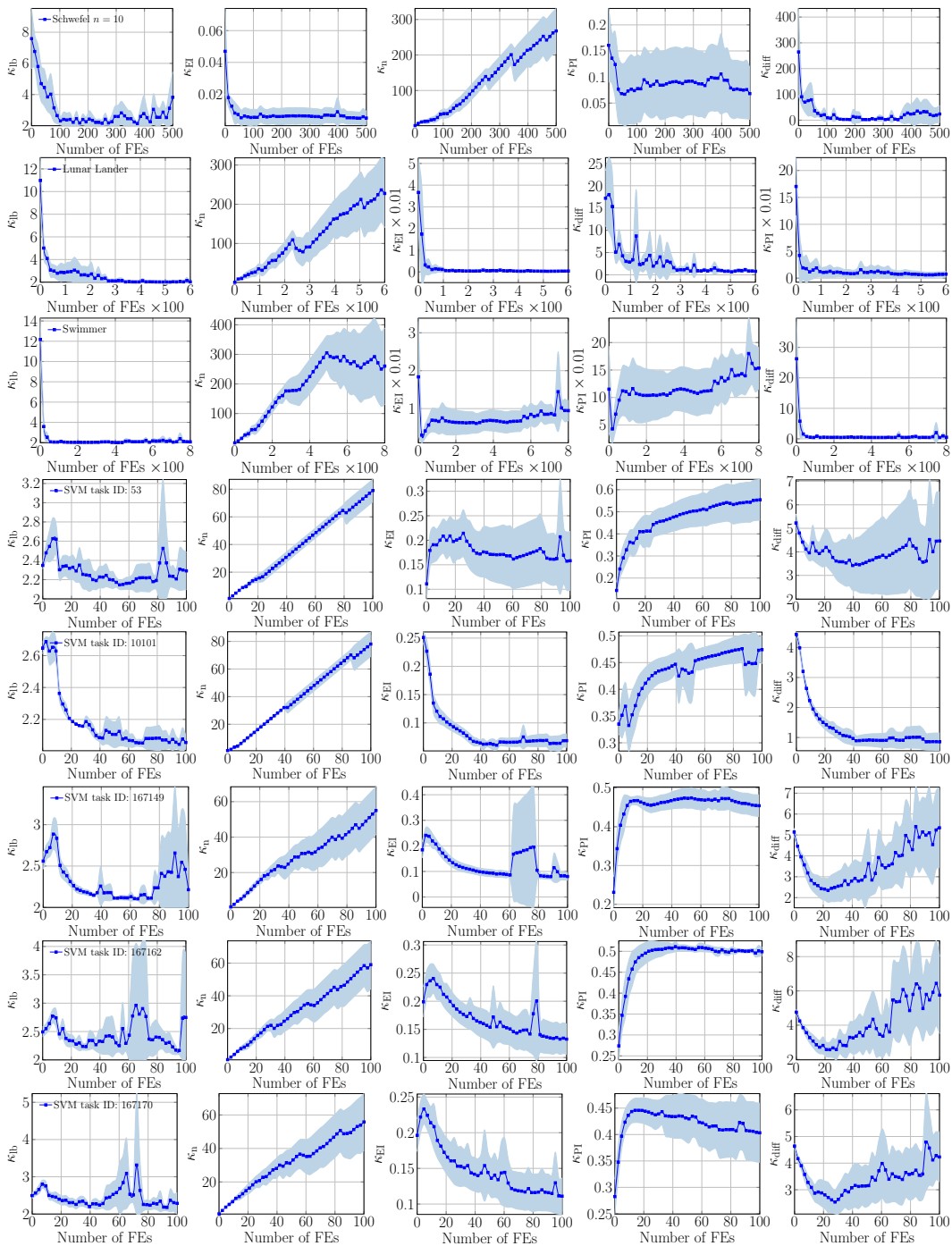

Figure 22: Trajectories of different termination indicators versus the number of FEs during the BO process of applying EI on Schwefel ($n = 10$), Lunar Lander, Swimmer and SVM (task ID: $\{53, 10101, 167149, 167162, 167170\}$).

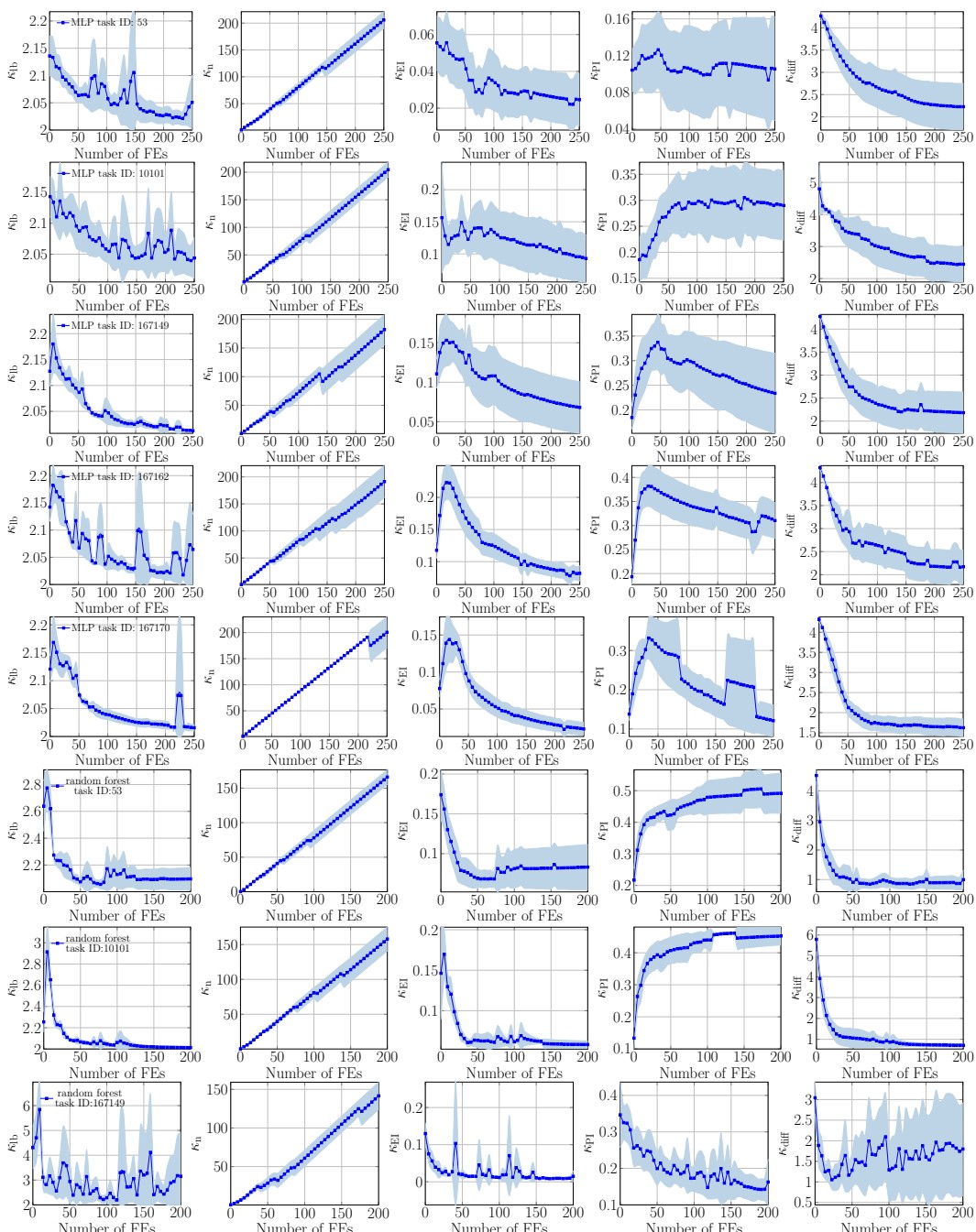

Figure 23: Trajectories of different termination indicators versus the number of FEs during the BO process of applying EI on MLP (task ID: $\{53, 10101, 167149, 167162, 167170\}$) and random forest (task ID: $\{53, 10101, 167149\}$).

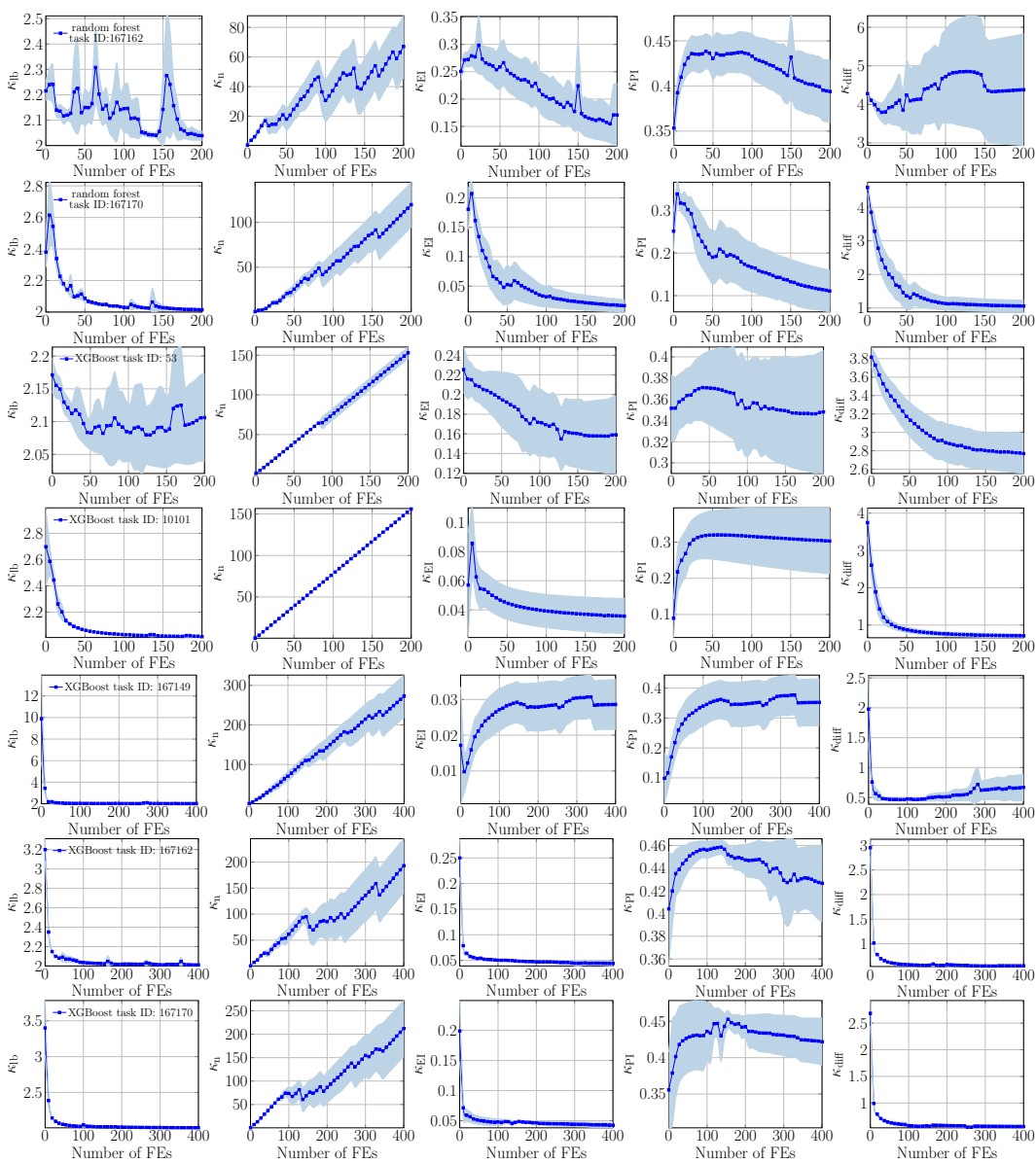

Figure 24: Trajectories of different termination indicators versus the number of FEs during the BO process of applying EI on random forest (task ID: $\{167162, 167170\}$) and XGBoost (task ID: $\{53, 10101, 167149, 167162, 167170\}$).

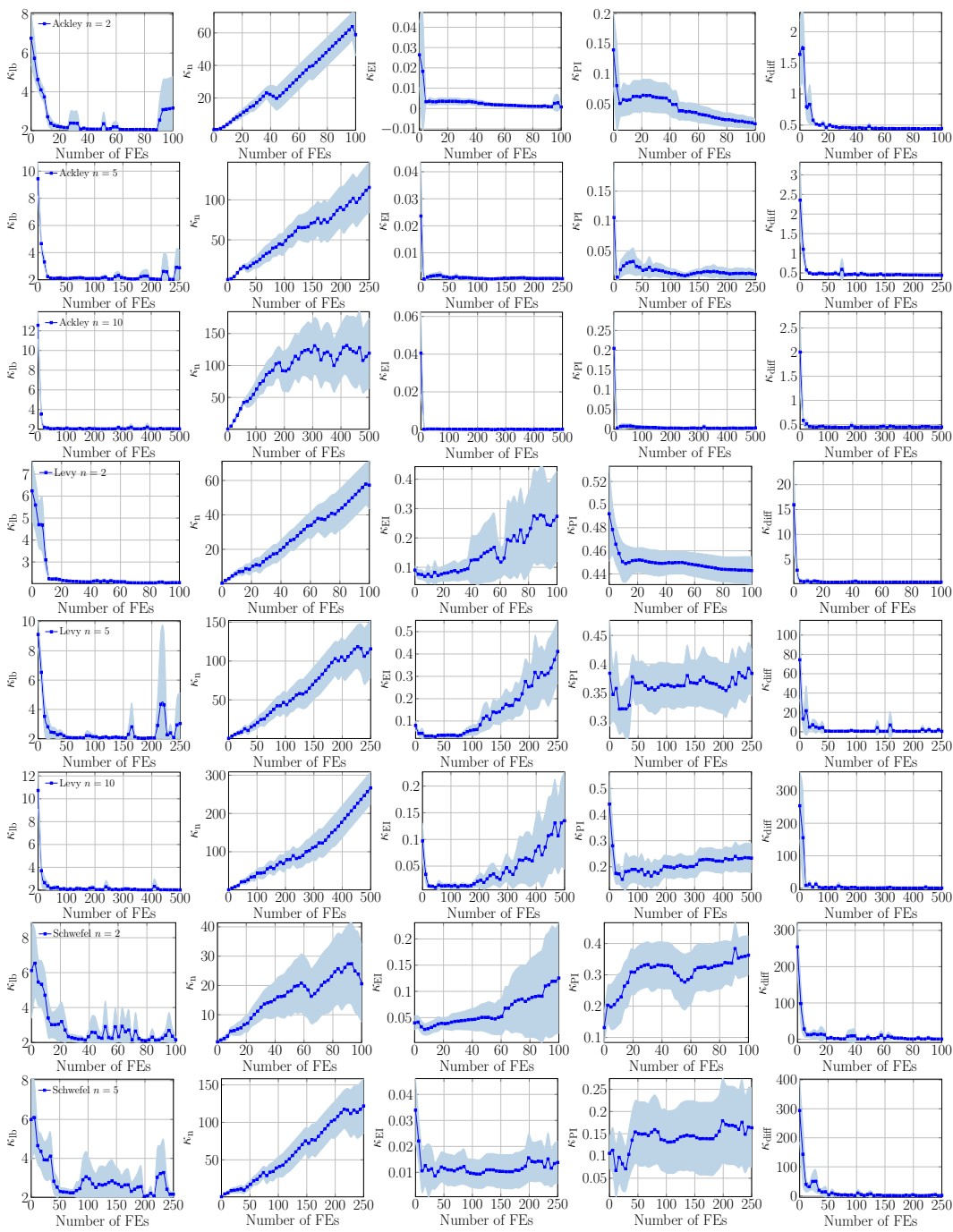

Figure 25: Trajectories of different termination indicators versus the number of FEs during the BO process of applying PI on Ackley ($n \in \{2, 5, 10\}$), Levy ($n \in \{2, 5, 10\}$) and Schwefel ($n \in \{2, 5\}$).

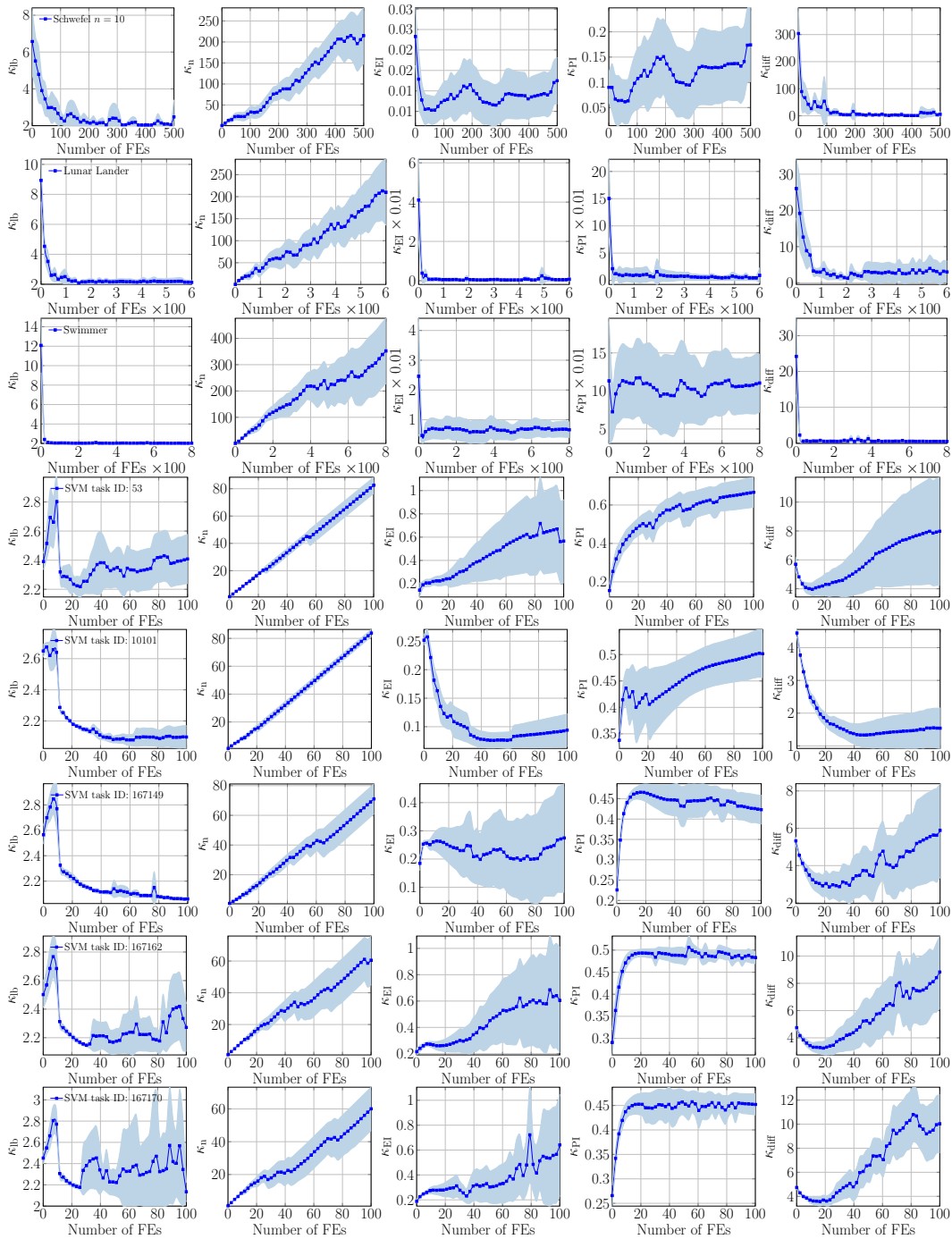

Figure 26: Trajectories of different termination indicators versus the number of FEs during the BO process of applying PI on Schwefel ($n = 10$), Lunar Lander, Swimmer and SVM (task ID: $\{53, 10101, 167149, 167162, 167170\}$).

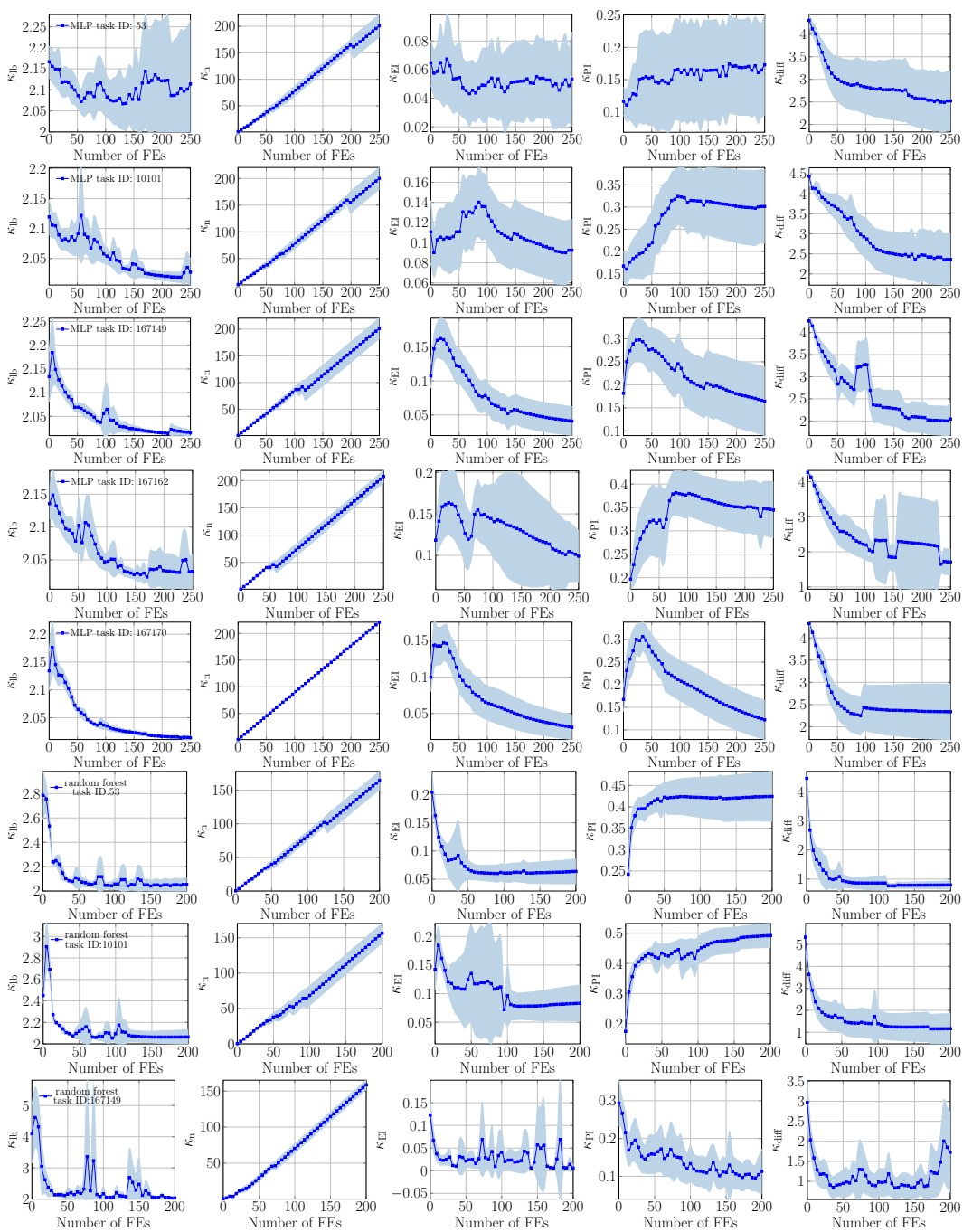

Figure 27: Trajectories of different termination indicators versus the number of FEs during the BO process of applying PI on MLP (task ID: $\{53, 10101, 167149, 167162, 167170\}$) and random forest (task ID: $\{53, 10101, 167149\}$).

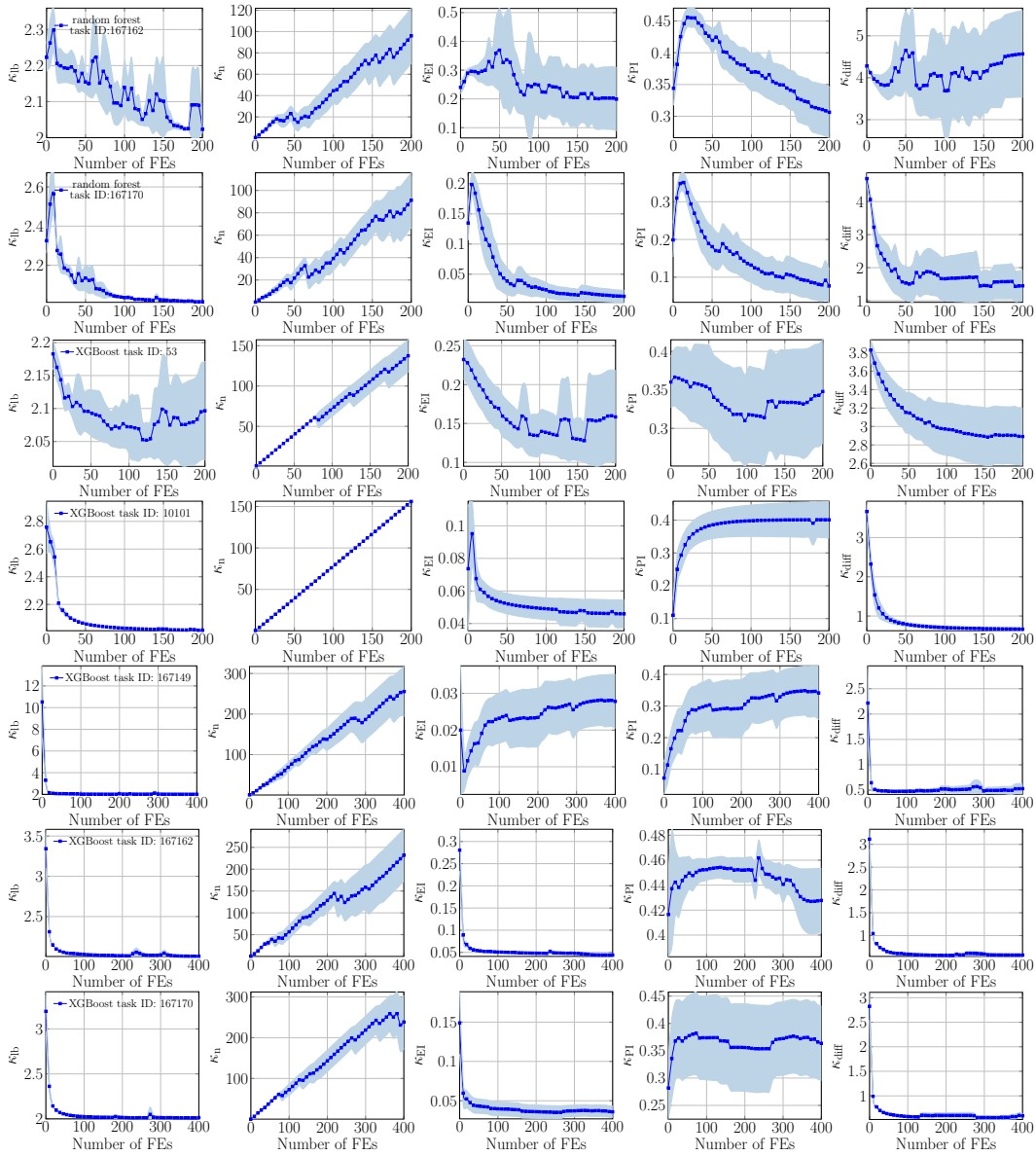

Figure 28: Trajectories of different termination indicators versus the number of FEs during the BO process of applying PI on random forest (task ID: $\{167162, 167170\}$) and XGBoost (task ID: $\{53, 10101, 167149, 167162, 167170\}$).

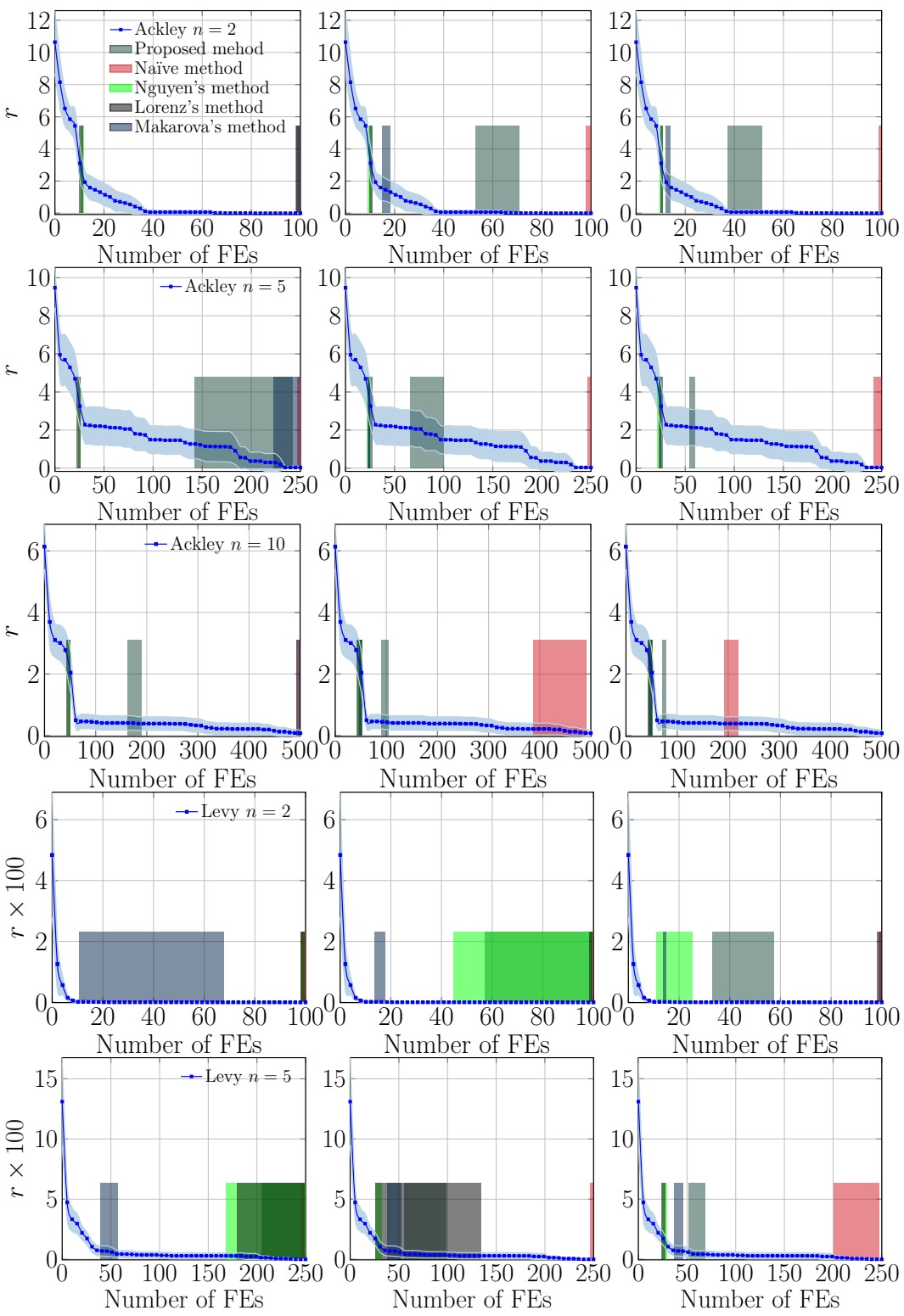

Figure 29: Trajectories of the regret of BO versus the number of FEs during the BO process of applying UCB on Ackley ($n \in \{2, 5, 10\}$) and Levy ($n \in \{2, 5\}$). The results of the first column, second column, and third column are obtained by using different settings of the termination threshold suggested in Section 3.2 of the main paper respectively.

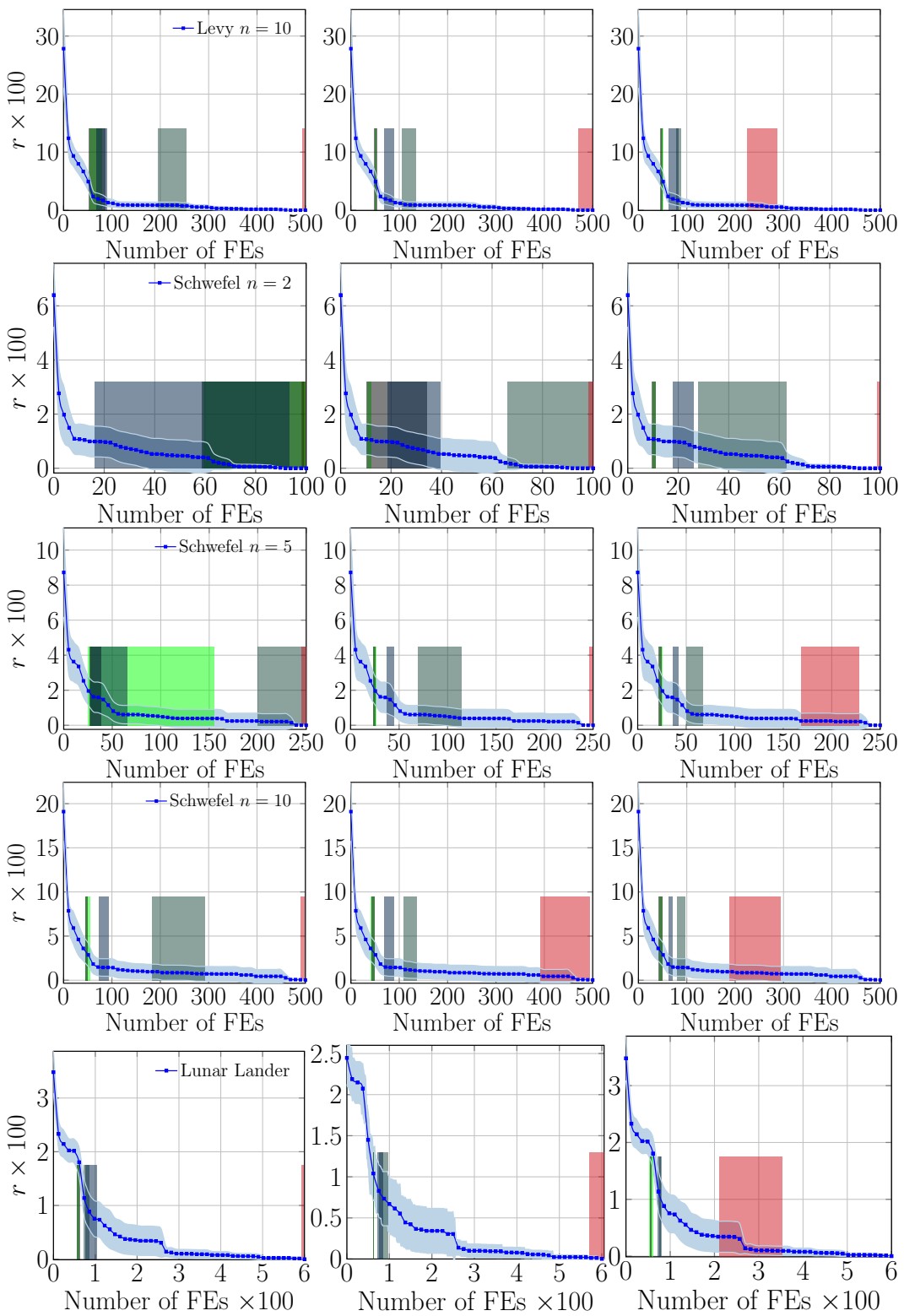

Figure 30: Trajectories of the regret of BO versus the number of FEs during the BO process of applying UCB on Levy ($n = 10$), Schwefel ($n \in \{2, 5, 10\}$) and Lunar Lander. The results of the first column, second column, and third column are obtained by using different settings of the termination threshold suggested in Section 3.2 of the main paper respectively.

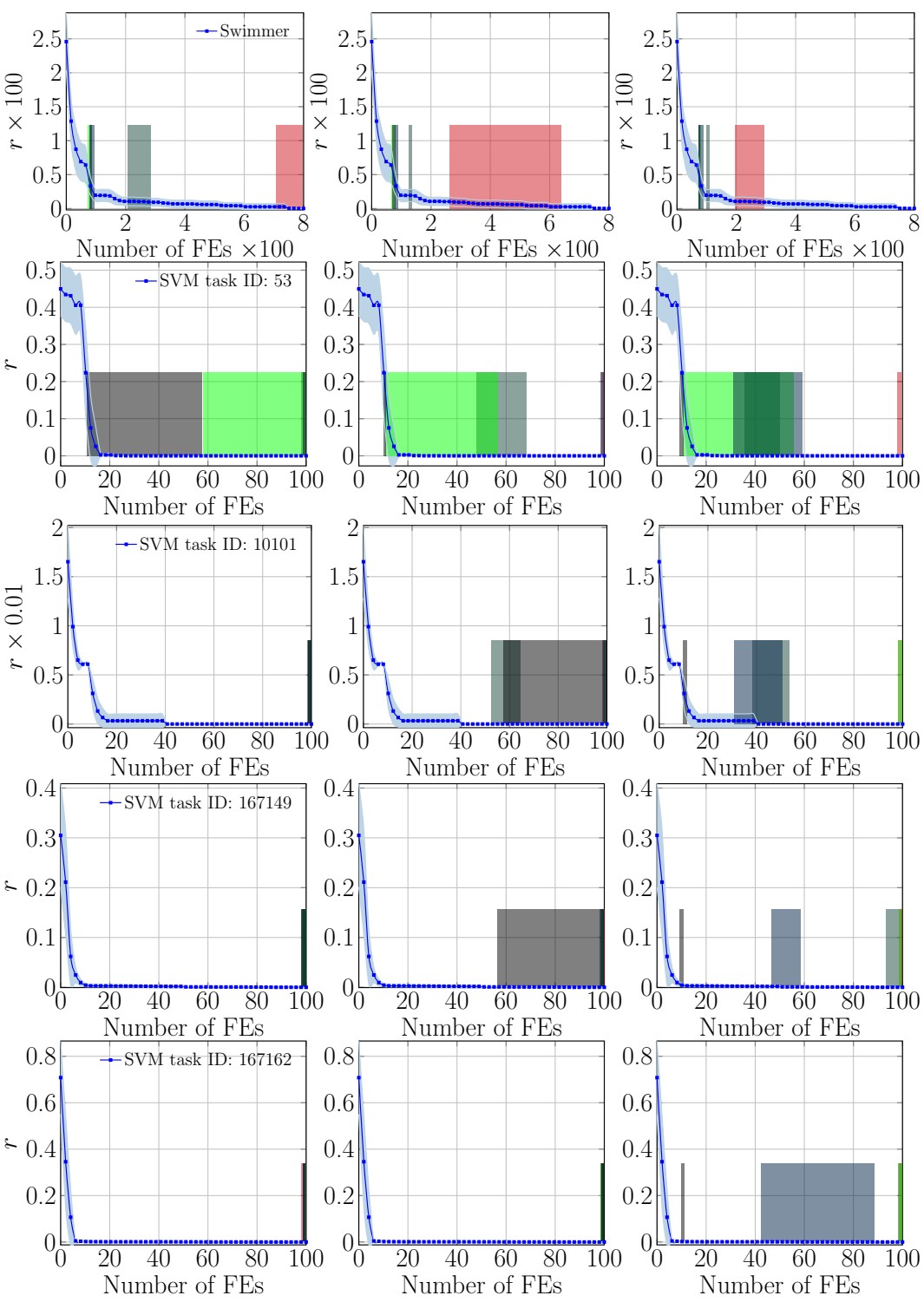

Figure 31: Trajectories of the regret of BO versus the number of FEs during the BO process of applying UCB on Swimmer and SVM (task ID: $\{53, 10101, 167149, 167162\}$). The results of the first column, second column, and third column are obtained by using different settings of the termination threshold suggested in Section 3.2 of the main paper respectively.

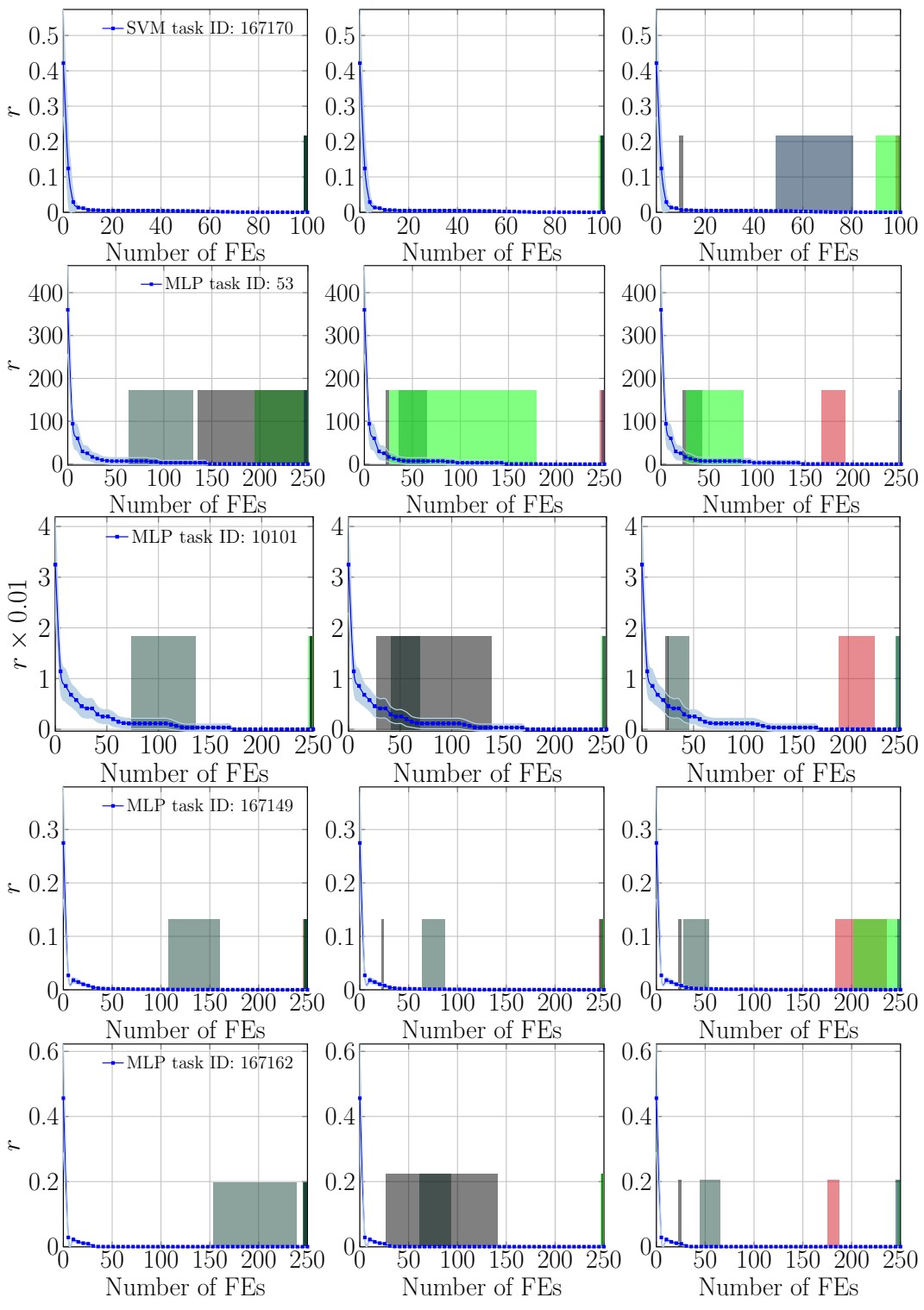

Figure 32: Trajectories of the regret of BO versus the number of FEs during the BO process of applying UCB on SVM (task ID: 167170) and MLP (task ID: {53, 10101, 167149, 167162}). The results of the first column, second column, and third column are obtained by using different settings of the termination threshold suggested in Section 3.2 of the main paper respectively.

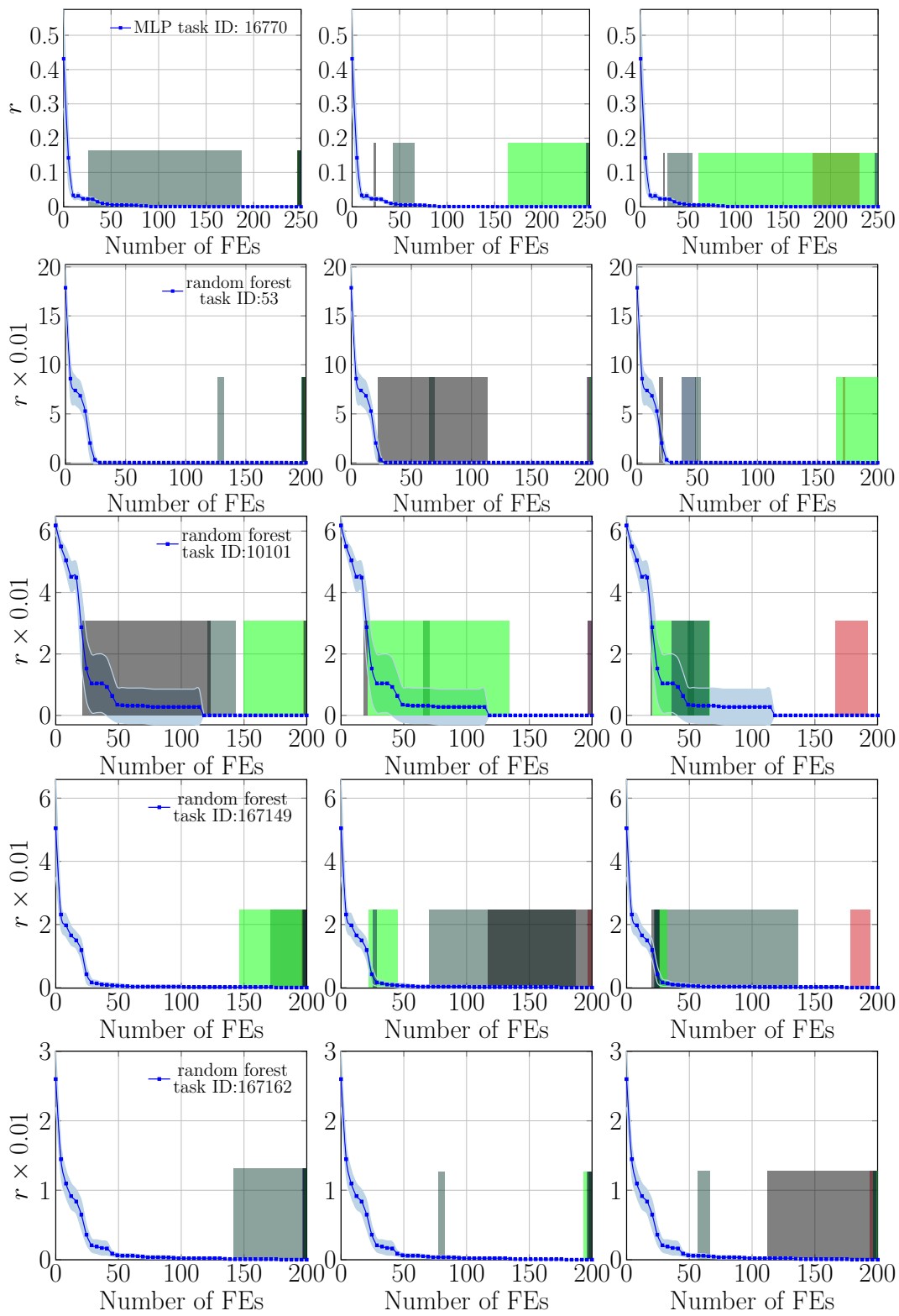

Figure 33: Trajectories of the regret of BO versus the number of FEs during the BO process of applying UCB on MLP (task ID: 167170) and random forest (task ID: {53, 10101, 167149, 167162}). The results of the first column, second column, and third column are obtained by using different settings of the termination threshold suggested in Section 3.2 of the main paper respectively.

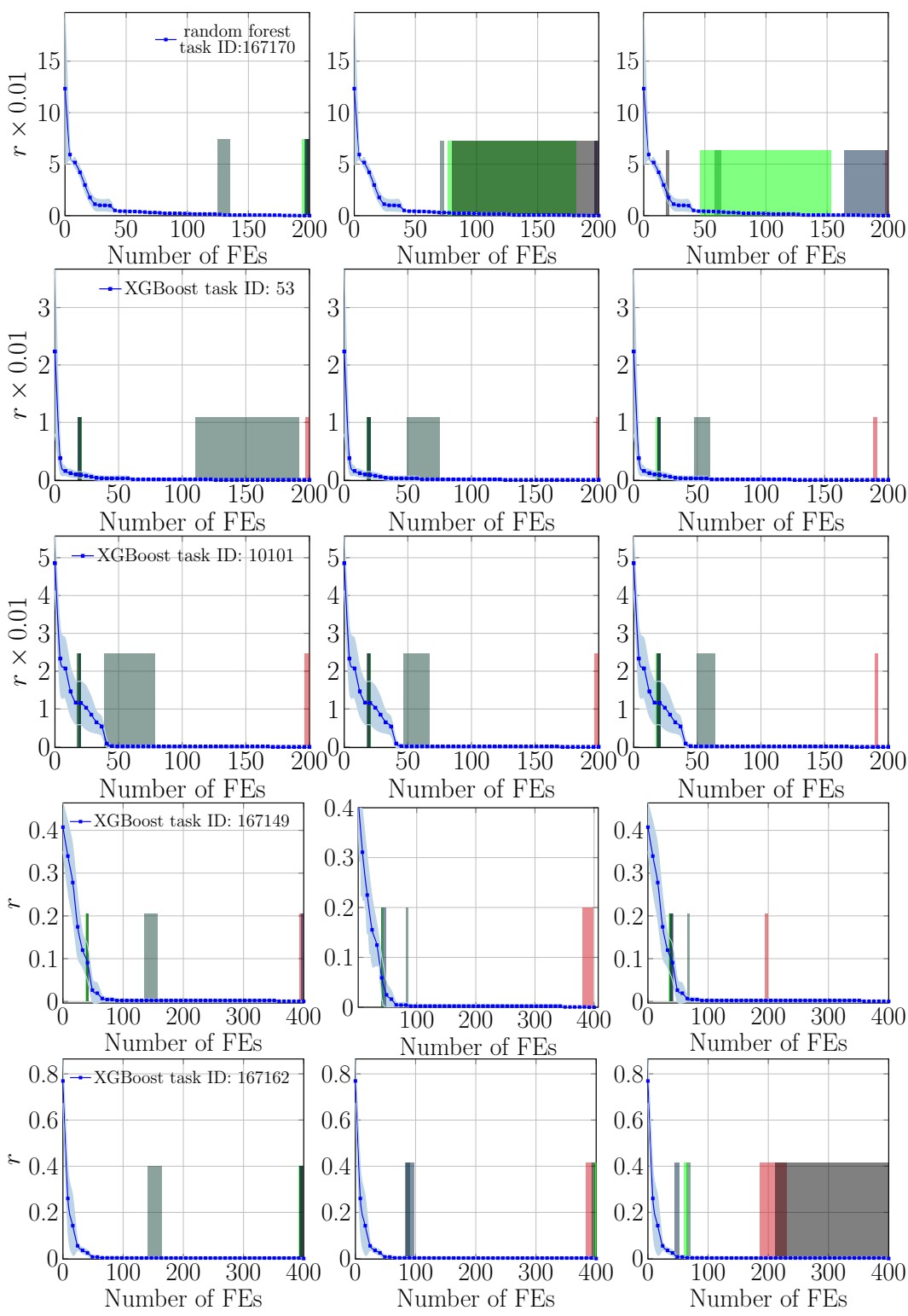

Figure 34: Trajectories of the regret of BO versus the number of FEs during the BO process of applying UCB on random forest (task ID: 167170) and XGBoost (task ID: {53, 10101, 167149, 167162}). The results of the first column, second column, and third column are obtained by using different settings of the termination threshold suggested in Section 3.2 of the main paper respectively.

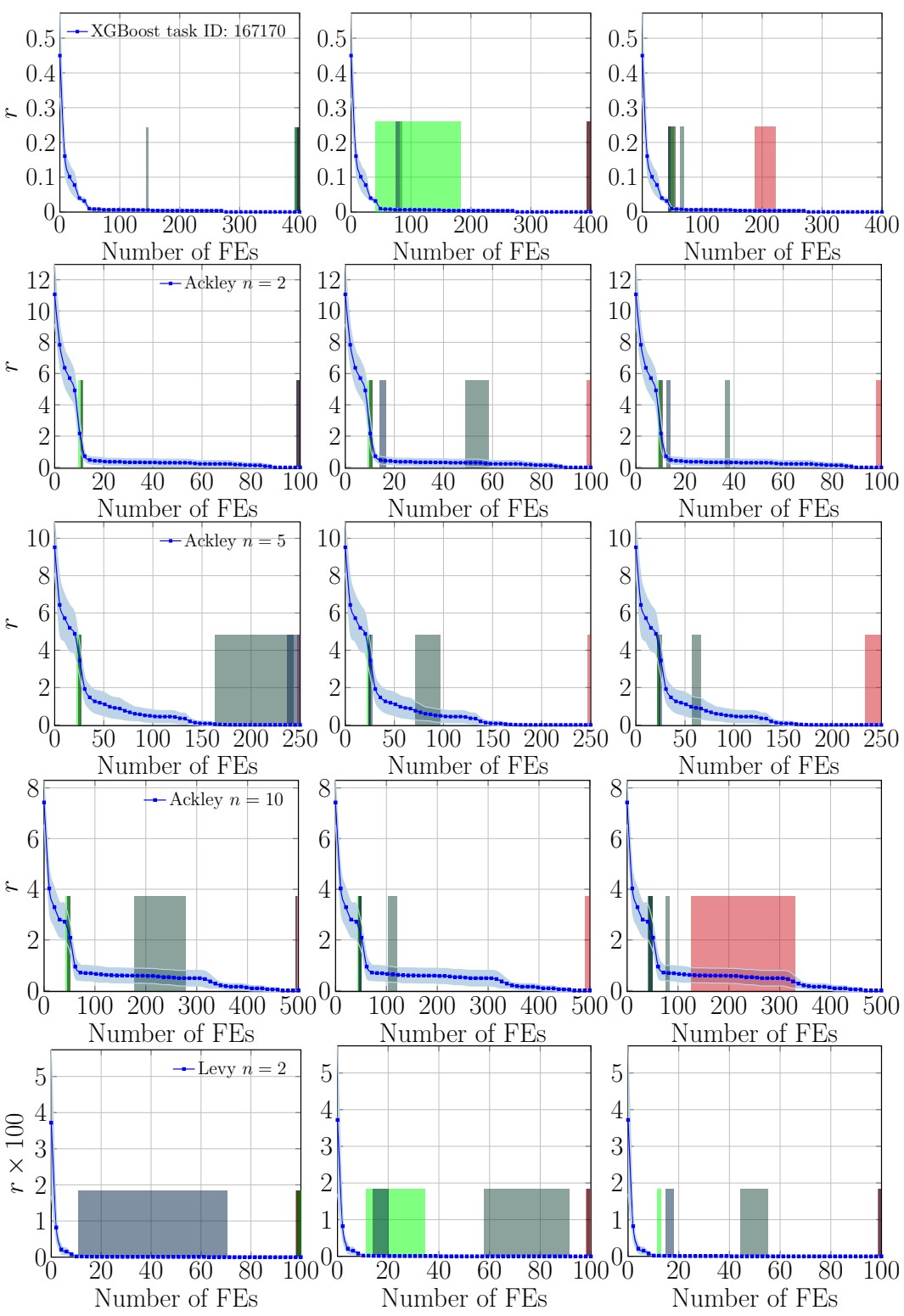

Figure 35: Trajectories of the regret of BO versus the number of FEs during the BO process of applying UCB on XGBoost (task ID: 167170), and applying EI on Ackley ($n \in \{2, 5, 10\}$) and Levy ($n = 2$). The results of the first column, second column, and third column are obtained by using different settings of the termination threshold suggested in Section 3.2 of the main paper respectively.

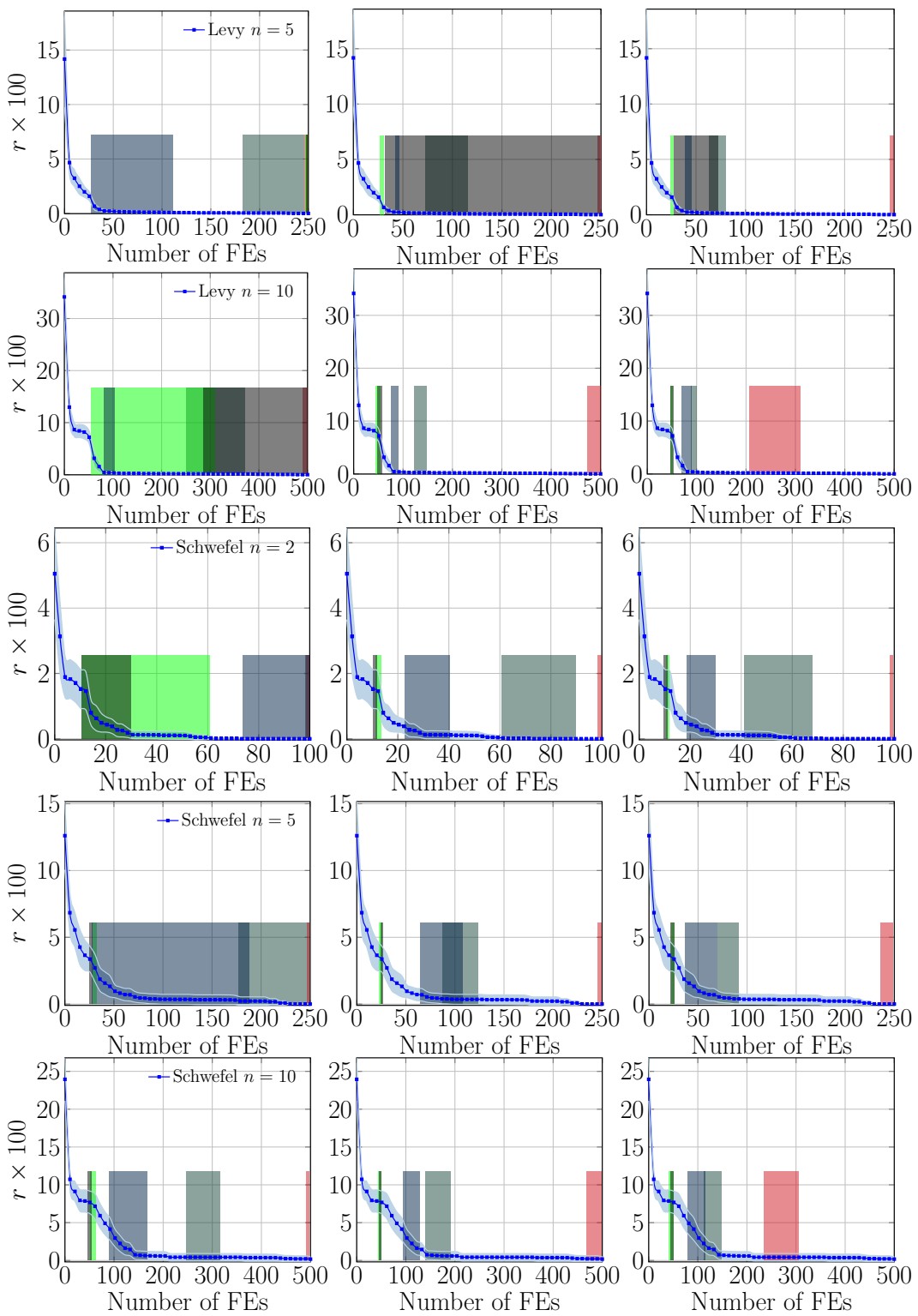

Figure 36: Trajectories of the regret of BO versus the number of FEs during the BO process of applying EI on Levy ($n \in \{5, 10\}$) and Schwefel ($n \in \{2, 5, 10\}$). The results of the first column, second column, and third column are obtained by using different settings of the termination threshold suggested in Section 3.2 of the main paper respectively.

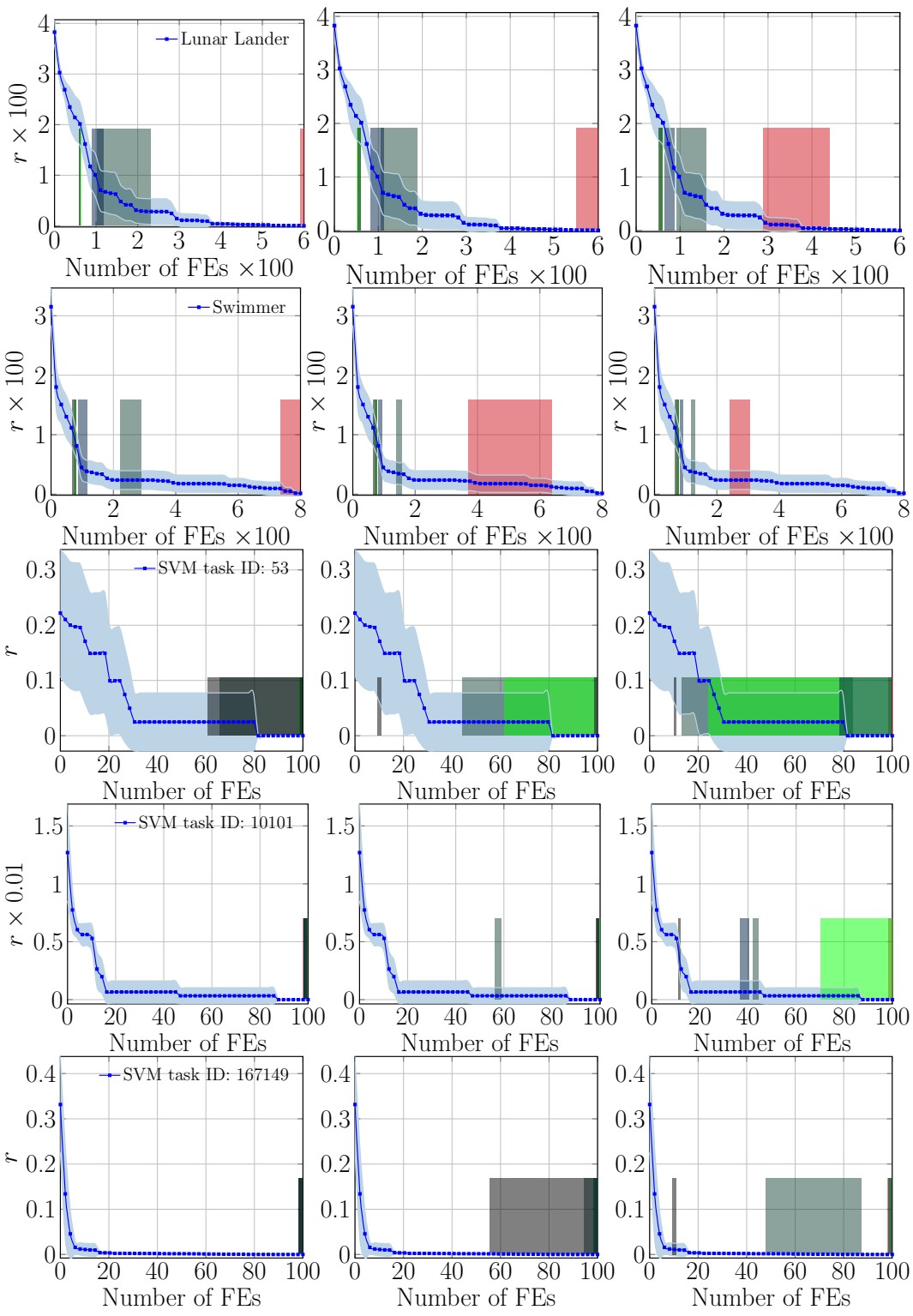

Figure 37: Trajectories of the regret of BO versus the number of FEs during the BO process of applying EI on Lunar Lander, Swimmer, and SVM (task ID: $\{53, 10101, 167149\}$). The results of the first column, second column, and third column are obtained by using different settings of the termination threshold suggested in Section 3.2 of the main paper respectively.

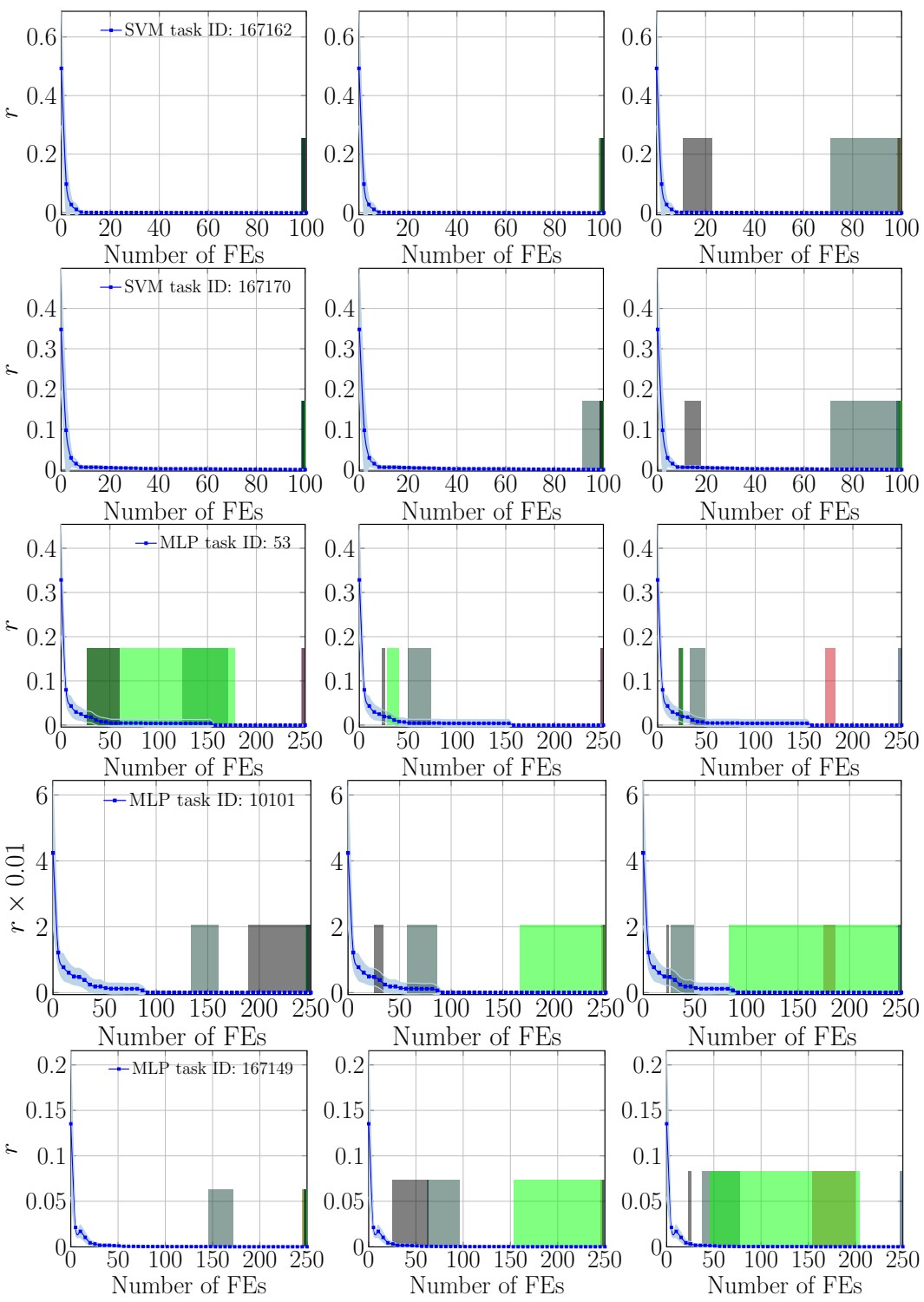

Figure 38: Trajectories of the regret of BO versus the number of FEs during the BO process of applying EI on SVM (task ID: {167162, 167170}) and MLP (task ID: {53, 10101, 167149}). The results of the first column, second column, and third column are obtained by using different settings of the termination threshold suggested in Section 3.2 of the main paper respectively.

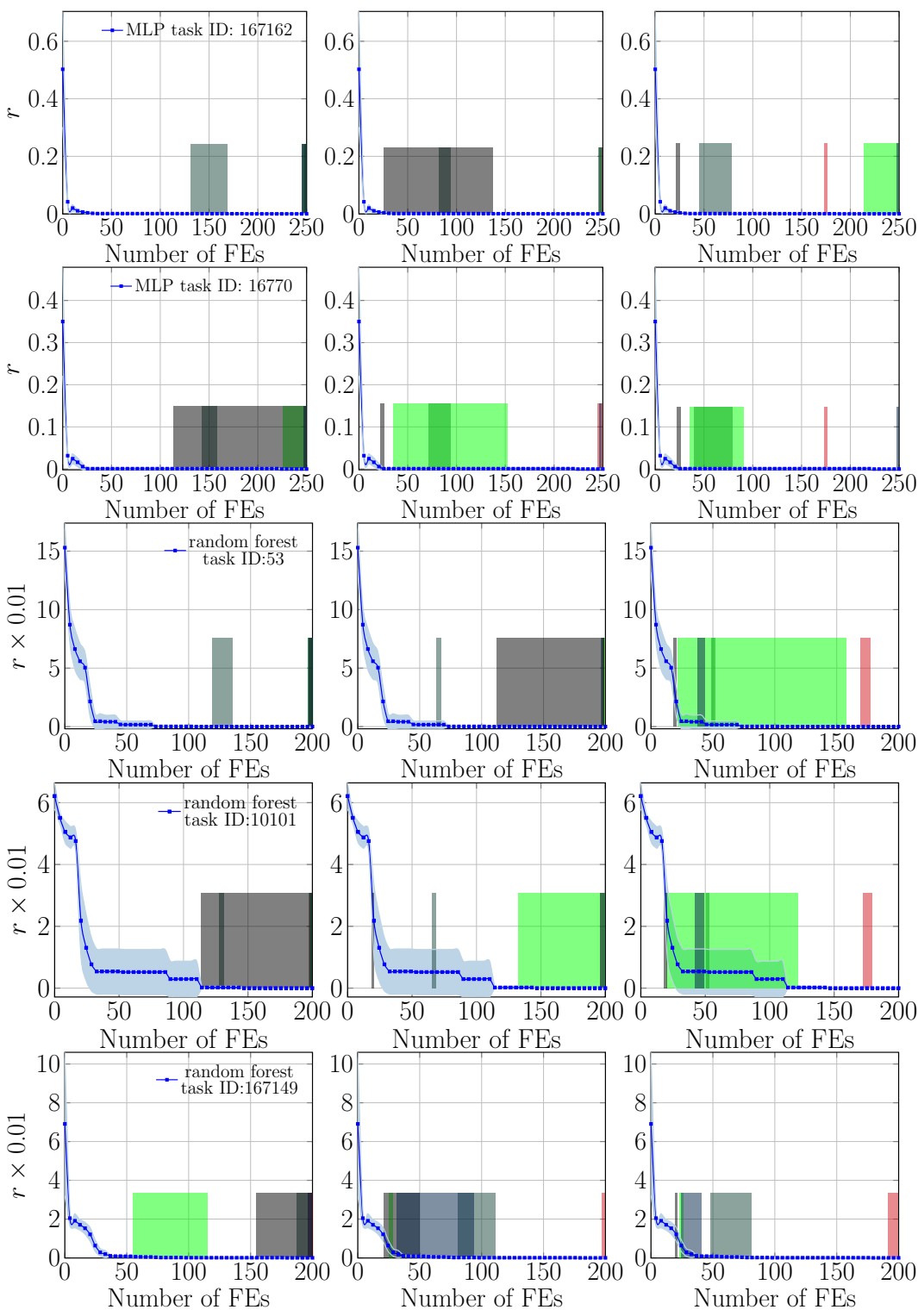

Figure 39: Trajectories of the regret of BO versus the number of FEs during the BO process of applying EI on MLP (task ID: {167162, 167170}) and random forest (task ID: {53, 10101, 167149}). The results of the first column, second column, and third column are obtained by using different settings of the termination threshold suggested in Section 3.2 of the main paper respectively.

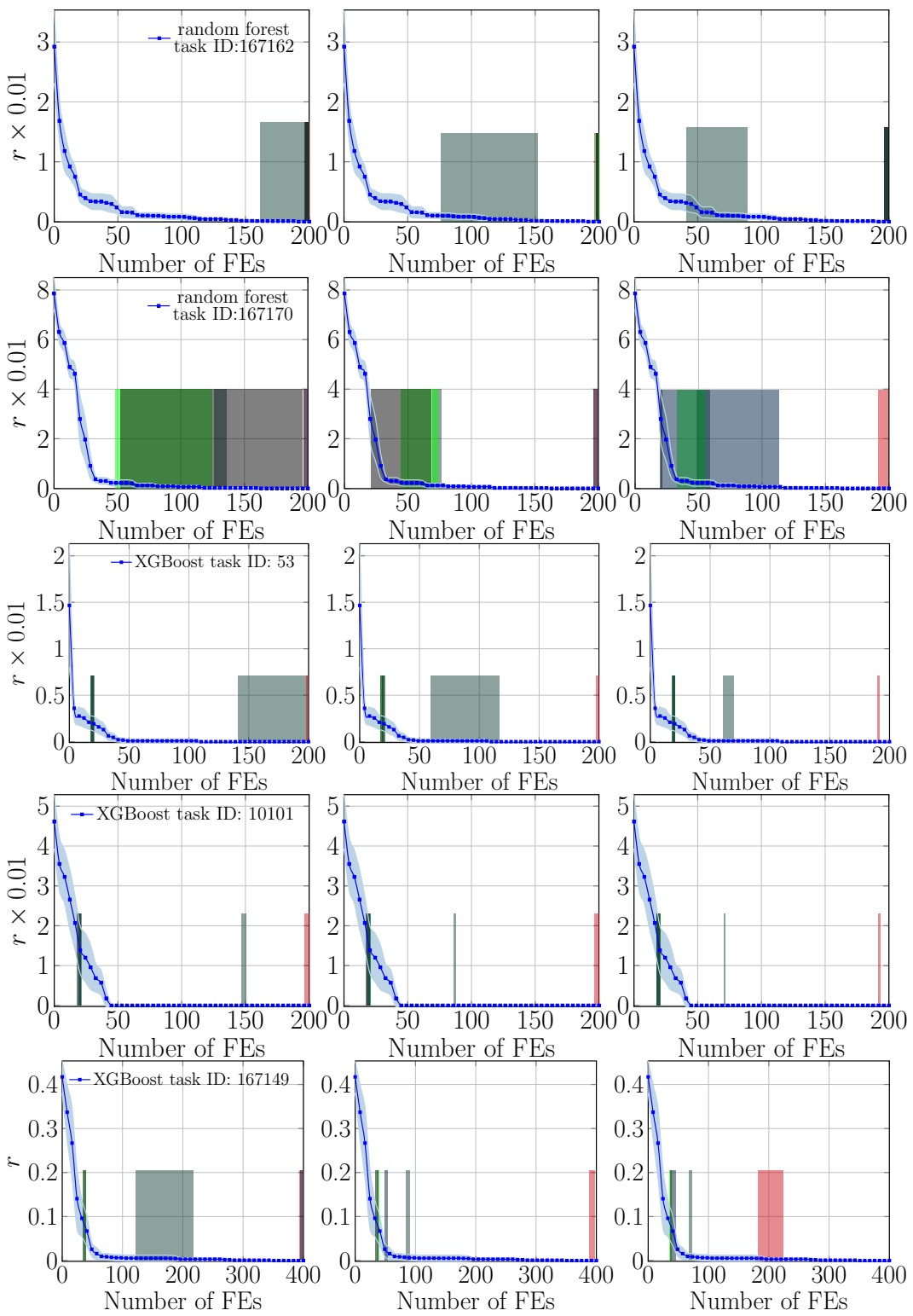

Figure 40: Trajectories of the regret of BO versus the number of FEs during the BO process of applying EI on random forest (task ID: {167162, 167170}) and XGBoost (task ID: {53, 10101, 167149}). The results of the first column, second column, and third column are obtained by using different settings of the termination threshold suggested in Section 3.2 of the main paper respectively.

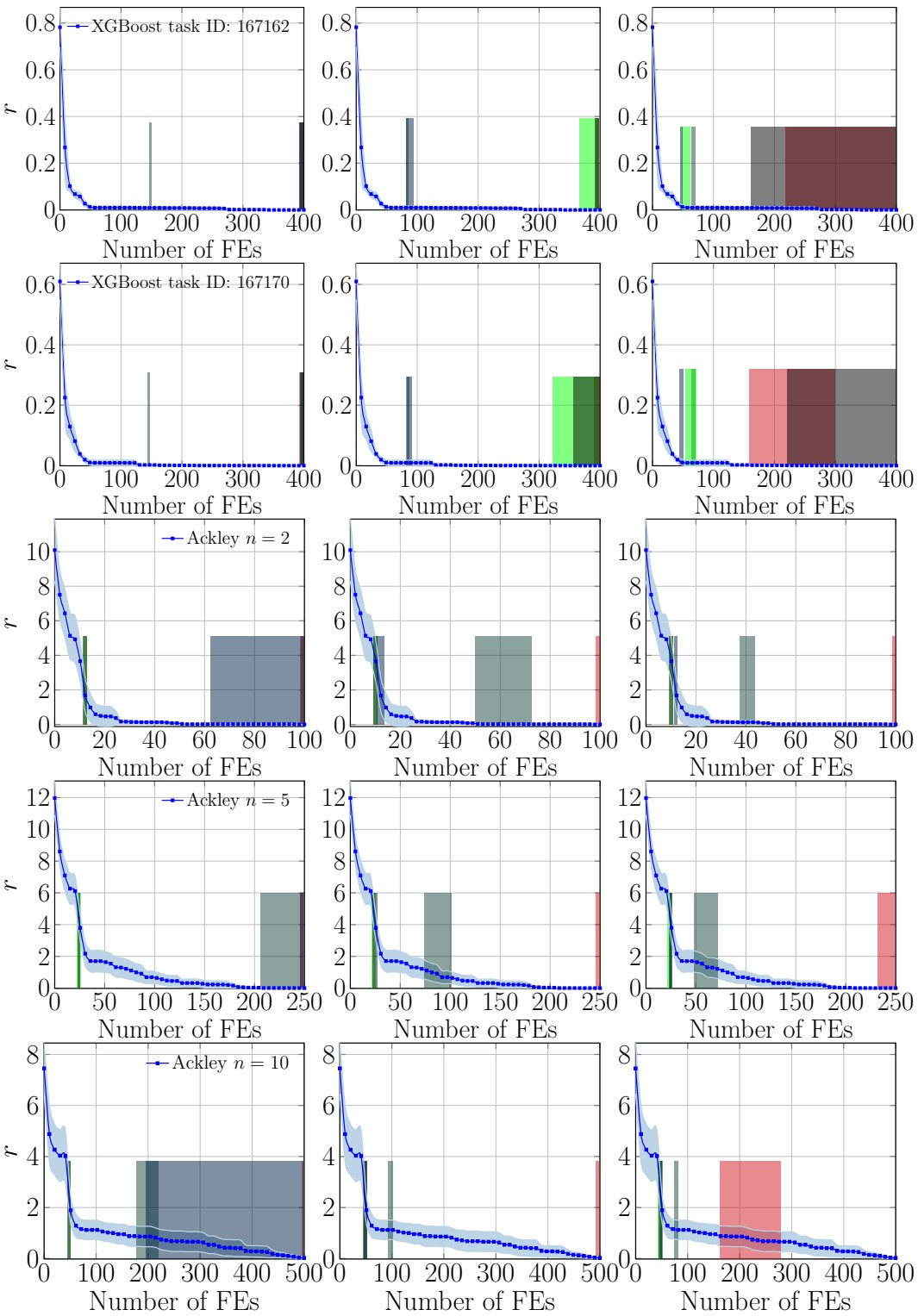

Figure 41: Trajectories of the regret of BO versus the number of FEs during the BO process of applying EI on XGBoost (task ID: {167162, 167170}), and applying PI on Ackley ($n \in \{2, 5, 10\}$). The results of the first column, second column, and third column are obtained by using different settings of the termination threshold suggested in Section 3.2 of the main paper respectively.

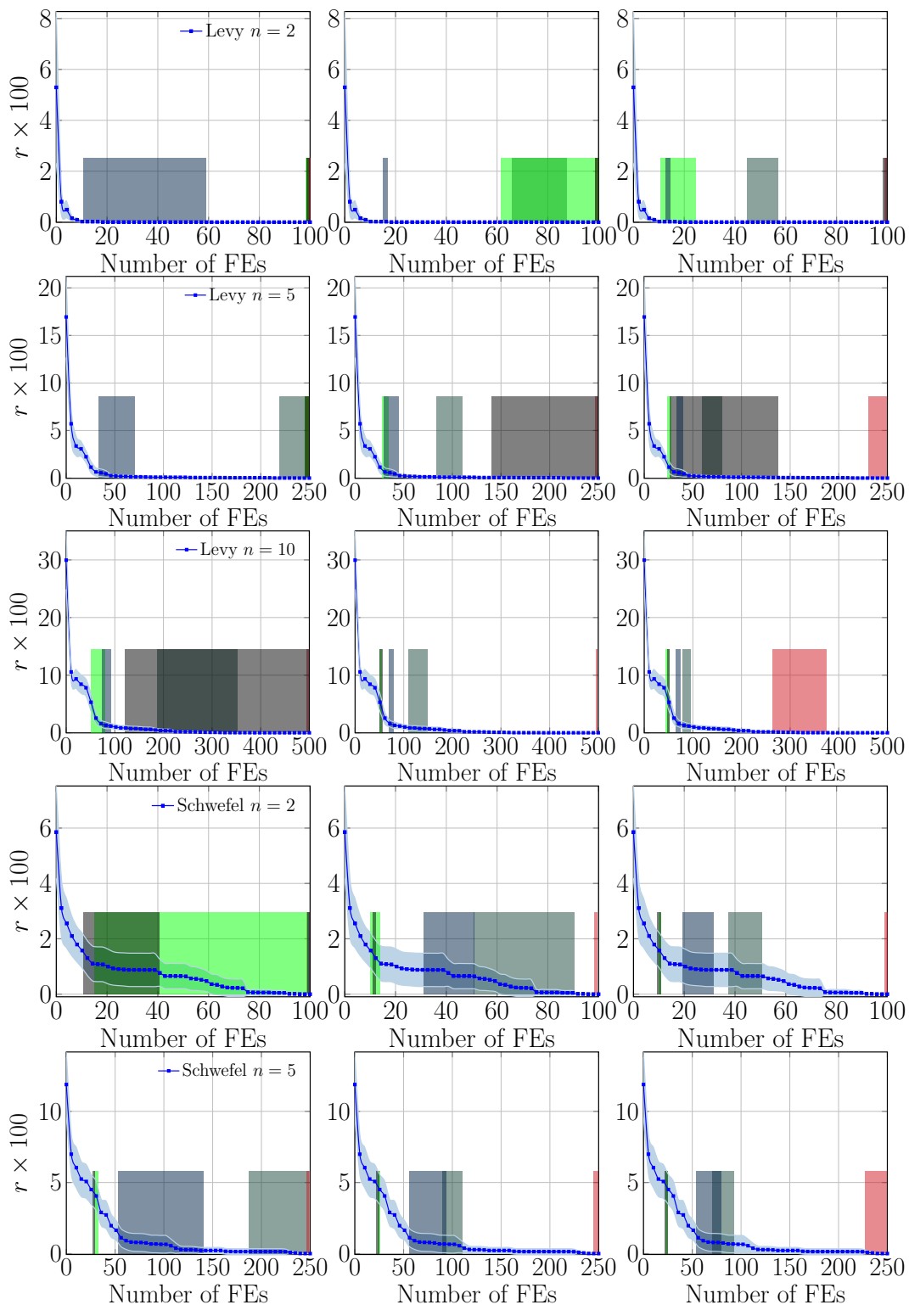

Figure 42: Trajectories of the regret of BO versus the number of FEs during the BO process of applying PI on Levy ($n \in \{2, 5, 10\}$) and Schwefel ($n \in \{2, 5\}$). The results of the first column, second column, and third column are obtained by using different settings of the termination threshold suggested in Section 3.2 of the main paper respectively.

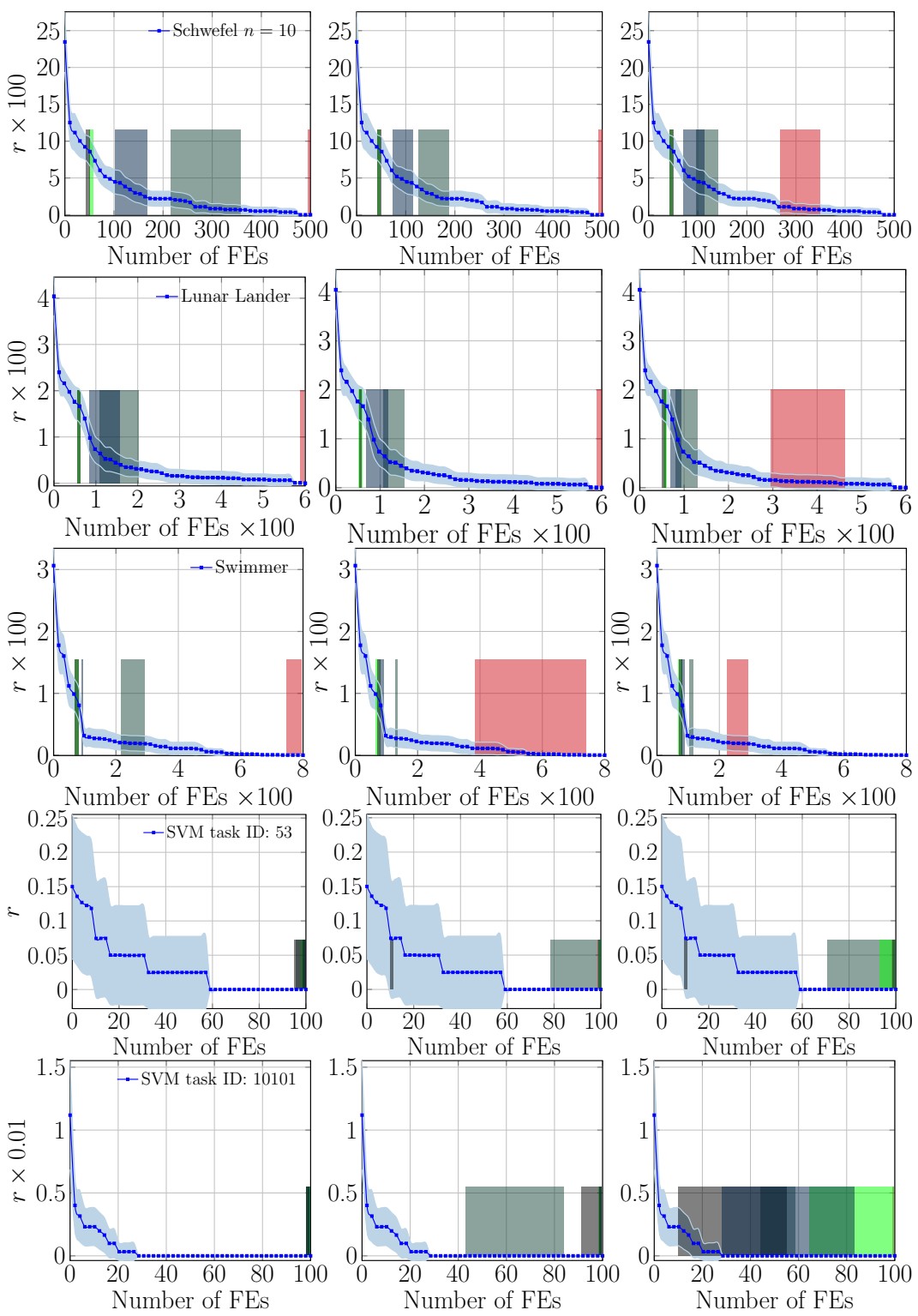

Figure 43: Trajectories of the regret of BO versus the number of FEs during the BO process of applying PI on Schwefel ($n = 10$), Lunar Lander, Swimmer and SVM (task ID: $\{53, 10101\}$). The results of the first column, second column, and third column are obtained by using different settings of the termination threshold suggested in Section 3.2 of the main paper respectively.

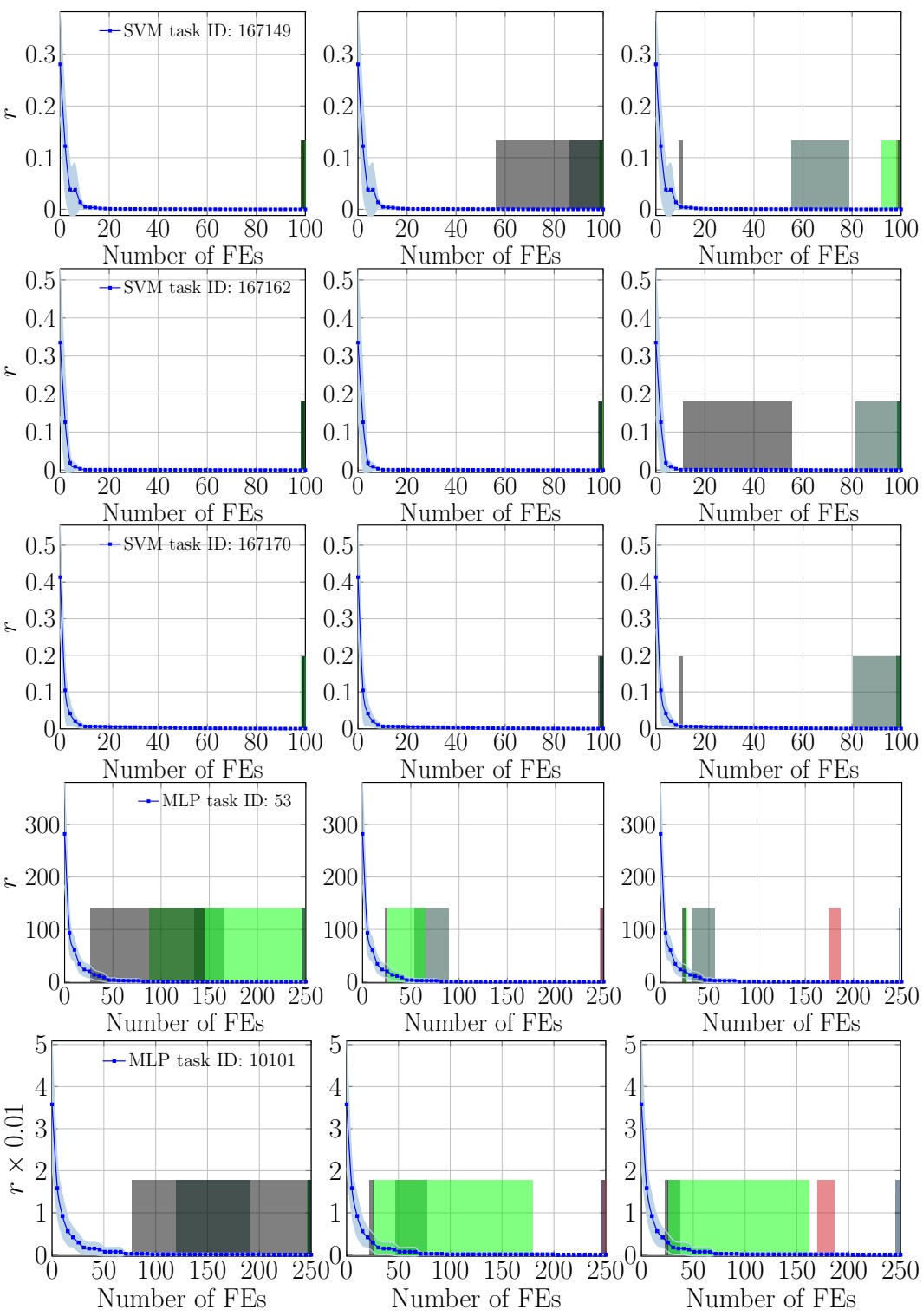

Figure 44: Trajectories of the regret of BO versus the number of FEs during the BO process of applying PI on SVM (task ID: $\{167149, 167162, 167170\}$) and MLP (task ID: $\{53, 10101\}$). The results of the first column, second column, and third column are obtained by using different settings of the termination threshold suggested in Section 3.2 of the main paper respectively.

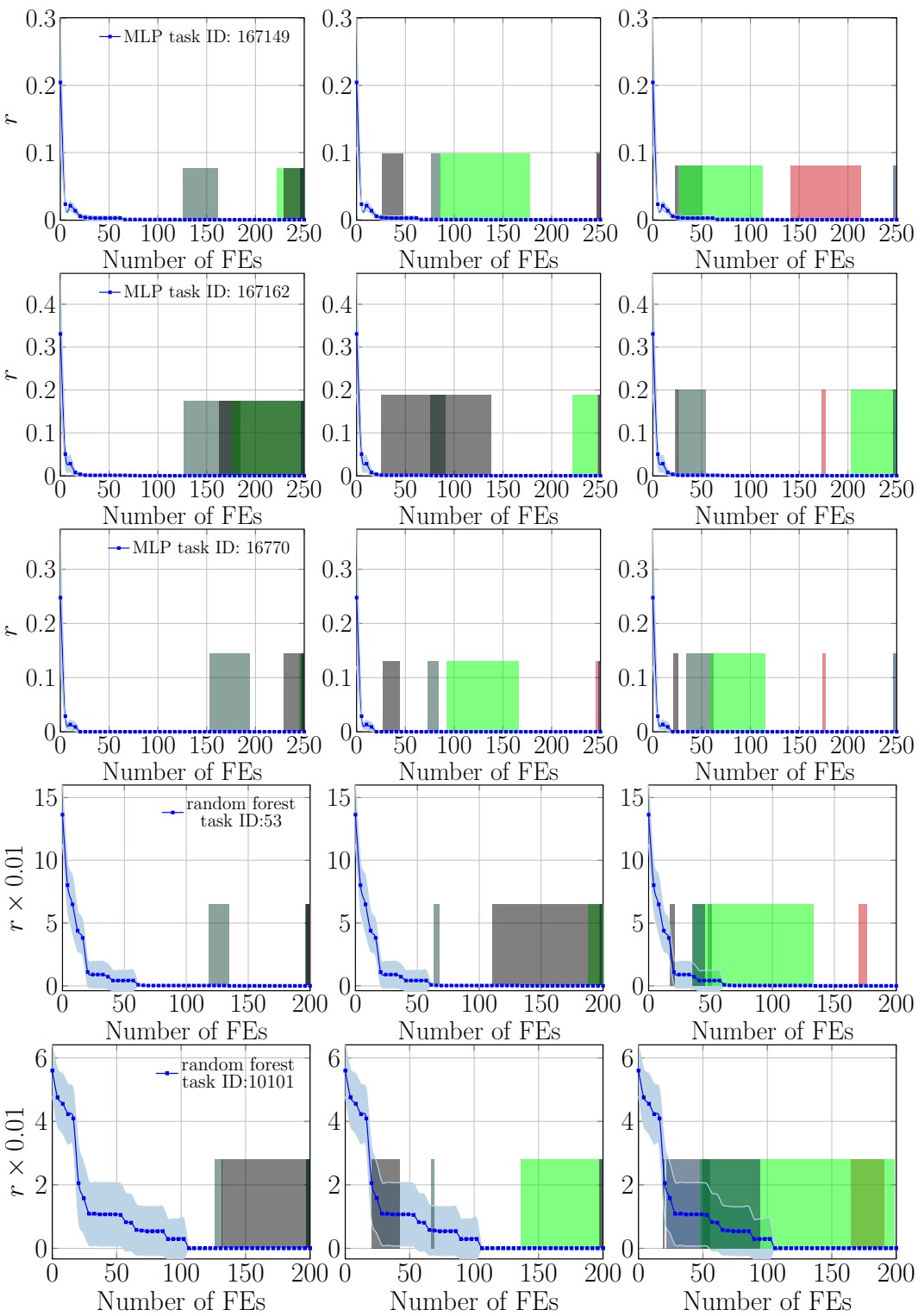

Figure 45: Trajectories of the regret of BO versus the number of FEs during the BO process of applying PI on MLP (task ID: $\{167149, 167162, 167170\}$) and random forest (task ID: $\{53, 10101\}$). The results of the first column, second column, and third column are obtained by using different settings of the termination threshold suggested in Section 3.2 of the main paper respectively.

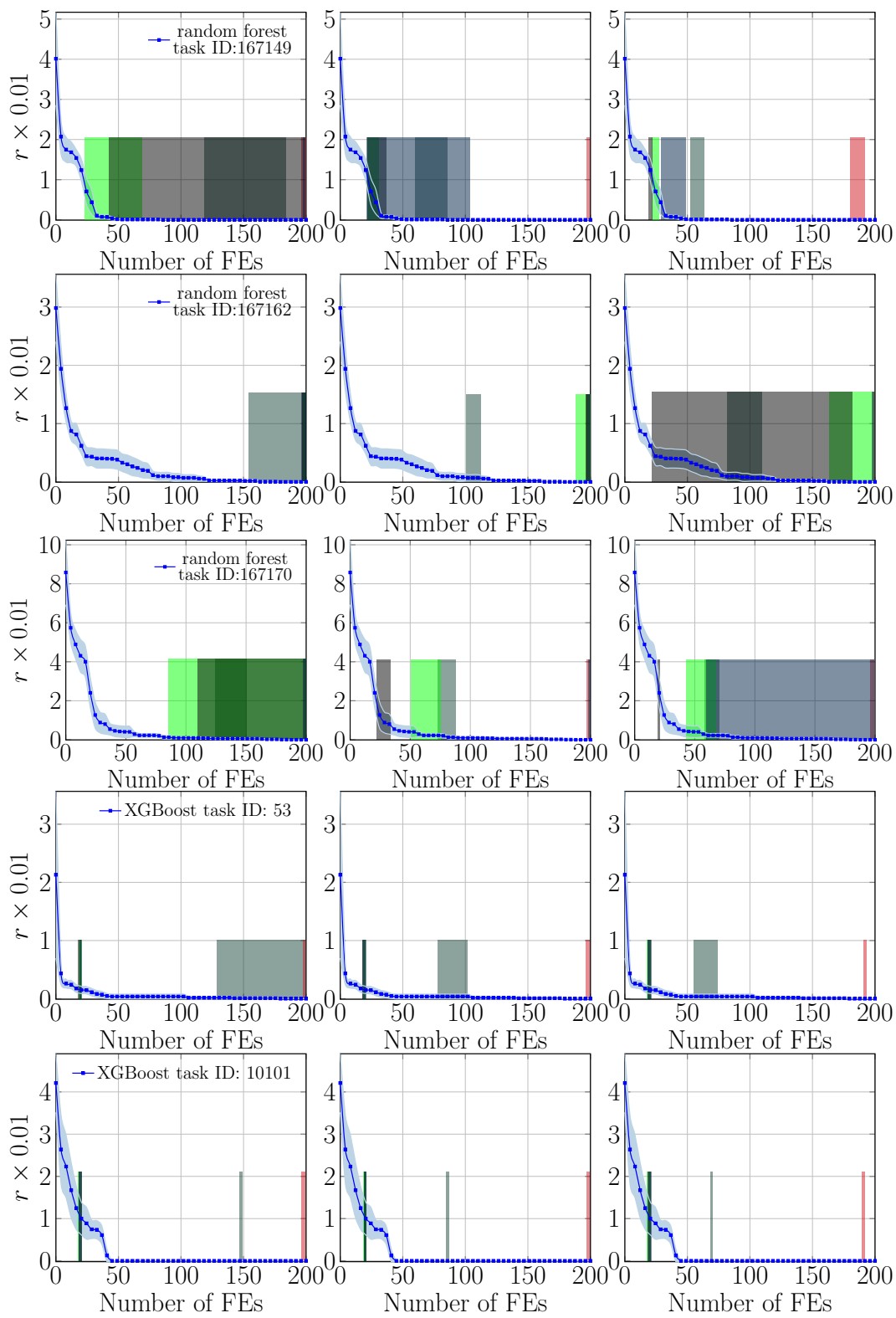

Figure 46: Trajectories of the regret of BO versus the number of FEs during the BO process of applying PI on random forest (task ID: {167149, 167162, 167170}) and XGBoost (task ID: {53, 10101}). The results of the first column, second column, and third column are obtained by using different settings of the termination threshold suggested in Section 3.2 of the main paper respectively.

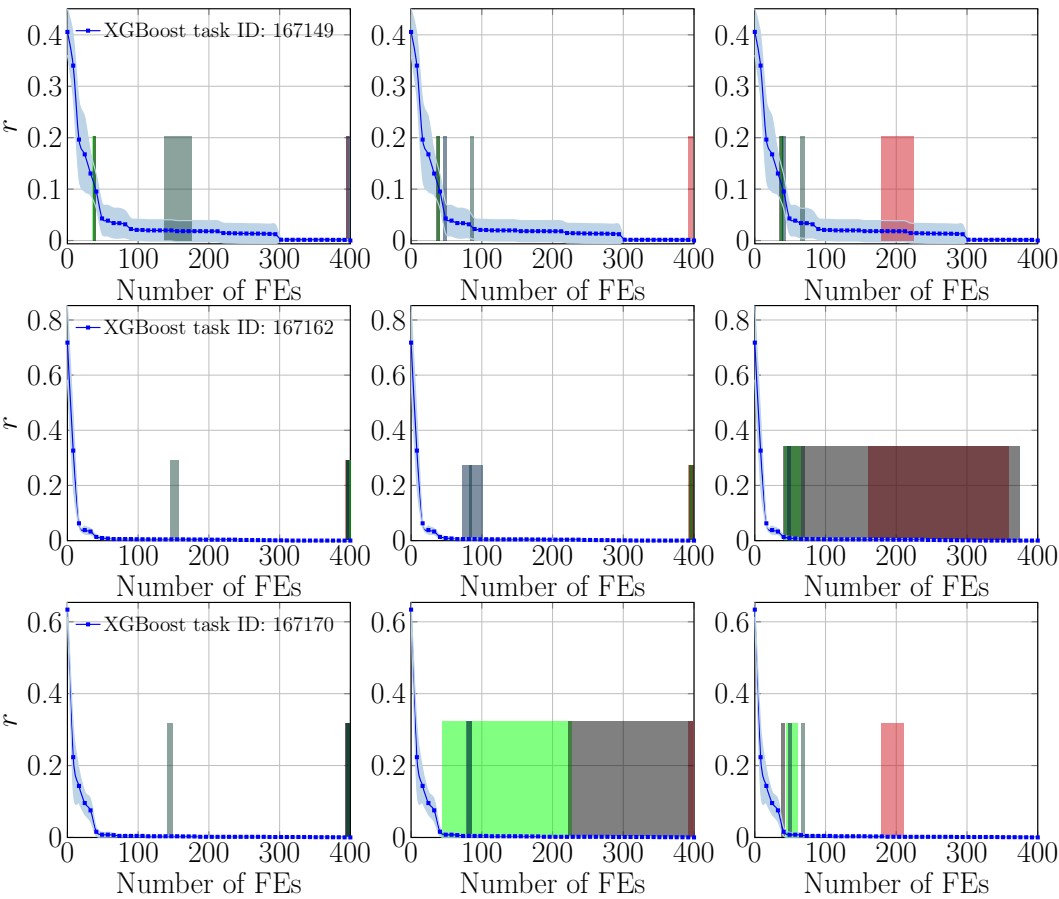

Figure 47: Trajectories of the regret of BO versus the number of FEs during the BO process of applying PI on XGBoost (task ID: {167149, 167162, 167170}). The results of the first column, second column, and third column are obtained by using different settings of the termination threshold suggested in Section 3.2 of the main paper respectively.