# OpenReview forum: "“Why Not Looking backward?” A Robust Two-Step Method to Automatically Terminate Bayesian Optimization"
_NeurIPS.cc/2023/Conference — NeurIPS 2023 poster_

### Official Review · Reviewer_ffwe · 2023-06-21

**Soundness:** 2 fair
**Presentation:** 2 fair
**Contribution:** 2 fair
**Rating:** 4
**Confidence:** 5

**Summary:**

The authors of this paper propose a new stopping criterion for Bayesian optimization. The idea is to consider the last designs visited during optimization to compute a local regret involving another optimization problem. A theoretical analysis is provided as well as an empirical comparison with some alternative termination criteria.

**Strengths:**

- a theoretical analysis of the proposed criterion is given.
- the empirical results are promising.

**Weaknesses:**

- stopping criteria have been proposed since the early works on BO, e.g., the discussion by Jones et al. It would be worth considering this option (optimization of EI with convergence guarantees). Another relevant work to compare with is Queipo, N. V., Pintos, S., & Nava, E. (2013). Setting targets for surrogate-based optimization. Journal of Global Optimization, 55, 857-875.
- the description of the BO setup is vague: what is used for the acquisition function search? Are the hyperparameters optimized? Plus it is typical to increase omega in UCB and decrease lengthscales over time.
- The selection of the threshold \eta and the chosen metrics are not easy to interpret (see more detailed questions below).

** The rebuttal and discussion have been lengthy. My main criticisms remains and can be summarized in two points:
- the link between the general idea and the actual criterion is not clear, and it is not obvious how to use it to control the level of convergence or accuracy in practice.
- the comparison with existing works lacks at least the works by Jones et al., to test the robustness against model mis-specification and quality of the acquisition function optimization.

**Questions:**

- Could you give a high level description of the conditions ?
- Could you give an interpretation of the threshold? For instance, criteria on expected improvement and probability of improvement are easy to choose. It would be more natural to select \tilde{r} as threshold and then do the empirical comparison based on it (this is partially done in Fig. 7).
- Similarly, in Section 3.2 the values considered seem quite arbitrary, how are they chosen?
- What happens on multi-modal test functions such as Branin, where local minima are also global ?
- Could you compare with McLeod et al., that also consider a convexity test?
- On figure 2: are the hyperparameters fixed? The lengthscale is clearly too large.
- Could you provide a pseudo-code of the computation of the stopping criterion?

---

> ### Author Rebuttal · Authors · 2023-08-09
>
> **W#1:** Thank you for your suggestion, we will add the two papers you mentioned to Section 5 of the main paper. However, both the reviewer mentioned works are not applicable in our context. As for the Jones et al.'s work, it terminates BO when EI is less than 1% of the current optimal value. However, since EI is always positive, it cannot be applied to problems with negative objective values. As for the Queipo et al.'s method, it estimates the optimum of the underlying problem as a probability density function based on the training data. It cannot be used as a termination criterion because the data is collected in a sequential manner on-the-fly in BO.
>
> **W#2:** In this work, the hyperparameters of a Gaussian process (GP) model are obtained by solving the maximal likelihood estimation via L-BFGS, so as the optimization of the acquisition function.
>
> **W#3:** The rationality of our threshold selection is delineated in the response to your **Q#3**.
>
> **Q#1:** **Condition 1** detects whether the search process of BO already reaches a convex region of $-f(\mathbf{x})$, by " looking back" $\tau>1$ observed points; while **Condition 2** evaluates whether the local regret within this convex region falls below a predefined threshold. When both conditions are met, we consider BO cannot find better candidates any longer and thus can be terminated.
>
> **Q#2:** There are two reasons for using $\frac{\tilde{r}}{\omega \sigma_\epsilon}$ as the termination indicator.
>
> - We found that $\mu(\dot{\mathbf{x}})-\mu(\tilde{\mathbf{x}})$ quickly diminishes to $0$ during the search process of BO. This leads to $\tilde{r}$ turn out to be $\omega(\sigma(\ddot{\mathbf{x}})+\sigma(\tilde{\mathbf{x}}))$. In this case, we want to mitigate the effect of $\omega$ on $\tilde{r}$ by using $\frac{\tilde{r}}{\omega\sigma_\epsilon}$.
> - The noise level, quantified by $\sigma_\epsilon$ in this paper, varies across different problems. Since $\sigma(\mathbf{x})$ will increase with $\sigma_\epsilon$, it implies that $\frac{\tilde{r}}{\omega \sigma_\epsilon}$ can reduce the effect of $\sigma_\epsilon$ on $\sigma(\ddot{\mathbf{x}})$ and $\sigma(\tilde{\mathbf{x}})$, thereby enhancing the adaptability of predefined threshold on different noise levels.
>
> **Q#3:** We argue that the threshold selection is not ad-hoc. It relies on the calculation of **$\mathrm{I}_\mathrm{cdf}$**, which is applied to estimate the distribution of different termination indicators with regard to **$\tilde{\kappa}_i$** collected from different problems in our experiments (details are given in Section 4.1). Such statistics constitute the rationality of the threshold selection provided in Section 3.2. To help your understanding, we provide an illustrative example on Ackley with **$N_\mathrm{FE}=5$** in **Fig. 3 of the author rebuttal PDF**.
>
> **Q#4:** Although we did not test on Branin, Levy function considered in our experiments is similar to the reviewer mentioned Branin function. Both of them have large flat regions, within which the global optimum is located. The empirical results showed that our proposed algorithm effectively saves FEs while concurrently minimizing regret. Specifically, across the three threshold configurations, our method achieves substantial FE savings (approximately **22.3%**, **56.1%**, and **68.1%**, respectively). Please refer to Pages 22, 23, 28, 29, and 35 in the supplementary file.
>
> **Q#5:** There are three reasons that make McLeod et al.'s method cannot be compared with us.
>
> - The motivation of the convexity test is different. McLeod et al. used the convexity test as an indicator to switch between the two different acquisition functions, i.e., the predictive entropy search and the global regret reduction. Differently, we use convexity test to evaluate the local regret within local convex regions.
>
> - The mathematics of the convexity test is different. McLeod et al. investigate the positive definiteness of a Hessian matrix; while we reply on the Jensen's inequality.
>
> - Since the switching criterion, search strategy, and the acquisition function are closely tiled in the McLeod et al.'s method, the convexity test itself cannot be decoupled as a standalone criterion to serve the early termination purpose.
>
>
> **Q#6:** As stated in our response to your **W#2**, the hyperparameters of the GP model in Figure 2 are not fixed a priori. They are obtained by maximizing the marginal likelihood function. As for the lengthscale, we set it to be within $[0.05,200]$ during the GP training process to make sure it remains within reasonable limits.
>
> **Q#7:** Sorry for this confusion, the corresponding pseudo-code is given in Page 1 of the supplementary file.

---

> > ### Comment · Reviewer_ffwe · 2023-08-14
> >
> > Thank you for the detailed reply. I have follow up comments and questions.
> >
> > W#1: The Jones et al. 1998 approach is used in practice, it is not very hard to handle the negative function values case. As for the Queipo approach, this again does not prevent using it. Other elements are provided in the recent book on BO by R. Garnett.
> >
> > Q#1: ok, the typical behavior of BO is to alternate between exploration and exploitation, so not sampling too often the same region.
> >
> > Q#2: ok, but what about the interpretation in practice?
> >
> > Q#5; it is still possible to at least compare condition 1 (on convexity) and McLeod's convexity test.

---

> > > ### Author Response · Authors · 2023-08-17
> > >
> > > Thanks for your comment, our responses are as follows:
> > >
> > > **W#1**: We address the reviewer’s concern from the following three aspects.
> > > - We agree it is not difficult to just transform the objective function with negative values of $f(\mathbf{x})$ into a positive objective space if we want to use Jones et al.’s approach. However, it is also worth noting that the transformation function itself can change the landscape of $f(\mathbf{x})$, so as the location of the corresponding optimum. This will lead to an unclear impact on the performance of the termination criterion. For example,
> > >    - if we choose an absolute transformation like $0.01*|f(\mathbf{x})|$, it will continuously increase as BO enters the negative region of $f(\mathrm{x})$. In this case, BO will face the risk of premature termination;
> > >    - if we execute an exponential transformation like $0.01*\exp({f(\mathbf{x})})$ will rapidly converge towards $0$ when BO enters the negative region of $f(\mathrm{x})$. In this case, BO will face the risk of termination failure.
> > >
> > >    Based on these justifications, we believe that the selection of an appropriate transformation approach is non-trivial. Moreover, our experimental study focuses on analyzing the robustness of termination indicators, instead of the adaptive threshold selection strategies. Consequently, we choose the method of setting a fixed predefined threshold for EI (Nguyen et al. 17) as the baseline algorithm.
> > > - Regarding the method of Queipo et al., we acknowledge its potential as a candidate for serving as a termination criterion. However, the method proposed in Queipo et al.'s paper may not be suitable for BO due to the following reasons.
> > >   - They did not address the estimation of the probability density function for the optimal values in the framework of BO, where sampling data will iteratively update and thus are biased.
> > >   - Moreover, they did not propose a method for utilizing the obtained function as a termination criterion. While it is indeed possible to use the expectation, upper or lower confidence bound of the obtained function as the termination criterion, the practical results and theoretical foundation of these approaches are yet to be developed.
> > > - As for Garnett's BO book, we found some interesting discussions of optimal stopping rules based on cost functions (in Chapters 5.4 and 11.1), stopping rules based on optimal observations and stopping rules based on iterations where the current optimal solution remains unchanged (in Chapter 9.3). While most of these topics are covered in our review of related works, we have not included studies on stopping rules involving unknown observation costs (in Chapter 11.1). To address this issue, we will include Garnett's BO book, along with works by F. Hutter et al. (2011), S. Zilberstein (1996), J. Snoek et al. (2012), and G. Malkomes et al. (2016), et al. into the related works section of our paper.
> > >
> > > All in all, we really appreciate the reviewer’s ideas and enjoyed the constructive discussions. We believe a principled termination criterion can go beyond the BO itself and be useful for other algorithmic purposes, because we always want to minimize the computational budget while also maximize the utility.
> > >
> > > **Q#1**: Yes, this is one of the reasons that we ‘look back’ a number of sampling steps of BO. In particular, $\tau$ corresponds to the number of samples a decision maker can tolerate for the BO to sample within the same region.
> > >
> > > **Q#2**: In practice, if there's a change in the confidence level of **condition 2**, modification is required for the threshold of $\tilde{r}$, while the threshold of $\frac{\tilde{r}}{\omega \sigma_\epsilon}$ can remain unchanged. Moreover, $\frac{\tilde{r}}{\omega \sigma_\epsilon}$ is less affected by noise level $\sigma_\epsilon$ than $\tilde{r}$. We hope our response addresses the reviewer's concern.
> > >
> > > **Q#5**: We justify this concern from the following three aspects.
> > >
> > > - First, given the assumption of black-box optimization, it is difficult to evaluate the accuracy of the convexity test, especially in a high-dimensional space.
> > > - Second, as our previous reply, McLeod et al.’s method is highly integrated. That is to say, the convexity test itself is not a standalone method. As for our proposed termination criterion, the **Condition 1** is bonded with **Condition 2**, which is the ultimate criterion for determining the termination of BO. Therefore, comparing only the accuracy of the convexity test does not serve the motivation of our paper.
> > > - Last but not the least, according to the results of Figure 8 in our manuscript (i.e., our ablation study), we find that BO can be terminated by only considering **Condition 2** while **Condition 1** is relaxed with $2\leq\tau\leq18$. Note that such termination is more aggressive, i.e., having the risk of premature termination. In other words, our method contains a certain level of tolerance towards the accuracy of the convexity test method in **Condition 1**.

---

> > > > ### Comment · Area_Chair_SqJs · 2023-08-20
> > > >
> > > > Thank ffwe for engaging with the authors in the discussion. As the author-review discussion period is coming to an end in one day and you hold the only negative rating for this submission, could you please let us know if the authors' comment help address your concern? It's the last chance for the authors to reply to your follow-up questions, if any.

---

> > > > ### Comment · Reviewer_ffwe · 2023-08-21
> > > >
> > > > Thank you for the continued discussion.
> > > >
> > > > W#1: Again, in the seminal work by Jones et al., the choice of a proper transformation is discussed. As done in the paper and completed here, there are many related works such that comparing only with simple baselines (e.g., fixed threshold) may not be suitable.
> > > >
> > > > Q#1: setting $\tau$ may be difficult in practice.
> > > >
> > > > Q#2: perhaps I have not been very clear. Do you recommend fixed values to be used all the time for the additional parameters, and if not, what are the guidelines?
> > > >
> > > > Q#5: ok, thanks

---

> > > > > ### Author Response · Authors · 2023-08-22
> > > > >
> > > > > Thanks for your comment, our responses are as follows:
> > > > >
> > > > > **W#1**: We address the reviewer’s concern from the following two aspects.
> > > > > - First, as emphasized in Jones et al.’s paper: “$\cdots$ **If the expected improvement is less than $1$% of the best current function value (on the untransformed scale), we stop. $\cdots$**” (last line on Page 19 of their paper). This implies that their transformation methods were not used to deliberately transform the objective function with negative values into the positive objective space. Instead, they employed the log and inverse transformation methods to enhance the accuracy of the surrogate model (as discussed on Pages 14 and 19 of their paper). All in all, we do not think their transformation functions can be used to address the problem of the objective function with negative values as discussed in our previous response to your comment W#1.
> > > > > - As discussed in the related work section of our manuscript as well as our previous response, termination criteria for BO has been an understudied, but very important, topic. The peer algorithms used in our experiments are the baseline algorithms we believe are fair and useful in comparison.
> > > > >
> > > > > **Q#1**: We use $\tau=10$ in our experiments, but we forgot to clarify this in our manuscript. We promise to amend this in the final version. Furthermore, we have conducted a sensitivity analysis on $\tau$ for the range of $[2,18]$ in Section 4.3 of our manuscript. As the results shown in Figure 8 of our manuscript, we can see that our termination criterion is not sensitive across different $\tau$ values. Based on this observation, we believe it is flexible to choose the value of $\tau$ within the range $[2,18]$, which is broad enough.
> > > > >
> > > > > **Q#2**: In fact, $\omega$ and $\sigma_\epsilon$ are not additional parameters. Specifically, $\omega$ is a hyperparameter of the UCB acquisition function, while $\sigma_\epsilon$ is a hyperparameter of the GP.

---

### Official Review · Reviewer_TmDT · 2023-06-30

**Soundness:** 3 good
**Presentation:** 2 fair
**Contribution:** 4 excellent
**Rating:** 7
**Confidence:** 4

**Summary:**

The paper proposes a new termination criteria for Bayesian Optimization (BO). Early stopping is an important, and understudied problem in BO, as function evaluations tend to be expensive and therefore terminating the optimization (once a good enough solution has been reached) can save a large amount of resources.

The criteria is based on two stages: in the first stage we check whether the search process can be said to have converged to a convex hull. This is checked by testing the convexity of the black-box function in the area around the last $\tau$ observations, where the mean of the GP is used to approximate $f$ where we do not have observations. If this condition is fulfilled, we then calculate the local regret and determine if it falls below a pre-specified threshold.

The calculation of local regret requires finding the values of $\arg\max_{x \in \tilde{\Omega}} \mu(x) $ and $\arg\max_{x \in \tilde{\Omega}} \sigma^2(x) $. This may lead to numerically unstable results if done naively, so the authors propose using L-BFGS on the convex hull with termination criteria: $|| \nabla \mu(x) ||_2 \leq \lambda $ and $|| \sigma^2 ||_2 \leq \lambda $ and with specific initializations. Propositions and Lemmas 1 and 2 back-up their recommendations and show we should expect numerically stable results.

There is then a theoretical analysis of the termination criteria, showing (through Theorems 1 and 2) that if the acquisition function is properly optimized, then their calculation of local regret represents an upper bound on global regret. Otherwise, if there is numerical error when optimizing the acquisition function, then at worst their calculation of local regret will be an upper-bound on the regret restricted to the convex hull $\tilde{\Omega}$.

Experiment results are carried out against 4 benchmarks: (a) a naive method that stops the optimization once there is no improvement in observations for a specific amount of observations, (b) Nguyen's method that considers the maximum expected improvement, (c) Lorenz's method that considers the maximum probability of improvement, and (d) Makarova's method that considers the difference between the lower and upper confidence bounds as the termination indicator. They further compare on 3 synthetic benchmarks, 2 reinforcement learning tasks, and 5 hyper-parameter tuning problems.

The metrics used to compare are: (i) the empirical cumulative probability which explores how consistent the termination criteria is across a range of tasks, (ii) the relative computational cost, which calculates which percentage of the budget is used during the optimization, and (iii) the relative performance degradation, which measures the regret sacrificed by early stopping against letting the optimization continue.

The results show the following: the proposed method is the most consistent across tasks, having the same termination threshold leads to similar performance no matter the task (something other methods fail at doing). In addition, the proposed method has the smallest performance degradation for all methods except the naive one (which mostly uses all the budget), with the proposed method allowing us to save up to 80% of the budget. Finally, an empirical study shows that the method is robust to the choice of $\tau$, given $\tau > 4$.

**Strengths:**

Originality: The proposed algorithm is novel condition very different from any I have seen before. The idea of testing for convexity and comparing local regret is excellent, and the theoretical study to justify the choice of criteria complements the paper very well.

Quality: The ideas used in the paper are well justified, and explored thoroughly. Previous literature is considered, and compared against fully, and the empirical results test in a large variety of tasks, and further explore the behavior of the algorithm for different hyper-parameter settings.

Clarity: The paper has a difficult task of explaining many different things it is introducing, including previous work, theoretical results, and evaluation metrics. It does a great job at many of these things: it explains previous work, gives an _intuition_ of the meaning of the performance metrics, and the figures showcasing the results are excellent. Nonetheless, there are parts of the paper that lack clarity and perhaps can be improved (see later).

Significance: Given that Bayesian Optimization has seen a surge of interest in the physical sciences where experiments can be very costly, it is very important to have strong termination criteria to save costs as much as possible. The same holds true for most applications of Bayesian Optimization. As the authors suggest, this has been an understudied area, so papers like this one should be very useful to the larger scientific community.

**Weaknesses:**

- I found a few vital parts of the paper difficult to follow, and perhaps some improvements in the clarity of exposition could improve it. In particular it was difficult to (i) follow some of the proofs in the appendix, in particular the proofs of Theorems 1 and 2 could be expanded upon as it required me to write down a few equations manually to understand them. I was unable to understand the proof of Lemma 2. (ii) understand the intuition behind some of the choices made by the authors, and (iii) it is still unclear to me exactly why $I_{cdf}$ is constructed in such a way. I leave the details to the 'questions' section.

- The naive baseline seems to be implemented _too naively_, indeed the termination thresholds in L176 initially gave me the impression of being too large, and this was confirmed by Figure 6. It would be better if a more diverse set of thresholds were investigated, including smaller values and scaling with the dimension of the problem (indeed for 2d problems a small amount of non-improvements could lead to good results, but early termination on high dimensions). Indeed, most of the budget is used in all benchmarks, showing it is a poor baseline to compare against.

Minor weaknesses:

- Figure 1 was initially a little confusing. Perhaps it would be a good idea to include the algorithm's explanations of section 3.2 into the introduction.

While I list important weaknesses, which have an important impact on the score, I want to clarify that I thoroughly enjoyed the paper and I think it is great work. The low score stems from reluctance to fully recommend acceptance when I feel I do not understand important parts of the paper, but I look forward to discussing with the authors and other reviewers and having many of my doubts clarified.

**Questions:**

- What is the intuition behind initializing the L-BFGS for $\arg\max_{x \in \tilde{\Omega}} \sigma^2(x)$ at the maximum of the lower bound?

- In the proof of Lemma 2, the authors mention "give a stationary, isotropic kernel $k(\cdot, \cdot)$, we have: $k^2(x, x^i) \propto - ||x - x^i||^2_2$". Is the "$\propto$" symbol meant to represent 'proportional to', if so, why is it the case that the statement holds? It does not seem to hold for the RBF kernel. After a few readings, I think it means that the LHS if an increasing function of the RHS, but usually I have seen the symbol to mean direct linear proportionality so maybe it should be clarified.

- Furthermore, how is it that equation (19) follow from equation (18) _in the appendix_, it is not clear to me (relating to the proof of Lemma 2).

- Is there a more intuitive explanation of $I_{cdf}$? I found it difficult to understand what the CDF variable is. Indeed, $\tilde{\kappa}_i$ seems to be fully defined in L200, but it is treated as a variable in Figure 3. I am also failing to understand how the maximum and minimum values of $\kappa$ are calculated, is it empirically from the test runs? If so, what variable can we vary? This made understanding Figures 3 and 4 difficult. Figure 5 was clear though, and backs up the claims of the authors.

- The termination criteria in equation (7) seems to depend on the level of noise in the black-box function. What would be the effect of varying levels of noise?

- It is my understanding (from reading the appendix) that $\varepsilon$ in equation (14) is the numerical error when optimizing the acquisition function. Is this correct? If so, it would be good to include in the main paper.

- In my experience of working with early termination BO, I sometimes found that model misspecification can lead to early termination. For example, by underestimating the EI. How would you expect model misspecification to affect your algorithm?

- What is the computational overhead of calculating the stopping criteria for your method? For other methods it seems to be very cheap / free.

**Limitations:**

The authors do not address any limitations. It would be good for the authors, who know the work best, to try and mention the main ones.

---

> ### Author Rebuttal · Authors · 2023-08-09
>
> **W#1:** We will amend the proofs to improve their readability in the final version. The proof of Lemma 2 is elaborated in the response to **Q#3**, while the intuition of $\mathrm{I}_\mathrm{cdf}$ is explained in the response to **Q#4**.
>
> **W#2:** The choice of the termination thresholds are non-trivial. They are determined according to the statistical analysis of $\mathrm{I}_\mathrm{cdf}$ (elaborated in the response to **Q#4**). Note that we need to consider two conflicting objectives when designing an early termination criterion, i.e, cost as few FEs as possible but also minimize regret as much as possible. In this case, a small termination threshold has a risk of premature termination for the other termination criteria. This is reflected in the experiments on the Lunar Lander in Fig. 30 of Page 23 and the Swimmer in Fig. 43 of Page 36 in the supplementary file.
>
> **Minor:** We will amend a statement in the caption of Figure 1: "Please refer to Section 3.2 for a description of these termination criteria, as well as the meaning of $\kappa_\mathrm{PI}$, $\kappa_\mathrm{EI}$, and $\kappa_\mathrm{diff}$".
>
> **Q#1:** As the results shown in Table 1 of Page 9 in the supplementary file, we find that initializing the L-BFGS at the maximum of the lower bound is helpful to improve its performance lower bound when dealing with $\underset{\mathbf{x}\in\tilde{\Omega}}{\operatorname{argmax}}\ \underline{\sigma}^2(\mathbf{x})$, as opposed to a random initialization.
>
> **Q#2:** Yes, $\propto$ does not represent "proportional to". We will add a footnote to eq (17) to clarify this: "$\propto$ symbol indicates that $k^2\left(\mathbf{x},\mathbf{x}^i\right)$ will increase with $-\left|\mathbf{x}-\mathbf{x}^i\right|_2^2$".
>
> **Q#3:** To find the exact solution of $\underset{\mathbf{x}\in\tilde{\Omega}}{\operatorname{argmax}}\ \underline{\sigma}^2(\mathbf{x})$, we reformulate it as a bilevel optimization problem, i.e., eq (19) in the supplementary file. The lower-level optimization problem is applied to determine the $\hat{\Omega}$ that contains the $\underset{\mathbf{x}\in\tilde{\Omega}}{\operatorname{argmax}}\ \underline{\sigma}^2 (\mathbf{x})$. Once $\hat{\Omega}$ is determined, we can simplify $\underline{\sigma}^2(\mathbf{x})$ as $\underline{\sigma}^2(\mathbf{x})=-(k^2(\mathbf{x},\mathbf{x}^{1})+k^2(\mathbf{x},\mathbf{x}^{2}))$. This is implemented by ignoring samples outside of $\hat{\Omega}$ as well as the effects of $k(\mathbf{x},\mathbf{x})$ and $c$. Since $k(\cdot,\cdot)$ is stationary and isotropic, the $\underset{\mathbf{x}\in\hat{\Omega}}{\operatorname{argmax}}\ -(k^2(\mathbf{x},\mathbf{x}^{1})+k^2(\mathbf{x}, \mathbf{x}^{2}))$ coincides with $\underset{\mathbf{x}\in\tilde{\Omega}}{\operatorname{argmax}}\ -d\left(\mathbf{x},\mathbf{x}^1,\mathbf{x}^2\right)$, which has the analytical solution. Thus, in practice, we use $\underset{\mathbf{x}\in\hat{\Omega}}{\operatorname{minimize}}\ d\left(\mathbf{x},\mathbf{x}^1,\mathbf{x}^2\right)$ to replace $\underset{\mathbf{x}\in\tilde{\Omega}}{\operatorname{maximize}}\ \underline{\sigma}^2(\mathbf{x})$ as the upper-level optimization problem of eq (19). In addition, we plotted the trajectories of $\underline{\sigma}^2(\mathbf{x})$ and $d\left(\mathbf{x},\mathbf{x}^1,\mathbf{x}^2\right)$ in **Fig. 2 of the rebuttal PDF** to illustrate our rationality.
>
> **Q#4:** The motivation of $\mathrm{I}_\mathrm{cdf}$ (note that the variable is $\tilde{\kappa}_i$ instead of $\kappa$.) is to estimate the distribution of different termination indicators with regard to $\tilde{\kappa}_i$ collected from different problems in our experiments. Such statistics constitute the rationality of the threshold selection provided in Section 3.2. Let us use an illustrative example **Fig. 3 of the rebuttal PDF** on Ackley with 5 FEs to help your understanding.
>
> **Q#5:** The termination criterion in eq (7) is indeed affected by the level of noise. When the noise level ($\sigma_\epsilon$) is extremely low, the numerical error of with $\frac{\tilde{r}}{\omega \sigma_\epsilon}$ will be magnified. To reduce the effect of $\sigma_\epsilon$ and improve the numerical stability of our method during experimentation, we set a lower bound of $\sigma_\epsilon$ as 0.05 during the training of the GP.
>
> **Q#6:** Yes, $\varepsilon$ is a numerical error when optimizing the acquisition function. We will clarify this in the final version.
>
> **Q#7:** We encountered a similar issue in our research. When the model is overfitting, BO tends to converge within the local region of the current best solution. In this case, both **Conditions 1** and **2** are easily met while BO will be terminated prematurely. On the other hand, when the model is underfitting, BO will explore $\Omega$ in a random manner. In this case, satisfying **Condition 1** becomes challenging, and BO will face the risk of failing to be terminated.
>
> We designed three mitigation strategies: 1) restrict the lengthscale to $[0.05,200]$ during GP training to prevent lengthscales from becoming excessively large or small; 2) normalize the input of training data to $[0, 1]$; and 3) standardize the output of the training data by centering it on the mean and scaling it by the variance.
>
> **Q#8:** Although we did not have a rigorous complexity analysis, we can estimate the computational costs of both our method and Makarova's method is relatively more computationally demanding than the Nguyen's and the Lorenz's methods. This is because both of us need to solve two optimization problems while the other two only need to solve one.
>
> **Limitations:** We will add a discussion of limitations in the final version. The primary limitation of the proposed termination criterion is that it requires a predefined termination threshold, which needs to be determined based on prior knowledge or empirical observations. Although a recommended threshold selection range is given here, finding an optimal threshold that suits a wide range of optimization problems remains a challenge.

---

> > ### Comment · Reviewer_TmDT · 2023-08-10
> > **Response to Rebuttal**
> >
> > Thank you for the response, the rebuttal PDF in particular was very illustrative. It would be useful to include some of it in the main paper, but I understand there will be space restrictions.
> >
> > Regarding W2: I agree that choosing the threshold is non-trivial, and I am very impressed by your results regarding robustness to the termination threshold. I still think a larger range of thresholds could be explored for the naive baseline, but it is minor.
> >
> > Regarding Q7: Ah, very interesting. I think it would be good to have a small discussion in the main paper about model misspecification and how this issues help fix it. In particular for the benefit of practitioners. This does lead to a follow-up question though: the local regret should scale with the function's output range -- do you think the normalization plays a big part in the robustness of thresholds? If so, are the same normalization procedures applied to all methods?
> >
> > I think most of my questions have been answered, and I will update my score accordingly.
> >
> > Edit: It seems I am unable to update my original review for the moment. My evaluation would be: Rating: 7 (Accept); Confidence: 4.

---

> > > ### Author Response · Authors · 2023-08-12
> > > **Reply to "Response to Rebuttal from the Reviewer TmDT"**
> > >
> > > Thank you for your valuable feedback and for taking the time to review the rebuttal PDF. We appreciate your understanding of the space restrictions in the main paper, we will include all the figures of the rebuttal PDF in the supplementary.
> > >
> > > Regarding W2: We acknowledge your suggestion to explore a larger range of thresholds for the naive baseline. While we have prioritized a conservative range for a fair comparison, we will consider your suggestion for future work.
> > >
> > > Regarding Q7: We agree that discussing the role of model misspecification and its implications for practitioners is important. In our revised manuscript, we will include a discussion that highlights how our approach addresses this issue and its practical implications.
> > > As for the normalization procedures, yes, the same normalization methods are applied to all methods, ensuring a fair and consistent comparison. This normalization does indeed contribute to the robustness of thresholds.
> > >
> > > We really appreciate the reviewer’s affirmation for our work.

---

### Official Review · Reviewer_pzre · 2023-07-04

**Soundness:** 2 fair
**Presentation:** 1 poor
**Contribution:** 2 fair
**Rating:** 6
**Confidence:** 3

**Summary:**

This paper proposes a two-step termination method for Bayesian optimization. The proposed termination method first identifies if the optimization has converged within convex region, then it would stop the optimization procedure when estimated local regret is below predefined threshold. The authors compute estimated the local regret by solving a bilevel optimization problem, and theoretically prove that the regret is upper bounded by the defined local regret. Experiment shows that the proposed methods is more rubust and sample-efficient than other termination methods.

**Strengths:**

1. The idea is novel and easy to implemented to various problems.

2. The experiment is comprehensive for various benchmarks and evaluation metrics.

**Weaknesses:**

1. During the local regret estimation, the proposed method assumes convexity of the variance function, which is hard to satisfy even if the Condition 1 is satisfied.

2. The paper writing and presentation need to be improved, which I found difficult to follow. For example, in section 2, some intuition or explanation should be provided after the lemma or theorem is proposed. It would be better if the proposed method is highlighted in the result figures.

**Questions:**

1. As mentioned in weakness part, is the assumption of convex variance function easy to satisfied on the benchmarks used in this paper?

2. In Lemma 2,  why does the equation need to satisfy $\hat \Omega \cap D = \emptyset$ ?

3. What is the error bar in result figures?

**Limitations:**

The author addressed the limilations. I did not see potential negative societal impact of their work.

---

> ### Author Rebuttal · Authors · 2023-08-09
>
> **W\#1:** We do not assume the existence of convexity across the entire search space $\Omega$. Instead, as claimed in the **Proposition 2** of our manuscript, we suppose $\underset{\mathbf{x}\in\tilde{\Omega}}{\operatorname{minimize}}-\sigma^2(\mathbf{x})$ exhibits convexity in its local optimal region(s). Note that there have been studies in the Bayesian optimization community leverage the assumption of the existence of local convexity either in the underlying black-box optimization problem (Kirschner et al., ICML'19) or the Gaussian process model (McLeod et al., ICML'18). Their rationality have been validated both theoretically and empirically in these two works. Different, this work makes an assumption of the local convexity of the variance estimated by the GP model. This is backed up by our theoretical analysis and empirical study.
>
>
> **W\#2:** To facilitate the understanding of our theoretical analysis, we have provided remarks in our manuscript. In particular, **Remarks 1** to **3** explain the **Conditions 1** and **2**. **Remark 4** explains the **Theorems 1** and **2**. In the experimental part, since there are many trajectories generated by different peer methods. To mitigate potential confusion, each comparison method is presented in a subplot while their names are explained in the caption (Figures 3 to 6 of our manuscript). We appreciate the reviewer pointed out our presentation issues. We will further polish our manuscript in the camera-ready version.
>
>
> **Q\#1:** We believe our local convexity assumption of the variance estimated by the GP is always met in the benchmarks used in this work. This is backed up by our validation experiments, although they were not presented in our manuscript. From our experiments, we found that the Hessian matrices of the solutions of $\underset{\mathbf{x}\in\tilde{\Omega}}{\operatorname{minimize}}-\sigma^2(\mathbf{x})$ are almost positive definite, thus supporting the convexity. For the ease of illustration, we plot two examples in **Fig. 1 of the author rebuttal PDF**. It is easy to identify local convexity of $\sigma^2(\mathbf{x})$, the variance estimated by the GP. This observation applies to other more complex scenarios.
>
>
> **Q\#2:** We argue that the constraint $\hat{\Omega}\cap D=\emptyset$ is important. Based on this constraint, we can utilize the sample points of $\mathcal{D}$ to split $\tilde{\Omega}$ into several non-overlapping convex hulls, i.e., $\hat{\Omega}$. Since the volume of $\hat{\Omega}$ is proportional (though not strictly linear) to the extreme value of $\underset{\mathbf{x} \in \hat{\Omega}}{\operatorname{maximize}}\ \sigma^2(\mathbf{x})$, we can find the $\hat{\Omega}$ where the $\underset{\mathbf{x}\in\tilde{\Omega}}{\operatorname{argmax}}\ \underline{\sigma}^2(\mathbf{x})$ is located by solving the lower-level optimization problem defined in eq $(11)$. Thereafter, we can find $\underset{\mathbf{x}\in\tilde{\Omega}}{\operatorname{argmax}}\ \underline{\sigma}^2(\mathbf{x})$ by solving the corresponding upper-level optimization problem. On the other hand, without this constraint, the $\hat{\Omega}$ obtained by solving the lower-level optimization problem will contain all the sample points of $\mathcal{D}$. For this reason, the volume of $\hat{\Omega}$ is no longer proportional to the extreme value of $\underset{\mathbf{x} \in \hat{\Omega}}{\operatorname{maximize}}\ \sigma^2(\mathbf{x})$. This means that the $\underset{\mathbf{x}\in\tilde{\Omega}}{\operatorname{argmax}}\ \underline{\sigma}^2(\mathbf{x})$ cannot be found by solving the upper-level optimization problem.
>
>
> **Q\#3:** The error bars are used to show the difference between the upper and lower quarterlies of **$\frac{\tilde{\kappa}_i}{\tilde{\kappa}_i^{\operatorname{min}}}$** in Figure 4, **$\mathrm{I}_\mathrm{cost}$** in the left subplot of Figure 6 and 8, and **$\mathrm{I}_\mathrm{perf}$** in the right subplot of Figure 6 and 8 respectively.

---

> > ### Comment · Reviewer_pzre · 2023-08-13
> >
> > Thanks for your clarification, and the figure help me to understand the local convexity of variance function. I have raised my score.

---

> > > ### Author Response · Authors · 2023-08-17
> > >
> > > We really appreciate the reviewer’s affirmation for our work.

---

### Author Rebuttal · Authors · 2023-08-10

To facilitate the understanding of reviewers, we provide three figures in the submitted PDF file. Specifically, the Figure 1 corresponds to the response to **pzre Q\#1**,  the Figure 2 corresponds to the response to **TmDT Q\#3**, the Figure 3 corresponds to the response to **TmDT Q\#4** and **ffwe Q\#3**.

---

### Decision · Program_Chairs · 2023-09-21

**Decision:**

Accept (poster)

**Comment:**

This paper proposes a novel termination method in Bayesian optimization. It first detects whether the search is in a convex region by examining previously observed samples and then terminates the search if the local regret falls below a predetermined threshold. The method is well motivated, justified with theories, and evaluated empirically in a variety of tasks.

There are divergent opinions among reviewer ffwe and TmDT on whether this paper requires a major revision to address the following concerns.
1. Stopping conditions have been proposed in early works such as Jones et al. 1998 and Queipo, N. V., Pintos, S., & Nava, E. (2013), and used in practice. One reviewer considers the set of alternative baselines to be too narrow.

2. How to set hyperparameters of the proposed method. One reviewer finds the stopping criterion difficult to understand and modify in practice "from a user point of view". Particularly, the predetermined thresholds such as $\tau$ and $\eta_{lb}$ lack interpretation and are therefore hard to set by a user to control the behavior. But some reviewer consider it a practical method as the suggested hyperparameter values are robust across experiments.

3. Lack of clarity. This is a concern raised by all reviewers but some reviewers consider the rebuttal provide sufficient explanation. One of the remaining concerns is around the example in Figure 2 where the hyperparameter of GP is clearly mis-estimated based on the observation. It leads to multiple confusing questions: whether the region of the global optimum can be reliably identified based on the posterior, the misspecification is not transparent in the proposed stopping criterion, etc

Despite the remaining different views among reviewers on the aforementioned points after extensive discussion, this submission proposes a valuable novel approach to the important problem and provides both theoretical and empirical justification. The concerns on clarity and interpretation of hyperparameters should be able to be addressed without a major revision. I strongly suggest the authors to take into consideration all reviewers' comments and discussion, and prepare the revision accordingly.